# SHAP-Guided Kernel Actor-Critic for Explainable Reinforcement Learning

**Na Li**[1 2]  **Hangguan Shan**[1]  **Wei Ni**[3]  **Wenjie Zhang**[2]  **Xinyu Li**[4]

## Abstract

Actor-critic (AC) methods are a cornerstone of reinforcement learning (RL) but offer limited interpretability. Current explainable RL methods seldom use *state attributions* to assist training. Rather, they treat all state features equally, thereby neglecting the heterogeneous impacts of individual state dimensions on the reward. We propose *RKHS-SHAP-based Advanced Actor-Critic (RSA2C)*, an attribution-aware, kernelized, two-timescale AC algorithm, including Actor, Value Critic, and Advantage Critic. The Actor is instantiated in a vector-valued reproducing kernel Hilbert space (RKHS) with a Mahalanobis-weighted operator-valued kernel, while the Value Critic and Advantage Critic reside in scalar RKHSs. These RKHS-enhanced components use sparsified dictionaries: the Value Critic maintains its own dictionary, while the Actor and Advantage Critic share one. State attributions, computed from the Value Critic via RKHS-SHAP (kernel mean embedding for on-manifold and conditional mean embedding for off-manifold expectations), are converted into Mahalanobis-gated weights that modulate Actor gradients and Advantage Critic targets. We derive a global, non-asymptotic convergence bound under *state perturbations*, showing stability through the perturbation-error term and efficiency through the convergence-error term. Empirical results on three continuous-control environments show that RSA2C achieves efficiency, stability, and interpretability. Our code is available at https://github.com/Na-Li66/RSA2C.

[1]College of Information Science and Electronic Engineering, Zhejiang University, Hangzhou, China [2]School of Computer Science and Engineering, University of New South Wales, Sydney, Australia [3]School of Engineering, Edith Cowan University, Perth, Australia [4]School of Mechanical Science and Engineering, Huazhong University of Science and Technology, Wuhan, China. Correspondence to: Hangguan Shan <hshan@zju.edu.cn>.

*Proceedings of the 43$^{rd}$ International Conference on Machine Learning*, Seoul, South Korea. PMLR 306, 2026. Copyright 2026 by the author(s).

## 1. Introduction

Policy-gradient (PG) methods (Williams, 1992; Sutton et al., 1999) are a cornerstone of reinforcement learning (RL) (Sutton et al., 1998), directly optimizing the policy to maximize the expected discounted return. Actor-Critic (AC) algorithms (Konda & Tsitsiklis, 1999; Bhatnagar et al., 2009) reduce the variance of PG by pairing a policy (Actor) with a learned value estimator (Critic), thereby stabilizing on-policy training. However, standard AC (Romero et al., 2024) architectures remain *opaque*. The policy updates are driven by the advantage estimates of the multi-dimensional state, yet the influence of each dimension on the return is not explicitly revealed. This has fueled a critical and growing interest in explainable RL (XRL) mechanisms that can make optimization dynamics of AC more transparent.

Existing XRL methods can be broadly classified into *post-hoc* and *intrinsic* approaches. Post-hoc methods interpret a trained policy without altering its learning process, using tools such as saliency maps (Rosynski et al., 2020), counterfactual analyses (Gajcin & Dusparic, 2024), and Shapley-value (SHAP)-based explanations (Shapley, 1953; Lundberg & Lee, 2017; Li et al., 2021). Intrinsic methods build interpretability directly into the model, as in programmatic or symbolic policies (Verma et al., 2018), decision-tree policies (Li et al., 2024), or fuzzy-logic rule abstractions (Fu et al., 2024). However, they predominantly explain decisions at the *task*, *trajectory*, or *policy-structure* level. Very few methods provide the *dimension-level attributions*.

The structural coupling between state features and the PG update implies that AC requires *dimension-level attributions* that align with the additive property, which cannot be satisfied by methods, such as LIME (Ribeiro et al., 2018) and saliency map (Rosynski et al., 2020). Shapley values (Shapley, 1953; Lundberg & Lee, 2017), as the general metric to provide principled and model-agnostic attributions, uniquely satisfy axiomatic properties including efficiency, linearity, symmetry, and the dummy property, making them well-suited for AC methods. These properties align naturally with the additive structure of value and advantage functions, where the overall value contribution is built from the joint influence of individual state dimensions. However, classical SHAP-based methods rely on Monte Carlo sampling over coalitions, leading to high computational cost.

A scalable alternative is provided by RKHS-SHAP (Chau et al., 2022), which computes SHAP analytically using kernel mean embeddings (KMEs) and conditional mean embeddings (CMEs) in reproducing-kernel Hilbert spaces (RKHS). Instead of sampling coalitions explicitly, RKHS-SHAP represents coalition expectations as inner products between kernel embeddings, converting the exponential computation cost into tractable linear operations in a feature space and greatly reducing variance. In addition, the reproducing property of RKHS, as a non-parametric method, enables these expectations to be computed in closed form through inner products with KME/CME, yielding stable and low-variance attributions. Specifically, KME yields *observational* (on-manifold) SHAP by averaging over the empirical state distribution, ensuring that attributions reflect environment-consistent configurations. CME further enables *interventional* (off-manifold) SHAP by conditioning on subsets of dimensions while respecting the underlying data geometry, thereby probing the effect of feature coalitions without generating invalid states. These properties address the core limitations like computational scalability, robustness under correlation, and manifold consistency in classical SHAP, providing a principled and efficient mechanism for dimension-level attribution in AC.

Integrating RKHS-SHAP into AC algorithms introduces two fundamental challenges: *attribution reliability* and *optimization stability*. On the one hand, using RKHS-based attributions modifies the functional form of the Critic, potentially violating compatible function approximation and disrupting the contraction properties of the (soft) Bellman operator, especially when the kernel dictionary evolves over time. Such shifts in the underlying RKHS can introduce bias or drift into value estimation, compromising convergence guarantees. On the other hand, RL states are rarely clean or stationary in practice. They are perturbed by stochastic transition dynamics, sensing and communication noise, representation drift, and distributional shifts across training. These perturbations propagate into advantage estimation, making attribution-sensitive methods highly vulnerable to spurious correlations unless carefully regularized. As the attribution directly shapes the policy-gradient update, instability in SHAP values can amplify approximation errors, leading to oscillatory or brittle learning dynamics. Hence, the key research question is *how to design an efficient and provable attribution-aware RKHS-based AC that computes state attributions online while maintaining stability.*

### 1.1. Contribution

This paper proposes *RKHS-SHAP-based Advanced Actor-Critic (RSA2C)*, an attribution-aware two-timescale AC that transforms state attributions into a Mahalanobis distance of the policy. By grounding attribution computation and policy updates in RKHS, RSA2C provides a principled mechanism

to highlight influential state dimensions, stabilize gradient updates. This explainable algorithm delivers stable and efficient training for continuous control with theoretical guarantees. Our contributions are summarized as follows:

**(i) Kernelized two-timescale Actor-Critic.** To jointly improve decision efficiency and interpretability, we propose *RSA2C*, an RKHS-enhanced AC framework that embeds an Actor and two Critics in an RKHS and injects adaptive state feature importance via RKHS-SHAP directly into a Mahalanobis-weighted operator-valued kernel (OVK), which models vector-valued policies and encodes correlations among state dimensions. The *Actor* is instantiated in an OVK RKHS with Mahalanobis weights, while the *Value Critic* for stable temporal difference (TD)-based policy evaluation and *Advantage Critic* constructed with compatible features are scalar RKHSs to approximate value and advantage functions, respectively. Computational complexity is controlled by a sparse dictionary maintained via approximate linear dependence (ALD). Its computation is closed-form and scales linearly with the controllable dictionary size, making it lightweight and suitable for online RL.

**(ii) State attribution to learning signal via RKHS-SHAP.** We compute SHAP *from Value Critic* using two RKHS-SHAP routes, i.e., KME for on-manifold expectations (RSA2C-KME) and CME for off-manifold expectations (RSA2C-CME). These signals are gated by Mahalanobis weights, and injected into the *Actor* and the *Advantage Critic* targets to modulate updates with budgeted online cost. Under this state-level auxiliary signal, RSA2C achieves both efficiency and intrinsic interpretability.[1]

**(iii) Global convergence under state perturbations.** We establish a global, non-asymptotic convergence bound for RSA2C under *state perturbations*. This is achieved by decomposing the learning gap into a perturbation error and a convergence error, which together quantify stability and efficiency. Moreover, the perturbation error is divided into an attribution-induced term and a policy-learning term, demonstrating stability. The convergence error is divided into a refined tracking term and a two-timescale approximation term, demonstrating efficiency.

**(iv) Empirics and intrinsic interpretability.** We conduct simulations on three standard continuous-control environments with a focus on effectiveness, stability, and interpretability. Results show that RSA2C improves returns

---

[1]The RKHS-SHAP used in RSA2C preserves the key interpretability properties of classical SHAP, including additive and consistent feature attribution. These attribution signals are not used solely for post-hoc explanation, but are directly incorporated into the learning process through the Mahalanobis-weighted kernel, which modulates both the Actor and Advantage Critic updates. Therefore, feature importance not only explains the learned policy but also actively shapes it during optimization.

while preserving intrinsic interpretability, incurring only a modest runtime overhead. Besides, RSA2C-CME maintains stable performance for various state perturbations, showing strong robustness to stochastic disturbances.

### 1.2. Related Works

**Explainable RL.** XRL generally includes post-hoc and intrinsic explainability (Milani et al., 2024). Post-hoc attribution methods from supervised learning have inspired XRL tools, such as saliency maps (Rosynski et al., 2020), RKHS-SHAP (Chau et al., 2022), LIME (Ribeiro et al., 2016; 2018), Integrated Gradients (Sundararajan et al., 2017), and SmoothGrad (Smilkov et al., 2017). However, they typically describe learned behavior, not utilizing it during training. Intrinsic interpretability lines constrain the policy class, e.g., rule-based policies (Bastani et al., 2018; Li et al., 2024) or decision-tree (Custode & Iacca, 2023), programmatic policies (Verma et al., 2018), and logic controller (Hein et al., 2018). These approaches improve transparency but ignore the ability of the learning loop.

**RKHS-based RL.** Kernel methods yield nonparametric function approximation with explicit control of geometry and capacity. Recent kernel RL provides finite-time guarantees and provably efficient algorithms in RKHS (Domingues et al., 2021). Sample-efficient GP-based Critics continue to advance via sparse or ensemble designs (Polyzos et al., 2021; Zhang et al., 2020). Vector-valued kernels further enable multi-output policies and critics, with recent theory and online identification methods (Alvarez et al., 2012). Yet, these kernel-based works lack transparency in decision.

**Two-timescale AC.** Early AC was formalized by Konda & Tsitsiklis (1999) and later extended to natural AC (NAC) using the natural policy gradient (Bhatnagar et al., 2009). Existing work has established asymptotic convergence for two-timescale AC/NAC under both independent and identically distributed and Markovian sampling (Agarwal et al., 2020; Hu et al., 2022; Cayci et al., 2024) or non-asymptotic convergence (Xu et al., 2020), yet non-asymptotic convergence guarantees for two-timescale RKHS-enhanced AC remains largely open gaps.

## 2. Preliminaries

### 2.1. Markov Decision Process

Consider a Markov decision process (MDP) $\mathcal{M} = (\mathcal{S}, \mathcal{A}, \mathcal{T}, r, \gamma)$ with discount factor $\gamma \in (0, 1)$, state space $\mathcal{S} \subseteq \mathbb{R}^d$, and action space $\mathcal{A} \subseteq \mathbb{R}^m$. At each time $t \geq 0$, the agent observes state $\mathbf{s}_t \in \mathcal{S}$, selects action $\mathbf{a}_t \sim \pi(\cdot \mid \mathbf{s}_t)$ with policy $\pi(\cdot \mid \mathbf{s}_t) \in \Delta(\mathcal{A})$, receives reward $r(\mathbf{s}_t, \mathbf{a}_t) \in [0, 1]$, and transitions to next state $\mathbf{s}_{t+1} \sim \mathcal{T}(\cdot \mid \mathbf{s}_t, \mathbf{a}_t)$, where $\mathcal{T}(\cdot \mid \mathbf{s}_t, \mathbf{a}_t)$ is a Markov kernel on $\mathcal{S}$. For conciseness, we write $r_{t+1} = r(\mathbf{s}_t, \mathbf{a}_t)$. The value and action-

value functions are $V^\pi(\mathbf{s}) = \mathbb{E}[\sum_{k=0}^\infty \gamma^k r_{k+1} \mid \mathbf{s}_0 = \mathbf{s}]$ and $Q^\pi(\mathbf{s}, \mathbf{a}) = \mathbb{E}[\sum_{k=0}^\infty \gamma^k r_{k+1} \mid \mathbf{s}_0 = \mathbf{s}, \mathbf{a}_0 = \mathbf{a}]$, respectively, so that both $V^\pi$ and $Q^\pi$ quantify the expected accumulated reward over the entire horizon. The advantage function is defined as $A^\pi(\mathbf{s}, \mathbf{a}) := Q^\pi(\mathbf{s}, \mathbf{a}) - V^\pi(\mathbf{s})$.

Define the discounted visitation distribution as $d_\gamma^\pi(\mathbf{s}) = (1 - \gamma) \sum_{t=0}^\infty \gamma^t \Pr(\mathbf{s}_t = \mathbf{s} \mid \pi, \rho_0)$ with $d_\gamma^\pi(\mathbf{s}, \mathbf{a}) = d_\gamma^\pi(\mathbf{s})\pi(\mathbf{a} \mid \mathbf{s})$ and the initial-state distribution $\rho_0$. The objective of RL is $J(\pi) = \frac{1}{1-\gamma}\mathbb{E}_{(\mathbf{s}, \mathbf{a}) \sim d_\gamma^\pi}[r(\mathbf{s}, \mathbf{a})] = \mathbb{E}_{\mathbf{s}_0 \sim \rho_0}[V^\pi(\mathbf{s}_0)]$. The discounted visitation rewrites time-averaged expectations as expectations w.r.t. distribution $d_\gamma^\pi$, which can be estimated using on-policy rollouts when the policy drifts slowly.

### 2.2. RKHS-SHAP

**Shapley value.** Let $\mathcal{X} = \{1, \dots, d\}$ index input features and $\mathcal{C} \subseteq \mathcal{X}$ be a coalition. For the input $\mathbf{x} \in \mathbb{R}^d$, a cooperative game is specified by a characteristic value function $v_\mathbf{x} : 2^\mathcal{X} \to \mathbb{R}$. The Shapley value for feature $i \in \mathcal{X}$ is
$\phi_i(v_\mathbf{x}) = \sum_{\mathcal{C} \subseteq \mathcal{X} \setminus \{i\}} \frac{|\mathcal{C}|!(d - |\mathcal{C}| - 1)!}{d!} \left( v_\mathbf{x}(\mathcal{C} \cup \{i\}) - v_\mathbf{x}(\mathcal{C}) \right)$,
which is exponential to compute naively. Shapley values average a feature's marginal contribution over all coalitions and satisfy efficiency, symmetry, null-player, and additivity.

**KernelSHAP.** Given a predictive model $f : \mathbb{R}^d \to \mathbb{R}$ and input $\mathbf{x}$, KernelSHAP (Lundberg & Lee, 2017) estimates $\{\phi_i(v_\mathbf{x})\}_{i=1}^d$ by fitting a locally linear surrogate to sampled coalitions with a Shapley-inspired kernel, avoiding retraining $f$ on all subsets. To define $v_\mathbf{x}(\mathcal{C})$, one must complete the unobserved coordinates $\bar{\mathcal{C}} = \mathcal{X} \setminus \mathcal{C}$:

$$v_\mathbf{x}(\mathcal{C}) = \mathbb{E}_{\mathbf{x}' \sim p(\mathbf{x}')} \left[ f \left( \text{concat}(\mathbf{x}^\mathcal{C}, \mathbf{x}'^{\bar{\mathcal{C}}}) \right) \right], \quad (1a)$$

$$v_\mathbf{x}(\mathcal{C}) = \mathbb{E}_{\mathbf{x}' \sim p(\mathbf{x}' \mid \mathbf{x}^c)} \left[ f(\mathbf{x}') \right], \quad (1b)$$

where (1a) corresponds to the off-manifold (interventional) SHAP and the on-manifold (observational) (1b) preserves feature correlations (Štrumbelj & Kononenko, 2014). Here, $\text{concat}(\cdot, \cdot)$ denotes the vector obtained by keeping the coordinates in $\mathcal{C}$ from $\mathbf{x}$ and imputing the remaining coordinates from $\mathbf{x}'$. In practice, off-manifold imputations are simple but may break natural dependencies (e.g., kinematics or physical constraints). On-manifold imputations respect correlations by conditioning on observed coordinates, but require conditional distributions, which are hard to model in high dimensions and expensive to estimate online.

**RKHS-SHAP.** By embedding marginal and conditional distributions into an RKHS, RKHS-SHAP (Chau et al., 2022) circumvents explicit density models and enables analytic evaluation of the expectations in (1). Let $k : \mathbb{R}^d \times \mathbb{R}^d \to \mathbb{R}$ be a bounded positive-definite kernel with feature map $\psi : \mathbb{R}^d \to \mathcal{H}_k$. The KME of $P_\mathbf{x}$ is $\mu_\mathbf{x} := \mathbb{E}[\psi(\mathbf{x})] \in \mathcal{H}_k$, with empirical estimate $\widehat{\mu}_\mathbf{x} = \frac{1}{n} \sum_{i=1}^n \psi(\mathbf{x}_i)$ (Muan-

det et al., 2017). For CME, we require the distribution of the missing block given the observed block; let $\mathbf{x}_{\bar{C}}$ be the unobserved coordinates and $\mathbf{x}_C$ the observed ones. With a kernel on $\mathbb{R}^{|\bar{C}|}$ and feature map $\varphi$, the CME is defined as $\mu_{\mathbf{x}_{\bar{C}}|\mathbf{x}_C} = \mathbb{E}[\varphi(\mathbf{x}_{\bar{C}}) \mid \mathbf{x}_C]$.

Assuming a product kernel $k = \prod_{i=1}^{d} k^{(i)}$ so that $\mathcal{H}_k = \bigotimes_{i=1}^{d} \mathcal{H}_{k^{(i)}}$, and assuming $f \in \mathcal{H}_k$, the coalition value functionals admit RKHS inner-product forms:

$$v_{\mathbf{x}}(\mathcal{C}) = \langle f, \, v^{\mathcal{C}}(\mathbf{x}) \rangle_{\mathcal{H}_k}, \tag{2}$$

$$v^{\mathcal{C}}(\mathbf{x}) = \begin{cases} \left( \bigotimes_{i \in \mathcal{C}} \psi^{(i)}(x_i) \right) \otimes \mu_{\mathbf{x}_{\bar{C}}}, & \text{off-manifold,} \\ \left( \bigotimes_{i \in \mathcal{C}} \psi^{(i)}(x_i) \right) \otimes \mu_{\mathbf{x}_{\bar{C}}|\mathbf{x}_C}, & \text{on-manifold.} \end{cases}$$

Here, $\otimes$ denotes the tensor (outer) product of feature maps. Formally, for any coordinates $i \in \mathcal{C}$ and $i \in \bar{\mathcal{C}}$, the tensor (outer) product feature map is defined as $\otimes_{i \in \mathcal{C}} \psi^{(i)}(x_i) \in \otimes_{i \in \mathcal{C}} \mathcal{H}_{k^{(i)}}$, with $\langle \bigotimes_{i \in C} \psi^{(i)}(x_i), \bigotimes_{i \in C} \psi^{(i)}(x'_i) \rangle = \prod_{i \in C} \langle \psi^{(i)}(x_i), \psi^{(i)}(x'_i) \rangle_{\mathcal{H}_{k^{(i)}}}$. The expectations in (1) reduce to inner products in $\mathcal{H}_k$ not conditional densities.

**Computational cost.** Exact CMEs require inverses of Gram operators and cost $\mathcal{O}(n^3)$ for $n$ samples. With Random Fourier Feature (RFF) (Rahimi & Recht, 2007) or Nyström method (Yang et al., 2012), one can reduce the complexity of evaluating empirical CME from $\mathcal{O}(n^3)$ to $\mathcal{O}(q^2 n + q^3)$ (Muandet et al., 2017), where $q$ is the feature map dimension and typically, $q \ll n$ (Li et al., 2019).

## 3. Algorithm Design of RSA2C

To jointly improve decision efficiency and interpretability, we propose RSA2C, a kernel-based algorithm that embeds an Actor and two Critics in RKHS and injects adaptive state feature importance via RKHS-SHAP directly into a Mahalanobis-weighted OVK. We utilize the ALD-based sparse dictionary to control time complexity. Here, RKHS-SHAP is not merely used for explanation. Its attributions, derived from the value approximation, emphasize stable and influential state dimensions, producing smoother advantages and more stable policy-gradient updates. An overview is given in Figure 1 and illustrated in Algorithm 1.

**Architecture.** RSA2C comprises three RKHS-enhanced components with two-timescale mechanisms: (i) an *Actor* whose stochastic Gaussian policy has its mean represented in a *vector-valued RKHS* $\mathcal{H}_K$; (ii) an *Advantage Critic* that estimates $A(\mathbf{s}, \mathbf{a})$ in a *scalar-valued RKHS* with *compatible features* (aligned with $\nabla \log \pi$) to yield low-variance policy gradients; (iii) a *Value Critic* that estimates $V(\mathbf{s})$ in a scalar-valued RKHS via the TD method. A restart-type transition kernel is incorporated to improve mixing and stabilize targets, i.e., $\mathcal{T}_\gamma(\mathbf{s}' \mid \mathbf{s}, \mathbf{a}) := (1 - \gamma)\rho_0(\mathbf{s}') + \gamma \mathcal{T}(\mathbf{s}' \mid \mathbf{s}, \mathbf{a})$. For any policy $\pi$, the stationary state distribution of the

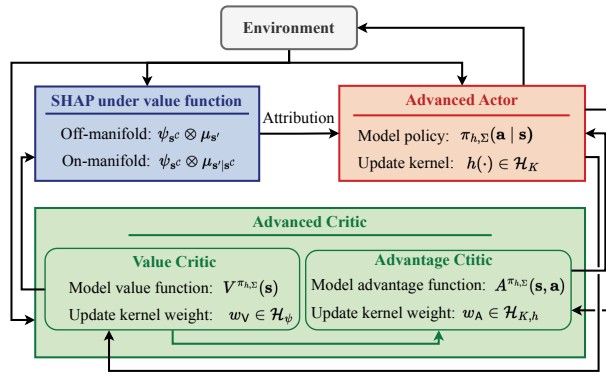

*Figure 1.* Overview diagram of RSA2C consisting of Actor, Value Critic and Advantage Critic.

---

**Algorithm 1** RSA2C

---

**input** Discount factor $\gamma$, kernels for Actor and Value Critic, stepsizes $\alpha_t^{\mathsf{h}}$ and $\alpha_t^{\mathsf{v}}$;

1: **initialize**: Initial state $\mathbf{s}_0$, empty dictionaries $\mathcal{D}_{\mathsf{A}}$ and $\mathcal{D}_{\mathsf{V}}$, covariance matrix $\Sigma_0$, Actor's feature mapping $h(\mathbf{s}_0) = 0$, Value Critic's feature mapping $w_{\mathsf{V}}$, and weight of Advantage Critic $w_{\mathsf{A}}$;
2: **for** Epoch $t = 1, \dots, T$ **do**
3:     Collect data from the system using policy $\pi_{h,\Sigma}$ with $\mathbf{s}_t \sim \mathcal{T}_\gamma(\cdot \mid \mathbf{s}_{t-1}, \mathbf{a}_{t-1})$;
4:     Compute SHAP under value function using (10a) and obtain the weighted kernel by (3);
5:     Optimize the mean of Actor by $h = h + \alpha_t^{\mathsf{h}} \nabla_h J(h, \Sigma)$ with $\nabla_h J(h, \Sigma)$ defined in (4);
6:     (Optional) Reduce the covariance $\Sigma$ of Actor;
7:     Optimize Value Critic by $\psi = \psi + \alpha_t^{\mathsf{v}} \nabla_\psi J(\psi)$ with $\nabla_{w_{\mathsf{V}}} J(w_{\mathsf{V}})$ defined in (8);
8:     Sparsify the dictionary sets $\mathcal{D}_{\mathsf{A}}$ and $\mathcal{D}_{\mathsf{V}}$;
9:     Update the weight of Advantage Critic;
10: **end for**

**output** policy $\pi_{h,\Sigma}$.

---

Markov chain induced by $\mathcal{T}_\gamma$ coincides with $d_\gamma^\pi$ (i.e., expectations under $d_\gamma^\pi$ equal stationary expectations under $\mathcal{T}_\gamma$). This equivalence facilitates writing policy gradients and TD objectives w.r.t. $d_\gamma^\pi$ while estimating the expectations online using streaming samples from the on-policy chain.

**On-policy two-timescale mechanism.** RSA2C is an online, on-policy method, which means rollouts at each iteration are sampled from the current policy $\pi_{h,\Sigma}$ until termination. Parameters are updated on mini-batches drawn from the most recent on-policy trajectories. Learning proceeds on two-timescale schedules: a *fast* timescale for the Value Critic and a *slow* one for the Actor. Since the SHAP attribution is computed from the Value Critic and then used to modulate the Actor and Advantage Critic updates, the critic must evolve on a faster timescale for attribution to remain suffi-

ciently stable before the policy is updated. Let $\alpha_t^{\mathsf{h}}$ and $\alpha_t^{\mathsf{v}}$ be the Actor and Value Critic stepsizes, respectively, satisfying

$$\sum_t \alpha_t^{\mathsf{h}} = \infty, \quad \sum_t (\alpha_t^{\mathsf{h}})^2 < \infty,$$
$$\sum_t \alpha_t^{\mathsf{v}} = \infty, \quad \sum_t (\alpha_t^{\mathsf{v}})^2 < \infty, \qquad \alpha_t^{\mathsf{v}}/\alpha_t^{\mathsf{h}} \to 0.$$

**SHAP-aware kernelization.** SHAP scores rescale state features inside the Mahalanobis weights, so importance weighting influences similarities, gradients, and parameter updates, not only providing a post-hoc explanation in prior works. Expectations needed for on-manifold or off-manifold imputations are computed via KME and CME as in RKHS-SHAP, avoiding explicit density models.

### 3.1. Advanced Actor-Critic Framework

We adopt three RKHS-enhanced components on *two time-scales*: a kernelized Gaussian *Actor* in a vector-valued RKHS, an *Advantage Critic* with compatible features sharing the Actor dictionary, and a *Value Critic* in a scalar RKHS with its *own* dictionary. Online sparsification for both dictionaries is performed via the ALD method. First, we define a new kernel as follows.

**Definition 3.1** (Adaptive Mahalanobis-weighted OVK)**.** Let $\mathcal{S} \subseteq \mathbb{R}^d$ denote the state space, and $\Sigma_{\mathsf{K}} \succeq 0$ be a positive semidefinite operator. Define the scalar kernel

$$\kappa_{\boldsymbol{\phi}}(\mathbf{s}, \mathbf{s}_j) = \exp\left(-\tfrac{1}{2}(\mathbf{s} - \mathbf{s}_j)^\top \mathbf{W}(\mathbf{s} - \mathbf{s}_j)\right), \quad (3)$$

where $\mathbf{W} = \mathrm{diag}(\tilde{\boldsymbol{\phi}})$ with $\tilde{\phi}_i = \max(\phi_i, \varepsilon_0)$ for SHAP-based importances $\phi_i \geq 0$ and a small floor $\varepsilon_0 > 0$ ensuring $\mathbf{W} \succ 0$. The corresponding OVK is $K(\mathbf{s}, \mathbf{s}_j) = \kappa_{\boldsymbol{\phi}}(\mathbf{s}, \mathbf{s}_j)\Sigma_{\mathsf{K}}$. Since $\kappa_{\boldsymbol{\phi}}$ is positive definite (PD) and $\Sigma_{\mathsf{K}} \succeq 0$, kernel $K$ is operator-valued PD (Micchelli & Pontil, 2005).

Unlike Lever & Stafford (2015), which treats all state features equally, $\mathbf{W}$ adapts the base kernel to heterogeneous feature relevance via RKHS-SHAP-derived importances. Notably, $\mathbf{W}$ is diagonal only in its matrix representation, but the quantities placed on the diagonal originate from interaction-aware SHAP features, and thus do not discard feature correlations. If cross-feature interactions are desired, $\mathbf{W}$ may be generalized from diagonal to a full SPD matrix; we adopt a diagonal $\mathbf{W}$ for robustness and efficiency. Also, we use two dictionaries for RSA2C: $\mathcal{D}_{\mathsf{A}} = \{\mathbf{s}_j\}_{j=1}^{q_{\mathsf{A}}}$ shared by the *Actor* and the *Advantage Critic*, and $\mathcal{D}_{\mathsf{V}} = \{\tilde{\mathbf{s}}_j\}_{j=1}^{q_{\mathsf{V}}}$ used solely by the *Value Critic*. Both are maintained online by ALD with standard residual result; see Appendix B.1.

**Advanced Actor.** With $\mathcal{D}_{\mathsf{A}} = \{\mathbf{s}_j\}_{j=1}^{q_{\mathsf{A}}}$ and stack the coefficients $\mathbf{C} = [\mathbf{c}_1, \cdots, \mathbf{c}_{q_{\mathsf{A}}}]^\top \in \mathbb{R}^{q_{\mathsf{A}} \times m}$ with $\mathbf{c}_j \in \mathbb{R}^m$. Under Definition 3.1, we represent the policy mean $h : \mathcal{S} \to \mathcal{A}$ in the vector-valued RKHS $\mathcal{H}_K$ as $h(\mathbf{s}) =$

$\sum_{j=1}^{q_{\mathsf{A}}} K(\mathbf{s}, \mathbf{s}_j)\mathbf{c}_j$. The Gaussian policy is

$$\pi_{h,\Sigma}(\mathbf{a} \mid \mathbf{s}) = \mathcal{N}\left(h(\mathbf{s}), \Sigma\right), \quad (4)$$

where $\Sigma \in \mathbb{R}^{m \times m}$ is a positive-definite covariance matrix. The policy gradient with an advantage baseline w.r.t. the discounted visitation distribution is

$$\nabla_h J(h, \Sigma) = \tfrac{1}{1-\gamma} \mathbb{E}_{(\mathbf{s}, \mathbf{a}) \sim d_\gamma^{\pi_{h,\Sigma}}}\left[A_{w_{\mathsf{A}}}^{\pi_{h,\Sigma}}(\mathbf{s}, \mathbf{a})\nabla_h \log \pi_{h,\Sigma}(\mathbf{a} \mid \mathbf{s})\right],$$

where $A_{w_{\mathsf{A}}}^{\pi_{h,\Sigma}}$ is the approximate advantage function in (5). We update RKHS policy with online sparsification and Fréchet gradient $\nabla_h J(h, \Sigma)$; see Appendices B.1 and B.2.

**Advanced Critic.** We estimate advantage and value with kernel approximators designed for policy-gradient compatibility, with the proof in Appendix B.3. The Advantage Critic is established under compatible features on dictionary $\mathcal{D}_{\mathsf{A}}$ and OVK $K$, giving scalar-valued kernel $K_h\left((\mathbf{s}, \mathbf{a}), (\mathbf{s}_j, \mathbf{a}_j)\right) = \left(\mathbf{a} - h(\mathbf{s})\right)^\top \Sigma^{-1/2} K(\mathbf{s}, \mathbf{s}_j)\Sigma^{-1/2}\left(\mathbf{a}_j - h(\mathbf{s}_j)\right)$, which is positive semidefinite if $K$ is PD. With the feature map $\nu(\mathbf{s}, \mathbf{a}) = K(\mathbf{s}, \cdot)\Sigma^{-1/2}\left(\mathbf{a} - h(\mathbf{s})\right) \in \mathcal{H}_K$, we approximate

$$A_{w_{\mathsf{A}}}^{\pi_{h,\Sigma}}(\mathbf{s}, \mathbf{a}) = \left\langle w_{\mathsf{A}}, \nu(\mathbf{s}, \mathbf{a})\right\rangle_{\mathcal{H}_K}, \quad (5)$$

and estimate $w_{\mathsf{A}}$ in closed-form with the ALD method.

For Value Critic, we avoid applying RKHS-SHAP reweighting to prevent circular dependence between the value estimates and their own attribution scores. Keeping the Value Critic unweighted avoids error amplification caused by biased SHAP values and provides a more stable baseline for policy evaluation. Using a scalar RKHS $\mathcal{H}_k$ with kernel $k(\mathbf{s}, \mathbf{s}_j) = \langle \psi(\mathbf{s}), \psi(\mathbf{s}_j) \rangle$ and dictionary $\mathcal{D}_{\mathsf{V}}$, we set

$$V_{w_{\mathsf{V}}}^{\pi_{h,\Sigma}}(\mathbf{s}) = \left\langle w_{\mathsf{V}}, \psi(\mathbf{s})\right\rangle_{\mathcal{H}_k} = \sum_{j \in \mathcal{D}_{\mathsf{V}}} \eta_j k(\mathbf{s}, \mathbf{s}_j), \quad (6)$$

with $w_{\mathsf{V}} = \sum_{j \in \mathcal{D}_{\mathsf{V}}} \eta_j \psi(\mathbf{s}_j)$ fitted by minimizing the regularized TD loss

$$J(w_{\mathsf{V}}) = \mathbb{E}\left[\tfrac{1}{2}\left(V_{w_{\mathsf{V}}}(\mathbf{s}) - (r(\mathbf{s}, \mathbf{a}) + \gamma V_{w_{\mathsf{V}}}(\mathbf{s}'))\right)^2\right] + \tfrac{\lambda}{2}\|w_{\mathsf{V}}\|_{\mathcal{H}_k}^2, \quad (7)$$
$$\nabla_{w_{\mathsf{V}}} J(w_{\mathsf{V}}) = \mathbb{E}\left[\left(V_{w_{\mathsf{V}}}(\mathbf{s}) - r(\mathbf{s}, \mathbf{a}) - \gamma V_{w_{\mathsf{V}}}(\mathbf{s}')\right)\psi(\mathbf{s})\right] + \lambda w_{\mathsf{V}}. \quad (8)$$

### 3.2. SHAP Computing.

We compute state feature attributions from the Value Critic via RKHS-SHAP. Let $\mathcal{X} = \{1, \ldots, d\}$ index state features and $\mathcal{C} \subseteq \mathcal{X}$ be a coalition. From (6), the coalition value is

$$v_{\mathbf{s}}(\mathcal{C}) = \mathbb{E}\left[V_{w_{\mathsf{V}}}^{\pi_{h,\Sigma}}(\mathrm{concat}(\mathbf{s}^{\mathcal{C}}, \mathbf{s}'^{\bar{\mathcal{C}}}))\right] = \left\langle w_{\mathsf{V}}, \mu_{\mathcal{C}}(\mathbf{s})\right\rangle_{\mathcal{H}_k}, \quad (9)$$

where the expectation follows either the off-manifold or on-manifold imputations, along with

$$\mu_{\mathcal{C}}^{(\mathrm{off})}(\mathbf{s}) = \left(\otimes_{i \in \mathcal{C}} \psi^{(i)}(\mathbf{s}_i)\right) \otimes \mu_{\mathbf{s}_{\bar{\mathcal{C}}}}; \quad (10a)$$
$$\mu_{\mathcal{C}}^{(\mathrm{on})}(\mathbf{s}) = \left(\otimes_{i \in \mathcal{C}} \psi^{(i)}(\mathbf{s}_i)\right) \otimes \mu_{\mathbf{s}_{\bar{\mathcal{C}}} \mid \mathbf{s}_{\mathcal{C}}}, \quad (10b)$$

with $\mu_{\mathbf{s}_{\bar{C}}}$ and $\mu_{\mathbf{s}_{\bar{C}}|\mathbf{s}_C}$ the (conditional) mean embeddings estimated via KME/CME, leading to RSA2C-KME/RSA2C-CME. Finally, the SHAP attribution for feature $i$ is

$$\phi_i(v_{\mathbf{s}}) = \sum_{\mathcal{C} \subseteq \mathcal{X} \setminus \{i\}} \frac{|\mathcal{C}|!(d - |\mathcal{C}| - 1)!}{d!} \Big(v_{\mathbf{s}}(\mathcal{C} \cup \{i\}) - v_{\mathbf{s}}(\mathcal{C})\Big). \quad (11)$$

## 4. Theoretical Guarantees

In this section, we show the stability and efficiency of RSA2C; see Theorems 4.8 and 4.10. Notably, Lemma C.3 in Appendix C.2 shows the stability of the RKHS-SHAP score produced by Value Critic. For conciseness, let $\pi_h$ denote $\pi_{h,\Sigma}$ herein. We first specify the assumptions considered.[2]

**Assumption 4.1** (Deterministic and Markovian adversary). $b(\mathbf{s})$ is a deterministic function $b: \mathcal{S} \to \mathcal{S}$, which only depends on the current state $\mathbf{s}$, and does not change over time.

Given the same $\mathbf{s}$, the adversary generates the same (stationary) perturbation. If the adversary can perturb a state $\mathbf{s}$ arbitrarily without bounds, the problem is trivial. To fit our analysis to the most realistic settings, we define perturbation set $B(\mathbf{s})$, to restrict the adversary to perturb a state $\mathbf{s}$ only to a predefined set of states:

**Definition 4.2** (Adversary perturbation set). Define a set $B(\mathbf{s})$ containing all allowed perturbations of the adversary, i.e., $b(\mathbf{s}) \in B(\mathbf{s})$ where $B(\mathbf{s})$ is a set of states and $\mathbf{s} \in \mathcal{S}$.

Here, $B(\mathbf{s})$ is usually a set of task-specific "neighboring" states of $\mathbf{s}$ (e.g., bounded sensor measurement errors), which makes the observation still meaningful (yet not accurate) even with perturbations. We first introduce Proposition 4.3 with its proof in Appendix C.3.

**Proposition 4.3.** *For all $h \in \mathcal{H}$, the Fisher information matrix induced by policy $\pi_h$ and initial state distribution $\rho_0$ satisfies: For some constant $\lambda_F > 0$,*

$$\mathbf{F}(h) = \mathbb{E}_{\nu_{\pi_h}}[\nabla_h \log \pi_h(\mathbf{a}|\mathbf{s}) \nabla_h \log \pi_h(\mathbf{a}|\mathbf{s})^\top] \succeq \lambda_F \cdot \mathbf{I}_m.$$

**Assumption 4.4** (Feature boundedness and Lipschitzness). The Value Critic feature map is bounded: $\|\psi(s)\|_{\mathcal{H}_\psi}^2 \leq M_k$. Whenever perturbation bounds are invoked, there exist finite constants $L_k, L_\psi$ such that $k$ and $\psi$ are Lipschitz in their arguments for Value Critic.

Under the minimum separation rule, the off-diagonal entries of the RBF Gram matrix decay exponentially, yielding a strictly diagonally dominant and PD matrix as follows.

[2]The assumptions here are standard in kernel-based RL and continuous-control systems. Specifically, bounded and Lipschitz-continuous kernels are naturally satisfied by RBF kernels, while geometric mixing and Lipschitz policy conditions hold in many smooth continuous-control environments with Gaussian policies.

**Assumption 4.5** (Gram invertibility). For the dictionaries $\mathcal{D}_V, \mathcal{D}_A$ of size $q_V, q_A$, the Gram matrix $\mathbf{K}_{V,V}$ and $\mathbf{K}_{A,A}$ are invertible. For RBF kernel with length-scale $l$, there are minimum separations $C_k, C_K > 0$ so that $\rho_V := \exp(-\frac{C_k^2}{2l^2})$, $\rho_A := \exp(-\frac{C_K^2}{2l^2})$, $\lambda_{\min}(\mathbf{K}_{V,V}) \geq 1 - (q_V - 1)\rho_V$, and $\lambda_{\min}(\mathbf{K}_{A,A}) \geq 1 - (q_A - 1)\rho_A$. Hence, $\|\mathbf{K}_{V,V}^{-1}\| \leq [1 - (q_V - 1)\rho_V]^{-1}$ and $\|\mathbf{K}_{A,A}^{-1}\| \leq [1 - (q_A - 1)\rho_A]^{-1}$.

When the environment transition dynamics are continuous and satisfy a mild drift condition, the resulting on-policy Markov chain is irreducible and uniformly ergodic.

**Assumption 4.6** (On-policy geometric mixing). The on-policy process is geometric mixing with effective sample size $n_{\mathrm{eff}} \asymp n/\tau_{\mathrm{mix}}$, where $n$ is the total samples and $\tau_{\mathrm{mix}}$ is the mixing time. There exists a constant $C_b \geq 0$ s.t. $g(h_t) = \nabla \log \pi_h(\mathbf{a}|\mathbf{s})$ obeys $\mathbb{E}\|g(h_t) - \hat{g}(h_t)\|_{\mathcal{H}_K}^2 \leq C_b/n_{\mathrm{eff}}$.

Under the requirement to characterize the stability of the policy under perturbations and the induced variations in the stationary distribution, we have the following assumption.

**Assumption 4.7.** Let $\mathcal{H}_k$ be the RKHS for Value Critic and $\mathcal{H}_K$ be the RKHS for Actor. The stationary state-action visitation distribution $\nu_{\pi_h}$ is $C_\nu$-Lipschitz continuous w.r.t. $h$ in the total variation norm $\|\nu_{\pi_h} - \nu_{\pi_{h'}}\|_{\mathrm{TV}} \leq C_\nu \|h - h'\|_{\mathcal{H}_K}, \forall h, h' \in \mathcal{H}_K$. The mapping $h \mapsto \pi_h(\cdot | \mathbf{s})$ is $C_\pi$-Lipschitz in $\mathcal{H}_K$; the underlying Markov chain induced by $\pi_h$ is uniformly ergodic. Assumption 4.5 holds with $C_\nu = O\Big(C_\pi\big(1 + \log \frac{1}{1-\rho_A}\big)\Big)$ for $\rho_A < 1$, analogously for $\rho_V < 1$.

Next, we analyze the convergence under perturbations in three steps. First, we examine the impact of perturbations on the value function. Then, we investigate the convergence of RSA2C without perturbations. Finally, we derive the convergence result under perturbations, establishing the non-asymptotic convergence and sample complexity for RSA2C.

**Step 1: Robustness with perturbation**  We first analyze the error caused by perturbations as a baseline for evaluating the impact of perturbations, with detailed proof in Appendix C.4.

**Theorem 4.8** (Performance gap under adversarial state perturbations). *Let $k$ be a product kernel with bounded components $|k^{(i)}(\mathbf{s}, \mathbf{s})| \leq M, \forall i \in \mathcal{X}$, and assume $\|w_V\|_{\mathcal{H}_k}^2 \leq M_V, M_S := \sup_{\mathbf{s}} \|\mathbf{s}\|_2$, and $M_\phi := \sup \|\phi\|$. Also, let $\varepsilon := \sup_{\widetilde{\mathbf{s}} \in B(\mathbf{s})} \|\mathbf{s} - \widetilde{\mathbf{s}}\|_2$, $\delta_\psi := \sup_{\mathcal{C} \subseteq \mathcal{X}} \|\psi(\mathbf{s}^{\mathcal{C}}) - \psi(\widetilde{\mathbf{s}}^{\mathcal{C}})\|_{\mathcal{H}_k}^2$, and $\delta_{\widetilde{V}}$ be the value-representation perturbation residual in Lemma C.3. Under Assumptions 4.2 and 4.4, there exists a*

*finite constant $C_0 > 0$ such that:*

$$\mathbb{E}\left[|J(\pi) - J(\tilde{\pi})|\right] \leq C_0\Big(2\left(\varepsilon M_S + \varepsilon^2 M_\phi\right) \tag{12a}$$
$$+ M_S^2 d\sqrt{\delta_{\tilde{V}}^2 M^d + M_V \delta_\psi M^d}\Big),$$

$$\mathbb{E}\left[J(\pi) - J(\tilde{\pi})\right] \leq C_0\Big(2\left(\varepsilon M_S + \varepsilon^2 M_\phi\right) \tag{12b}$$
$$+ M_S^2 d\sqrt{\delta_{\tilde{V}}^2 M^d + 2M_V M_\Gamma \delta_\psi M^d}\Big),$$

*where $M_\Gamma := \sup_{\mathcal{C} \subseteq \mathcal{X}} \|\mu_{\mathbf{s}\bar{c}|\mathbf{s}^c}\|^2_{\Gamma_{\mathbf{s}^c}}$. If $k$ is the RBF kernel with length-scale $l$ and Assumption 4.5 holds, then $\delta_\psi = 2 - 2\exp(-\varepsilon^2/(2l^2))$ and*

$$\mathbb{E}\left[|J(\pi) - J(\tilde{\pi})|\right] \leq C_0\Big(2\left(\varepsilon M_S + \varepsilon^2 M_\phi\right) \tag{13a}$$
$$+ M_S^2 d\sqrt{\delta_{\tilde{V}}^2 M^d + M_V \delta_\psi M^d}\Big),$$

$$\mathbb{E}\left[|J(\pi) - J(\tilde{\pi})|\right] \leq C_0\Big(2\left(\varepsilon M_S + \varepsilon^2 M_\phi\right) \tag{13b}$$
$$+ M_S^2 d\sqrt{\delta_{\tilde{V}}^2 M^d + 2M_V M_\Gamma \delta_\psi M^d}\Big).$$

Notably, $\delta_{\tilde{\psi}}$ and $\delta_{\tilde{V}}$ correspond with the SHAP loss and value function loss, respectively. Once the perturbation tends to zero, the difference $\mathbb{E}\left[J(\pi) - J(\tilde{\pi})\right]$ tends to zero.

**Step 2: Convergence without perturbation** Next, we investigate the convergence of RSA2C without perturbations and derive non-asymptotic bounds that quantify its progression toward the final policy. This result is formally stated in Theorem 4.9, with its proof in Appendix C.5.

**Theorem 4.9** (Non-asymptotic convergence without perturbations). *Let the stepsizes be polynomial $\alpha_t^h = \alpha_0(t+1)^{-\sigma}$, $\beta_t = \beta_0(t+1)^{-\nu}$, $\sigma \in (0,1)$, $\nu \in (0,1)$, and choose the canonical coupling $\nu = \frac{2}{3}\sigma$. Suppose in addition that the implementation/noise schedules satisfy $n_{\mathrm{eff},t}^{-1} = \Theta((t+1)^{-2\sigma})$. Define a random index $\tilde{t} \in \{0, 1, \ldots, T-1\}$ with $\Pr(\tilde{t} = t) = \alpha_t^h / \sum_{k=0}^{T-1} \alpha_k^h$. Under Assumptions 4.3-4.7, there exists a constant $C > 0$ such that*

$$(1-\gamma)\mathbb{E}\left[J(\pi^\star) - J(\pi_{h_t})\right] \leq \begin{cases} \zeta_a + \mathcal{O}\left(T^{-(1-\sigma)}\right), & \sigma > \frac{3}{4}, \\ \zeta_a + \mathcal{O}\left(\frac{\log^2 T}{T^{1/4}}\right), & \sigma = \frac{3}{4}, \\ \zeta_a + \mathcal{O}\left(\frac{\log T}{T^{1-\frac{2}{3}\sigma}}\right), & 0 < \sigma < \frac{3}{4}, \end{cases}$$

*where $\zeta_a$ is the (compatible) approximation error of $A^\pi$. When $\sigma = \frac{3}{4}$ and $\nu = \frac{1}{2}$, $(1-\gamma)\mathbb{E}\left[J(\pi^\star) - J(\pi_{h_t})\right] \leq \zeta_a + \mathcal{O}\left(\log^2 T / T^{1/4}\right)$, which implies $\mathcal{O}\left((1-\gamma)^{-5}\epsilon^{-4}\log^2\frac{1}{\epsilon}\right)$ sample complexity for $\mathbb{E}[J(\pi^\star) - J(\pi_{h_t})] \leq \epsilon + \mathcal{O}(\zeta_a)$.*

**Step 3: Convergence with perturbation** Employing the Cauchy inequality, we derive a bound on the perturbation-induced error, ensuring that the learning process remains stable. Combining Theorems 4.8 and 4.9, we establish the convergence of RSA2C summarized in Theorem 4.10.

**Theorem 4.10** (Non-asymptotic convergence of RSA2C). *Let state perturbation be $\tilde{s} \in \mathcal{B}(s) = \{\tilde{s} : \|\tilde{s} - s\|_2 \leq \varepsilon\}$ with stepsizes $\alpha_t^h = \alpha_0(t+1)^{-3/4}$, $\beta_t = \beta_0(t+1)^{-1/2}$. Let $\tilde{t}$ be drawn with $\Pr(\tilde{t} = t) = \alpha_t^h / \sum_{k=0}^{T-1} \alpha_k^h$. Under Assumptions 4.3-4.7, for constant $C_1 > 0$,*

$$\mathbb{E}\left[J(\pi^\star) - J\left(\tilde{\pi}_{h_t}\right)\right] \leq \zeta_a/(1-\gamma) + C_1 \mathcal{B}_k(\varepsilon)$$
$$+ \mathcal{O}\left(\log^2 T/(T^{1/4}(1-\gamma))\right),$$

*where $\mathcal{B}_k(\varepsilon)$ is the perturbation term from Theorem 4.8.*

## 5. Simulation Results

In this section, we present simulations on three continuous-control environments, with details provided in Appendix A.1. The full set of hyperparameters is shown in Appendix A.2. Results for Pendulum-v1 are shown below; the other environments BipedalWalker-v3 and Ant-v5 are reported in Appendix A.3 and A.4. Besides, the additional results are shown in Appendix A.5, including high-noise robustness tests, computational cost, and memory footprint analyses based on RKHS dictionary size. All curves and tables report results averaged over 10 random seeds.

We evaluate the efficiency, stability, and interpretability of RSA2C. For *efficiency*, we conduct ablations and report the overhead as FLOPs and runtime-to-solution. For *stability*, we evaluate robustness to state perturbations by injecting zero-mean noise with varying covariance into the states. For *interpretability*, we track the RKHS-SHAP computed from the *Value Critic* by beeswarm and heatmap plots to show how attribution affects learning stability and efficiency.

To verify the theoretical results, we additionally design a discounted linear-quadratic regulator (LQR) benchmark based on a linearized inverted pendulum, for which the optimal value function admits a closed-form solution via the discrete-time algebraic Riccati equation. In this LQR setting, we can explicitly compute the cost gap $J(\pi^\star) - J(\pi)$ along training, and thus empirically examine the convergence behaviour of our convergence rate, as presented in Appendix A.6.

**Evaluation on Efficiency.** We conduct the ablation study including two variants (RSA2C-KME and RSA2C-CME), RSA2C with uniform SHAP, RSA2C without RKHS-SHAP (Advanced AC), and traditional AC under RKHS (Lever & Stafford, 2015) (RKHS-AC). As shown in Figure 2, both RSA2C-CME and RSA2C-KME converge to substantially higher returns than these ablations. This indicates that the performance gain comes from the learned RKHS-SHAP signal rather than simply from using a kernelized actor-critic architecture. The learned attributions help identify the state dimensions that are most relevant to value improvement. Between the two RSA2C variants, KME reaches the solution threshold earlier, while CME gives a slightly

*Table 1.* Time-to-solution analysis across environments. Convergence epochs are estimated from learning curves.

| Environment | Algorithm | Epochs | Runtime (ms) | Time-to-Solution (s) |
|---|---|---|---|---|
| Pendulum-v1 | RSA2C-KME (CPU) | 864 | 667.7 | 576.9 |
| | RSA2C-CME (CPU) | 1083 | 685.6 | 742.5 |
| | SAC (GPU) | 29 | 7346.2 | **213.0** |
| | PPO (GPU) | 1280 | 2713.3 | 3473.0 |
| BipedalWalker-v3 | RSA2C-KME (CPU) | 1200 | 1354.7 | **1625.7** |
| | RSA2C-CME (CPU) | 1200 | 1448.3 | 1737.9 |
| | SAC (GPU) | 450 | 23982.8 | 10792.2 |
| | PPO (GPU) | 500 | 8119.8 | 4059.9 |
| Ant-v5 | RSA2C-KME (CPU) | 500 | 1358.7 | **679.4** |
| | RSA2C-CME (CPU) | 500 | 1700.4 | 850.2 |
| | SAC (GPU) | 1400 | 9623.5 | 13472.9 |
| | PPO (GPU) | – | 1183.0 | – |

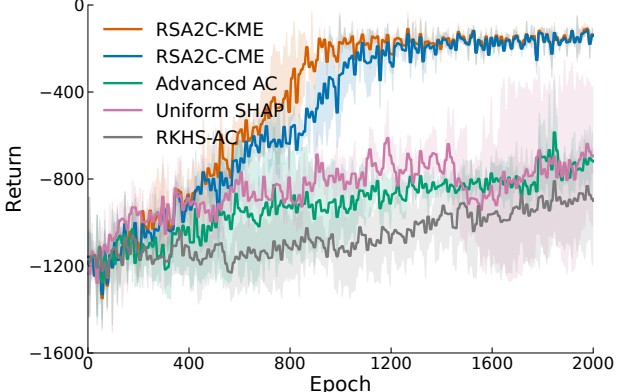

*Figure 2.* Ablation study.

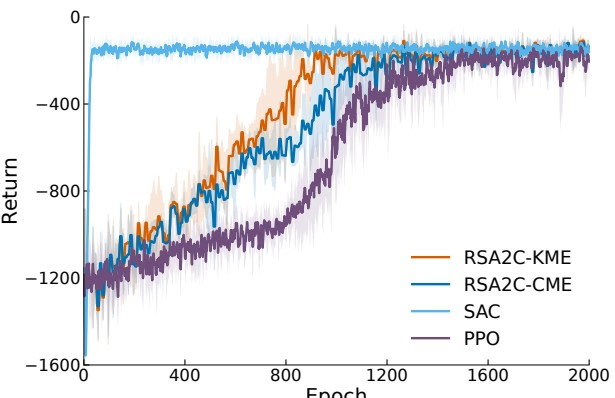

*Figure 3.* Comparison study.

*Table 2.* Stability under different scales of noise variance on Pendulum-v1

| | 0 | 0.001 | 0.005 | 0.01 |
|---|---|---|---|---|
| RSA2C-KME | $\mathbf{-139.83 \pm 26.00}$ | $\mathbf{-147.49 \pm 32.40}$ | $\mathbf{-150.30 \pm 40.32}$ | $\mathbf{-153.41 \pm 40.01}$ |
| RSA2C-CME | $-141.86 \pm 26.77$ | $-152.85 \pm 38.89$ | $-154.38 \pm \mathbf{39.51}$ | $-154.55 \pm 40.18$ |
| Advanced AC | $-719.00 \pm 69.16$ | $-748.46 \pm 151.98$ | $-766.73 \pm 111.20$ | $-784.69 \pm 159.68$ |
| Uniform SHAP | $-693.08 \pm 334.82$ | $-711.10 \pm 366.88$ | $-714.45 \pm 408.83$ | $-762.02 \pm 368.08$ |
| RKHS-AC | $-899.85 \pm 285.37$ | $-906.67 \pm 168.06$ | $-908.35 \pm 289.3$ | $-952.13 \pm 265.74$ |
| SAC | $-160.50 \pm 41.19$ | $-154.80 \pm 44.61$ | $-158.25 \pm 36.10$ | $-155.26 \pm 44.70$ |
| PPO | $-204.83 \pm 85.31$ | $-195.06 \pm 71.38$ | $-176.08 \pm 70.94$ | $-271.61 \pm 198.36$ |

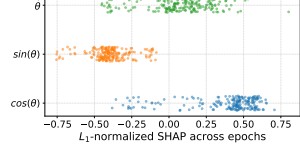

(a) Beeswarm under KME

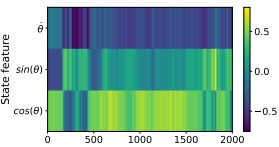

(b) Heatmap under KME

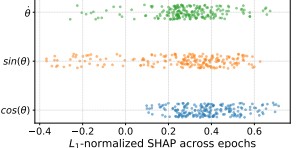

(c) Beeswarm under CME

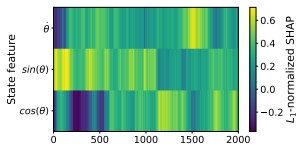

(d) Heatmap under CME

*Figure 4.* Visualization on interpretability of RSA2C on Pendulum-v1.

smoother learning curve because it accounts for conditional feature dependence. Figure 3 further compares RSA2C with standard deep RL algorithms, including soft Actor-Critic (SAC) (Haarnoja et al., 2018) and proximal policy optimization (PPO) (Schulman et al., 2017). Furthermore, Table 1 reports the FLOPs and runtime-to-solution measured on an

Intel(R) Xeon(R) Gold 6248 processor. SAC reaches the solution threshold fastest on Pendulum-v1 because the task is low-dimensional and SAC benefits from highly optimized GPU training. However, RSA2C achieves comparable final returns with a non-neural, CPU-friendly representation and substantially lower per-epoch runtime. PPO is slower and

more variable in this setting.

**Evaluation on Stability.** We evaluate the performance of RSA2C-KME and RSA2C-CME under noisy state perturbations with zero mean and varying variance in Table 2. The two RSA2C variants are the most stable among the RKHS-based methods. Their mean returns change only mildly as the perturbation level increases: RSA2C-KME decreases from $-139.83$ to $-153.41$, and RSA2C-CME decreases from $-141.86$ to $-154.55$. Their standard deviations also remain small, staying around 26-40 across all tested noise levels. In contrast, removing learned RKHS-SHAP causes a large degradation. Advanced AC remains around $-719$ to $-785$, Uniform SHAP has a very large variance, and RKHS-AC stays below $-899$ even without perturbation. These results show that learned feature attributions are important for stabilizing the network update, while fixed weights or no SHAP cannot provide the same robustness.

Compared with deep RL baselines, RSA2C maintains competitive final returns while preserving an explicit attribution mechanism. SAC achieves similar returns across the tested noise levels, but it does not provide the dimension-level training signal used by RSA2C. PPO is more sensitive to larger perturbations, with its mean return dropping to $-271.61$ and its standard deviation increasing to 198.36 at noise level 0.01. Overall, the results indicate that RKHS-SHAP improves both robustness and interpretability.

**Evaluation on Interpretability.** We analyze the interpretability of RSA2C-KME/CME using beeswarm and heatmap visualizations in Figure 4. The attribution patterns are consistent with the underlying physical dynamics of the Pendulum task, indicating that RSA2C captures meaningful feature contributions rather than arbitrary correlations. Since RSA2C-KME does not explicitly account for state correlations, its decisions are often dominated by a single feature dimension, whereas RSA2C-CME models joint feature dependencies and therefore produces more balanced attributions across state dimensions. In the early training stage, the importance ranking under RSA2C-KME is $\cos(\theta) > \dot{\theta} > \sin(\theta)$. Here, $\cos(\theta)$ and $\dot{\theta}$ jointly determine whether the pendulum can accumulate sufficient momentum to swing upward, while $\sin(\theta)$ primarily determines the swing direction. Once effective swinging begins, maintaining directional consistency becomes increasingly important, leading to a higher attribution weight for $\sin(\theta)$. Upon convergence, the ranking shifts to $\cos(\theta) > \sin(\theta) > \dot{\theta}$, reflecting that angle stabilization becomes dominant while angular velocity plays a secondary role. By contrast, RSA2C-CME emphasizes $\sin(\theta)$ at the early stage due to its conditional-distribution modeling, enabling the policy to identify the correct swing-up direction more efficiently. As training progresses, the attributions become more evenly distributed

across the three dimensions, with $\sin(\theta)$ and $\cos(\theta)$ jointly dominant and $\dot{\theta}$ serving a supportive role. This smoother and more balanced attribution allocation leads to more stable policy updates and improved robustness during training.

# 6. Conclusion

We introduce *RSA2C*, an attribution-aware two-timescale AC algorithm that turns state feature importance into a Mahalanobis distance on an OVK-based policy. The proposed algorithm is achieved by enhancing sample-efficient learning with interpretability. Specifically, we derive adaptive signals from RKHS-SHAP (via KME for on-manifold expectations and CME for off-manifold expectations) by instantiating Actor, Value Critic, and Advantage Critic in RKHSs. Our analysis provides a global, non-asymptotic convergence guarantee under state perturbations by decomposing the learning gap into an attribution-induced perturbation term and a refined two-timescale convergence term. Empirically, across three continuous-control environments, RSA2C remains efficient and stable under injected state perturbations. Attribution visualizations reveal environment-relevant state features, showing interpretability. Extending RSA2C to high-dimensional or pixel-based observations via scalable kernel approximations (e.g., RFFs) and combining kernel-based attributions with deep RL to improve stability and expressiveness are interesting research directions.

**Limitations** The scalability of RKHS-based representations in high-dimensional continuous-control tasks limits RSA2C. As the state dimension increases, kernel similarity estimation and sparse dictionary maintenance become increasingly difficult, potentially reducing representation efficiency relative to deep RL, as shown in the results for Ant-v5. In addition, the memory footprint of RKHS dictionaries will increase with task complexity and long training horizons, which also limits the extension to complex tasks.

## Acknowledgements

This work was supported in part by the Zhejiang Provincial Natural Science Foundation of China under Grant LR23F010006, in part by the National Natural Science Foundation Program of China (NSFC) under Grant 62531019, and in part by the Pioneer and Leading Goose R&D Program of Zhejiang under Grants 2026LDC01013(JT) and 2025C01039.

## Impact Statement

This paper presents work whose goal is to advance the field of Explainable Reinforcement Learning. There are many potential societal consequences of our work, none of which we feel must be specifically highlighted here.

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

# A. Additional Experiments

We evaluate on three continuous-control environments from Gymnasium, including Pendulum-v1 and BipedalWalker-v3.

## A.1. Details for Environments

We use the default observation and action spaces and keep the environment settings consistent across methods. A concise summary of each environment is given in Tables 3-5.

*Table 3.* Pendulum-v1 summary

| | |
|---|---|
| Observation | $(\cos\theta,\ \sin\theta,\ \dot{\theta}) \in [-1,1] \times [-1,1] \times [-8,8]$ |
| Action | Torque $\tau \in [-2,2]$ (1D) |
| Reward | $-(\theta^2 + 0.1\dot{\theta}^2 + 0.001\tau^2)$ |
| Horizon | 200 steps |

*Table 4.* BipedalWalker-v3 summary

| | |
|---|---|
| Observation | 24D proprioception + 10 lidar ranges |
| Action | Motor speeds in $[-1,1]^4$ |
| Reward | Forward progress (small torque cost); $-100$ on fall |
| Horizon | Up to $\sim$1600 steps or on failure |

*Table 5.* Ant-v5 summary

| | |
|---|---|
| Observation | 105D state (proprioception, contacts, joint angles/velocities) |
| Action | 8D joint torques in $[-1,1]^8$ |
| Reward | Forward progress + alive bonus $-$ control and contact costs |
| Horizon | 1000 steps or termination on fall |

## A.2. Hyper-parameters

The hyper-parameters configured in experiments are summarized in Table 6.

## A.3. Results of BipedalWalker-v3

**Evaluation on Efficiency.** In the continuous control environment with multi-dimensional actions (BipedalWalker-v3), we compare two variants of RSA2C (RSA2C-CME and RSA2C-KME), our Advanced AC framework without SHAP, and the classical RKHS-AC. As shown in Figure 5, Advanced AC exhibits faster improvement in the early stage, but suffers from larger variance and slightly lower returns in the later phase compared to RSA2C. In contrast, both RSA2C variants achieve more stable trajectories and higher final returns, indicating that the benefit of SHAP lies in stabilizing training and enhancing long-term convergence rather than accelerating the initial learning. Among them, CME yields smoother curves than KME.

Figure 6 compares RSA2C with the deep RL algorithms SAC and PPO. SAC reaches high returns quickly but fluctuates substantially throughout training, while PPO shows slower and less stable progress. The RSA2C variants eventually catch up and achieve comparable or superior asymptotic performance despite their lightweight non-neural architecture. This confirms that RSA2C can remain competitive with deep RL even in more challenging multi-dimensional action spaces.

Regarding computational complexity, illustrated Table 1, RSA2C requires approximately 1.427 GFLOPs for CME setting and 0.381 GFLOPs for KME setting, which is higher than Advanced AC with 0.346 GFLOPs and RKHS-AC with 0.264 GFLOPs, but all methods remain within the same millisecond runtime scale. We can found similar trend on runtime. Specifically, CME increases theoretical FLOPs, while the empirical runtime grows by approximately 6.9%. This discrepancy arises because FLOPs measure the theoretical number of floating-point operations, ignoring parallelized matrix computations in practice. Importantly, both RSA2C variants remain orders of magnitude faster and lighter than SAC, which rely on deep neural networks and require far more computation even when executed on GPU.

*Table 6.* Hyper-parameters on different environments

| Description | Pendulum-v1 | BipedalWalker-v3 | Ant-v5 |
|---|---|---|---|
| Epoch | 2000 | 1500 | 2000 |
| Discount factor $\gamma$ | 0.99 | 0.99 | 0.99 |
| Initial $\Sigma$ | 0.35$\mathbf{I}$ | 0.9$\mathbf{I}$ | $\mathbf{I}$ |
| Final $\Sigma$ | 0.25$\mathbf{I}$ | 0.4$\mathbf{I}$ | 0.3$\mathbf{I}$ |
| Variance of RBF kernel | 0.8 | 3.0 | 10.0 |
| Max size of dictionary $\mathcal{D}_V$ | 384 | 1024 | 4096 |
| Max size of dictionary $\mathcal{D}_A$ | 384 | 1024 | 4096 |
| Episodes of evaluation | 5 | 5 | 5 |
| Learning rate of Value Critic $\nu$ | 0.01 | 0.005 | 0.05 |
| Learning rate of Actor $\sigma$ | 1.0 | 0.08 | 0.25 |

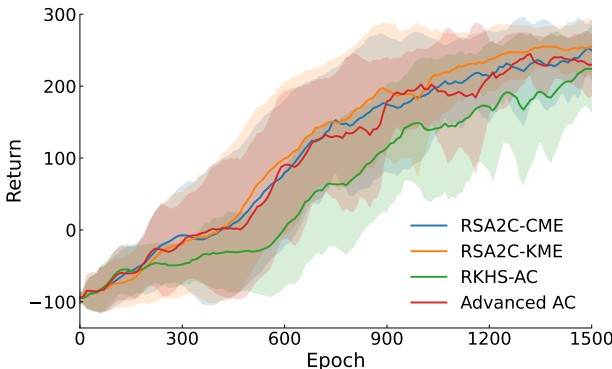

*Figure 5.* Ablation study on BipedalWalker-v3.

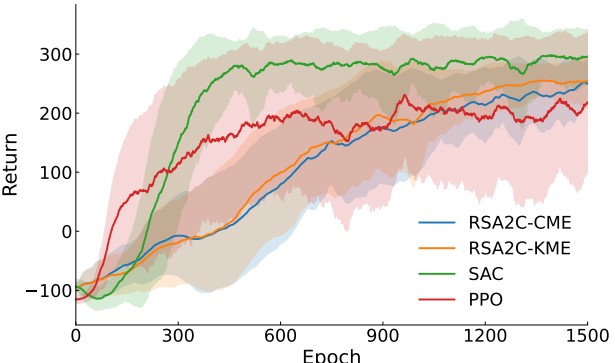

*Figure 6.* Performance on BipedalWalker-v3.

**Evaluation on Stability.** We evaluate RSA2C-KME and RSA2C-CME on BipedalWalker-v3 under zero-mean state noise with varying variance, illustrated in Table 7. Across nonzero noise levels, RSA2C-CME shows markedly lower variability, with standard deviations from 24.75 to 43.06, whereas RSA2C-KME ranges from 46.74 to 56.03. For the mean return, the two variants are similar at light noise 0 and 0.001, and CME outperforms KME at higher noise 0.005 and 0.01, yielding 264.59 versus 242.89 and 254.85 versus 242.05, respectively. Both variants exceed the Advanced AC baseline of 229.89 without perturbation. This improvement in stability stems from SHAP-guided state reweighting, which resists noisy gradients and stabilizes policy updates. In addition, we attribute CME's superior stability to its explicit modeling of feature dependencies, which enables adaptive reweighting under noise and mitigates error amplification from single-dimension dominance, whereas KME ignores correlations and is more vulnerable when noise strikes dominant features.

*Table 7.* Stability under different scales of noise variance of `BipedalWalker-v3`

| | 0 | 0.001 | 0.005 | 0.01 |
|---|---|---|---|---|
| RSA2C-KME | $255.09 \pm 37.19$ | $249.72 \pm 56.03$ | $242.89 \pm 54.94$ | $242.05 \pm 46.74$ |
| RSA2C-CME | $249.72 \pm 41.47$ | $246.65 \pm 43.06$ | $264.59 \pm 24.75$ | $254.85 \pm 27.31$ |
| SAC | $277.59 \pm 93.18$ | $292.42 \pm 57.25$ | $299.98 \pm 37.43$ | $234.80 \pm 121.49$ |
| PPO | $241.41 \pm 135.30$ | $174.59 \pm 129.96$ | $259.03 \pm 96.26$ | $136.56 \pm 169.12$ |

**Evaluation on Interpretability.** In the BipedalWalker-v3 environment, we compare the feature importance of KME and CME using beeswarm and heatmap visualizations, as shown in Fig. 7. The SHAP distribution of KME is generally small and concentrated, with most values near zero except for a few dimensions such as vel x, vel y, and joint speeds, exhibiting a sparse and nearly fixed pattern. Its heatmap remains largely yellow in the later training stages, indicating weak temporal dynamics. In contrast, CME demonstrates a wider SHAP range with both positive and negative contributions, consistently capturing the effects of hip and knee joints and their velocities, body posture (hull angle and angle speed), and leg contacts across multiple dimensions. Moreover, several key features are gradually reinforced in the second half of training, reflecting stronger temporal adaptability. This richer structure is also consistent with the noise-perturbation results, where SAC and PPO exhibit substantially larger fluctuations across all noise scales, whereas RSA2C maintains stable performance. Overall,

compared with KME, CME provides richer, direction-sensitive, and training-evolving explanatory signals, with a greater emphasis on the coupling mechanisms among state features.

## A.4. Results of Ant-v5

**Evaluation on Efficiency.** In the high-dimensional control task Ant-v5, the behavior of RSA2C differs markedly from its performance on lower-dimensional environments. As shown in Figure 8, all RKHS-based variants converge to a similar performance plateau around 1000. This indicates that RSA2C remains stable but becomes trapped in a local optimum when the state dimensionality is large and the underlying dynamics are highly nonlinear. Figure 9 shows that deep RL algorithms are able to discover significantly better solutions in high-dimensional state space. SAC continues to improve and surpasses 4000 return, clearly outperforming all RKHS-based methods. The limited gap among variants further suggests that, under such complexity, the linear RKHS function approximation combined with RBF kernels cannot fully capture the richer structure required for escaping suboptimal regions, thus diminishing the relative advantage of SHAP-guided feature weighting.

Table 1 summarizes the computational cost. RSA2C-KME (2.154 GFLOPs) and RSA2C-CME (3.853 GFLOPs) require more computation than RKHS-AC and Advanced AC, but they remain substantially lighter than deep RL baselines. SAC demands 29.28 GFLOPs and over 9600 ms per iteration on GPU, whereas PPO requires 2.27 GFLOPs but still incurs higher runtime than RSA2C-KME. Although CME is more costly than KME, both RSA2C variants maintain a favorable efficiency-stability trade-off. Importantly, the computational results reinforce the earlier observation that RSA2C is stable and lightweight, but its linear-RKHS formulation limits representational capacity in high-dimensional dynamics.

*Table 8.* Stability under different scales of noise variance of `Ant-v5`

|  | 0 | 0.001 | 0.005 | 0.01 |
|---|---|---|---|---|
| RSA2C-KME | $960.36 \pm 22.09$ | $957.37 \pm 13.85$ | $954.13 \pm 23.67$ | $958.54 \pm 21.90$ |
| RSA2C-CME | $934.15 \pm 44.39$ | $953.84 \pm 17.65$ | $948.86 \pm 26.02$ | $959.65 \pm 21.86$ |
| SAC | $4477.33 \pm 1950.69$ | $3761.98 \pm 2276.18$ | $4067.61 \pm 1410.60$ | $2619.72 \pm 1516.74$ |
| PPO | $17.34 \pm 45.12$ | $155.64 \pm 260.74$ | $32.47 \pm 65.07$ | $-11.15 \pm 32.11$ |

**Evaluation on Stability.** Table 8 reports the robustness of different algorithms under zero-mean Gaussian perturbations with varying noise variance. Both RSA2C-KME and RSA2C-CME remain remarkably stable across all noise levels, with returns consistently around 940-960 and standard deviations below 45, indicating that the RKHS-based linear approximation maintains strong resistance to state corruption even in high-dimensional tasks. In contrast, SAC exhibits extremely large variance at every noise setting, with fluctuations exceeding 1400-2200, reflecting high sensitivity of deep RL. PPO is even more unstable. Although it performs moderately under small noise (variance 0.001), its return collapses toward zero or even negative values under larger perturbations.

**Evaluation on Interpretability.** Figure 10 visualizes the SHAP-based feature attributions of KME and CME using beeswarm and temporal heatmaps. For RSA2C-KME, the SHAP scores are highly sparse and concentrated around zero, with only a few contact-related dimensions exhibiting noticeable contributions. Moreover, the temporal heatmap under RSA2C-KME remains mostly uniform and lacks clear structure, indicating that the learned importance is nearly static throughout training. This suggests that RSA2C-KME fails to capture the intricate dependencies among high-dimensional proprioceptive and contact features, resulting in weak and non-evolving explanatory signals. In contrast, RSA2C-CME yields substantially richer and more diverse attributions. The beeswarm plot reveals distinct positive and negative contributions across many joint, contact, and orientation dimensions. The heatmap further shows clear temporal patterns that key contact forces and ankle/hip interactions activate selectively at different training stages, and several features-such as joint torques, angular velocities, and contact normals-exhibit gradually strengthened importance. These structured and evolving patterns indicate that RSA2C-CME successfully models feature correlations and adapts attribution as the policy improves.

## A.5. Additional Results

The computational efficiency of RSA2C comes from its sparse RKHS representation. Instead of operating over all collected transitions, the Actor and Critics rely on dynamically maintained dictionaries of representative states, so the dominant kernel operations scale with dictionary size rather than the full sample history. We further analyze the memory footprint as a function of dictionary size. The results show that memory grows in a controlled manner because the ALD update keeps the dictionaries bounded, avoiding the prohibitive storage cost of naive kernel methods. This also highlights a useful

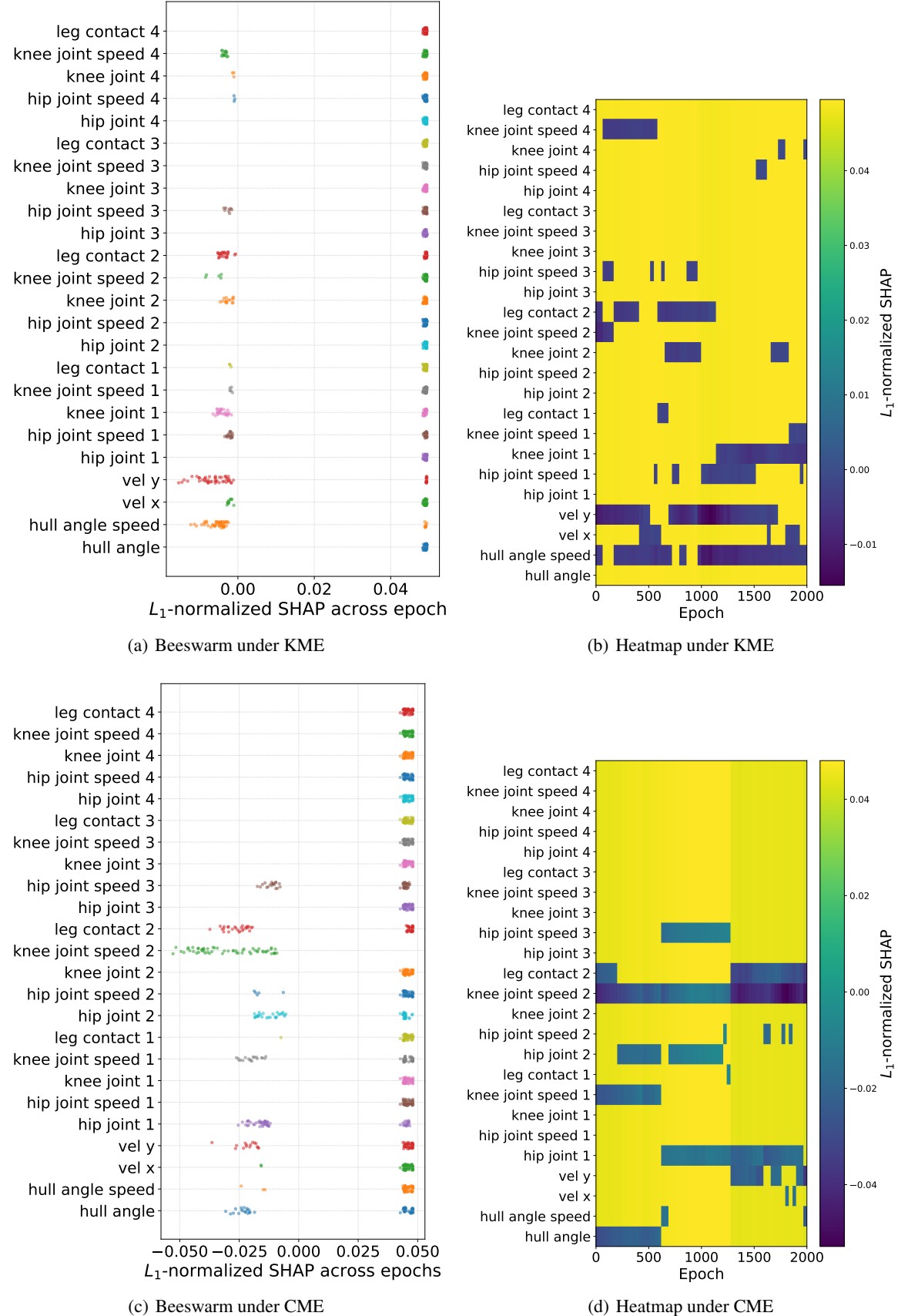

(a) Beeswarm under KME

(b) Heatmap under KME

(c) Beeswarm under CME

(d) Heatmap under CME

*Figure 7.* Visualization on interpretability of RSA2C on BipedalWalker-v3.

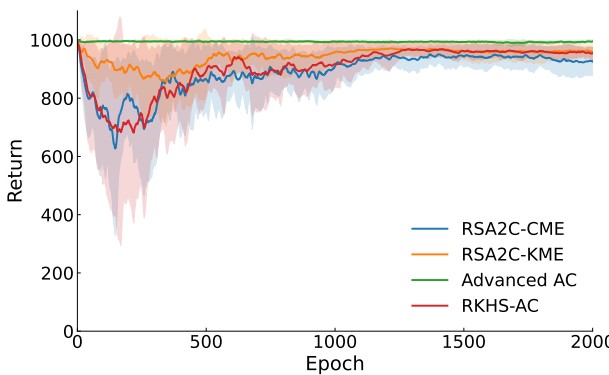

*Figure 8.* Ablation study on Ant-v5.

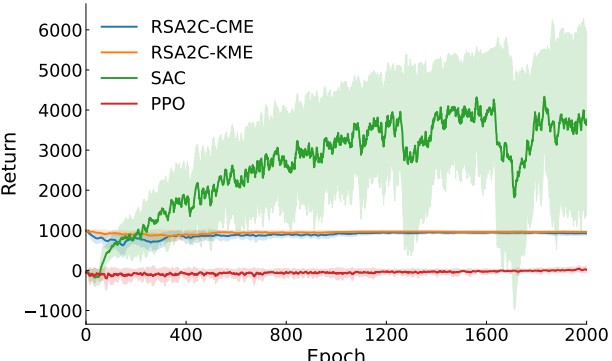

*Figure 9.* Performance on Ant-v5.

*Table 9.* Performance of three tasks on high noise.

|  |  | 0.05 | 0.1 | 0.5 |
|---|---|---|---|---|
| Pendulum-v1 | RSA2C-KME | $-258.37 \pm 225.39$ | $-252.10 \pm 198.12$ | $-807.49 \pm 62.21$ |
|  | RSA2C-CME | $-231.41 \pm 134.29$ | $-274.22 \pm 229.07$ | $-815.64 \pm 53.71$ |
| BipedalWalker-v3 | RSA2C-CME | $246.66 \pm 25.66$ | $232.34 \pm 5.70$ | $-12.33 \pm 29.02$ |
|  | RSA2C-KME | $241.83 \pm 10.07$ | $185.04 \pm 15.86$ | $-23.46 \pm 29.91$ |
| Ant-v5 | RSA2C-CME | $970.91 \pm 6.43$ | $965.67 \pm 12.33$ | $-830.79 \pm 389.33$ |
|  | RSA2C-KME | $956.38 \pm 5.33$ | $967.76 \pm 3.78$ | $-1109.16 \pm 140.58$ |

trade-off: neural networks obtain compactness through parametric compression, whereas RSA2C preserves representative states explicitly through adaptive dictionaries. In high-dimensional environments, stronger kernel locality may require larger dictionaries to cover the state space, but the empirical memory and runtime results indicate that the resulting overhead remains manageable under the proposed sparsification scheme.

## A.6. Simulation on Theoretical non-asymptotic rate

### A.6.1. DETAILS FOR LQR

To numerically examine the convergence guarantees in a controlled setting, we introduce a discounted LQR environment constructed from the linearization of the classic cart-pole (inverted pendulum) around the upright equilibrium. The continuous-time dynamics are linearized and then discretized with step size $\Delta t = 0.02$, yielding a four-dimensional state $\mathbf{s}_t = [x, \dot{x}, \theta, \dot{\theta}]^\top$ and a scalar control input $\mathbf{a}_t$. Specifically, $x_t$ and $\dot{x}_t$ are the cart position and velocity, and $\theta_t$ and $\dot{\theta}_t$ are the pole angle (measured from the upright) and angular velocity, respectively.

The instantaneous quadratic cost is given by

$$r(\mathbf{s}_t, \mathbf{a}_t) = \mathbf{s}_t^\top P_1 \mathbf{s}_t + \mathbf{a}_t^\top P_2 \mathbf{a}_t,$$

with $P_1 = \mathrm{diag}(1, 0.1, 10, 0.1)$ and $P_2 = 0.01$, and the return is defined as the discounted cumulative cost with factor $\gamma = 0.99$. Since this is a linear-quadratic system, the optimal value function takes the form $J^\star(\mathbf{s}) = \mathbf{s}^\top P_3 \mathbf{s}$, where $P_3$ is obtained by solving the discrete-time algebraic Riccati equation associated with the discounted dynamics. This allows us to compute, for specific discounted visitation distribution, the gap $J(\pi) - J(\pi^\star)$ of the learned policy and to track the decay of this gap over training epochs, providing a quantitative convergence testbed for our non-asymptotic theory.

### A.6.2. RESULTS OF THEORETICAL GAP

Figure 11 illustrates the convergence of the optimality gap $J(\pi^\star - J(\tilde{\pi}_{h_t})$ in the LQR setting, where the optimal value function is available in closed form via the Riccati equation. Both RSA2C-CME and RSA2C-KME exhibit a rapid and consistent reduction of the gap, significantly outperforming the Advanced-AC baseline in terms of convergence speed and stability, indicating that the improved training stability is primarily attributed to the SHAP-guided attribution mechanism. In particular, RSA2C-CME achieves the lowest steady-state gap with markedly reduced variance, indicating that its CME-

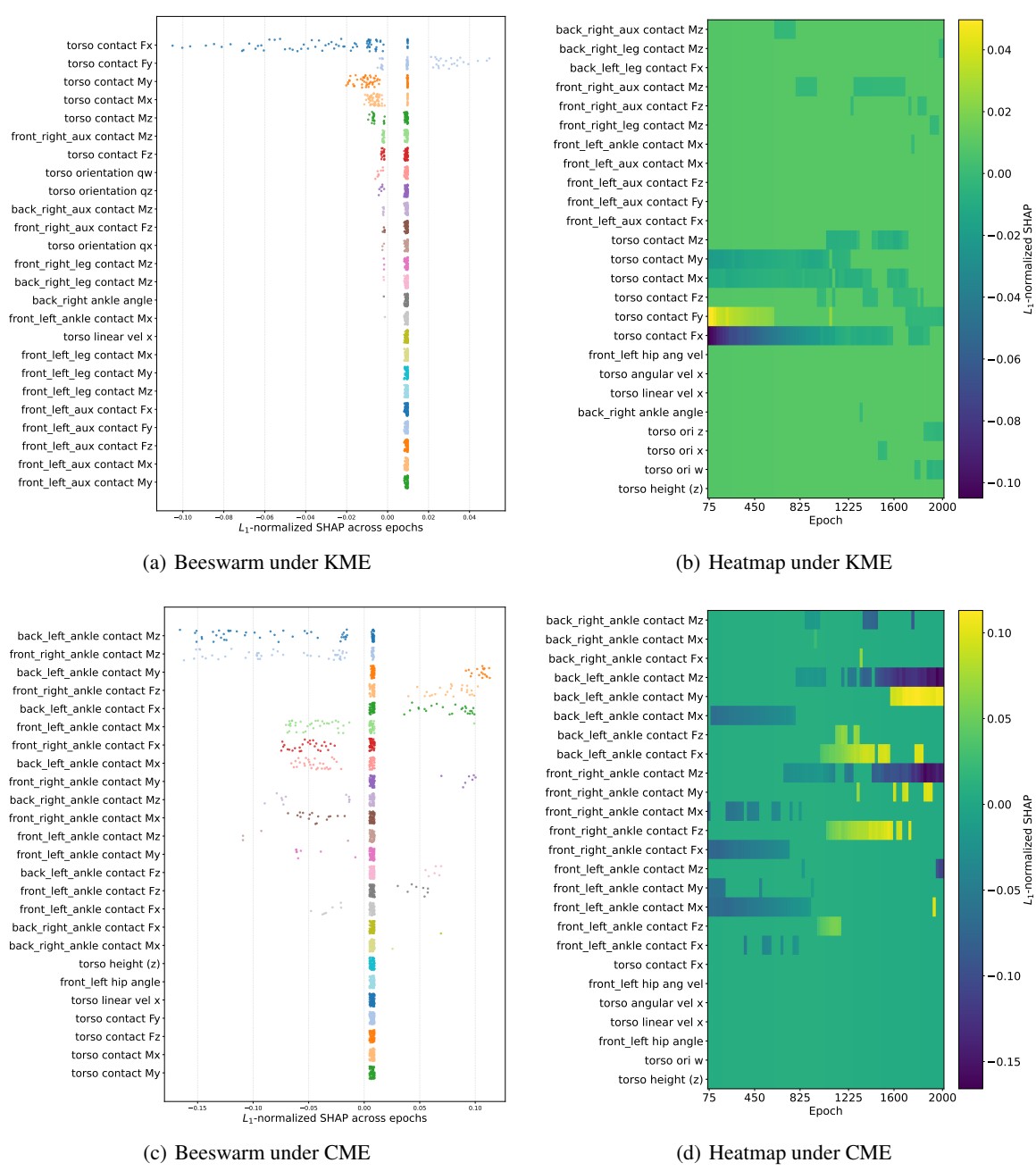

(a) Beeswarm under KME

(b) Heatmap under KME

(c) Beeswarm under CME

(d) Heatmap under CME

*Figure 10.* Visualization on interpretability of RSA2C on Ant-v5.

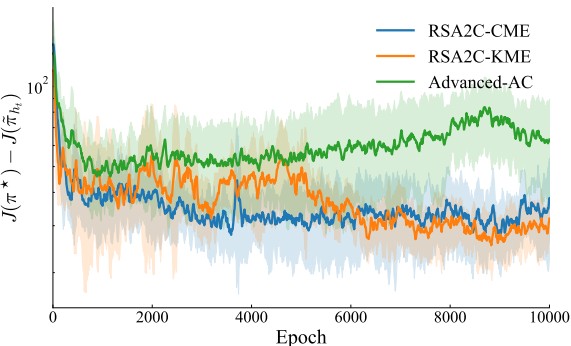

*Figure 11.* onvergence of the result gap $J(\pi) - J(\pi^\star)$ on the LQR environment

*Figure 12.* The performance of computational cost.

based structural attribution leads to more reliable and directionally accurate policy updates. RSA2C-KME also improves substantially over the baseline, demonstrating the benefit of gradient-based feature weighting. Overall, the results confirm that RSA2C provides a more stable and sample-efficient path toward the optimal LQR controller.

## B. Algorithm details

### B.1. Online Sparsification in RSA2C

Unlike parametric models with fixed-size representations, successively updating RSA2C by incrementally adding new components leads to increasingly complex function representations. While this enables the modeling of expressive policies and value or advantage functions, it also introduces substantial computational overhead, particularly during evaluation. To ensure tractability, it is therefore essential to control the complexity of the kernel expansions for both the Actor function $h(\cdot)$ and the Critic features $\psi(\cdot)$, as well as their gradients.

To address this, we introduce an online sparsification scheme that incrementally prunes the kernel representation in a data-efficient manner. This scheme is inspired by the sparse kernel learning framework in (Yang et al., 2022), and is applicable to both vector-valued and scalar-valued kernels.

We first describe the sparsification strategy for the advanced Actor. Given a kernel-based policy representation of the form

$$h(\mathbf{s}) = \frac{1}{N} \sum_j K(\mathbf{s}, \mathbf{s}_j)\mathbf{c}_j \in \mathcal{H}_K,$$

our goal is to obtain a sparse approximation:

$$\hat{h}(\mathbf{s}) = \frac{1}{q_{\mathsf{A}}} \sum_{(\mathbf{s}_j, \mathbf{c}_j) \in \mathcal{D}_{\mathsf{A}}} K(\mathbf{s}, \mathbf{s}_j)\mathbf{c}_j \in \mathcal{H}_K,$$

where only $q_{\mathsf{A}} \ll N$ coefficients are non-zero and $\mathcal{D}_{\mathsf{A}}$ denotes the dictionary of selected kernel centers and their corresponding coefficients.

To enable this, we adopt an online kernel ALD method (Aissing & Monkhorst, 1992), which incrementally constructs a compact and informative dictionary during policy optimization in RKHS.

If the current dictionary $\mathcal{D}_{\mathsf{A}}$ has not yet reached the predefined size limit, a new sample $(\mathbf{s}_\iota, \mathbf{c}_\iota)$ is directly added. The coefficient $\mathbf{c}_\iota$ is computed as:

$$\mathbf{c}_\iota = A^{\pi_h, \Sigma}_{\overline{w}_{\mathsf{A}}}(\mathbf{s}_\iota, \mathbf{a}_\iota)\Sigma^{-1}(\mathbf{a}_\iota - h(\mathbf{s}_\iota)),$$

where $A^{\pi_h, \Sigma}_{\overline{w}_{\mathsf{A}}}(\cdot, \cdot)$ is the advantage-weighted Actor gradient term, and $h(\cdot)$ denotes the current RKHS-based policy.

Once the dictionary reaches its size constraint, for each policy update step (see 4), we must decide whether the new sample provides sufficiently novel information to warrant inclusion. This decision is made by solving the following kernel projection problem:

$$\min_{\{\mathbf{c}_j\} \in \mathcal{D}_{\mathsf{A}}} \left\| \sum_j K(\mathbf{s}_j, \cdot)\mathbf{c}_j - K(\mathbf{s}_\iota, \cdot)\mathbf{c}_\iota \right\|^2,$$

which yields the projection residual $\xi_{\mathsf{A}, \iota}$. If this residual exceeds a predefined threshold $\eta$, the dictionary entry with the largest historical approximation error is replaced by $(\mathbf{s}_\iota, \mathbf{c}_\iota)$. The residual is then recomputed using the remaining $q-1$ dictionary elements, and the stored values are updated accordingly.

If the residual falls below the threshold, we retain the existing dictionary structure but update the coefficients $\{\mathbf{c}_j\}$ using the newly optimized projection, thereby refining the approximation under the current basis.

Once the dictionary $\mathcal{D}_{\mathsf{A}}$ is finalized, the associated Critic weights $w_{\mathsf{V}}$ can be directly computed using the resulting sparse kernel representation.

Finally, the sparsification process for Value Critic mirrors that of the Actor, with a key distinction: Value Critic employs a scalar-valued kernel to approximate the scalar value function $V(\mathbf{s})$, whereas the Actor uses a vector-valued kernel to represent the policy mapping $h(\mathbf{s})$ in the action space. Due to this structural similarity, we omit redundant derivations for Value Critic and refer the reader to the Actor sparsification procedure for details.

## B.2. Advanced Actor update

We now consider how to compute the steepest ascent direction of the return $J(\theta)$ in (4) w.r.t. $h$ when the policy is modelled as in (4) and $h$ is modelled non-parametrically in a (vector-valued) RKHS $\mathcal{H}_K \subseteq \mathcal{A}^\mathcal{S}$. Importantly, the gradient will be an entire function in $\mathcal{H}_K$. Recalling (4), to compute the steepest ascent direction we first need to compute the gradient of $\log \pi_{h,\Sigma}(\mathbf{a}_t \mid \mathbf{s}_t)$ which is then integrated together with the $Q$-function. This will be a functional gradient and the notion of the Fréchet derivative is sufficient for us.

Different from the goal of standard RL as maximizing the cumulative discounted reward $J(h, \Sigma) = \mathbb{E}\left[\sum_{t=1}^\infty \gamma^{t-1} r_t\right]$, we here consider a more general maximum objective, which is

$$J(h, \Sigma) = \mathbb{E}\left[\sum_{t=1}^\infty \gamma^{t-1} r_t\right]. \tag{14}$$

According to policy gradient theorem, there is

$$\nabla_h J(h, \Sigma) = \mathbb{E}_{\mathbf{s} \sim D^{\pi_{h,\Sigma}}, \mathbf{a} \sim \pi_{h,\Sigma}(\mathbf{a}|\mathbf{s})}\left[Q_\psi^{\pi_{h,\Sigma}}(\mathbf{s}, \mathbf{a}) \nabla_h \log \pi_{h,\Sigma}(\mathbf{a} \mid \mathbf{s})\right]. \tag{15}$$

**The policy gradient w.r.t. $h$**  To derive the policy gradient w.r.t. $h$ for Actor, we begin with the definition of Fréchet derivative.

**Definition B.1.** The Fréchet derivative is the derivative for functions on a Banach space. Let $\mathcal{V}$ and $\mathcal{W}$ be Banach spaces, and $\mathcal{U} \subset \mathcal{V}$ be an open subset of $\mathcal{V}$. A function $f : \mathcal{U} \to \mathcal{W}$ is called Fréchet differentiable at $x \in \mathcal{U}$ if there exists a bounded linear operator $Df|_x : \mathcal{V} \to \mathcal{W}$ such that,

$$\lim_{r \to 0} \frac{\|f(x + r) - f(x) - Df|_x(r)\|_\mathcal{W}}{\|r\|_\mathcal{V}} = 0.$$

According to the kernel-based policy in (4), we derive

$$\log \pi_{h,\Sigma}(\mathbf{a}_t \mid \mathbf{s}_t) = -\log((2\pi)^{\frac{m}{2}}(\det(\Sigma))^{\frac{1}{2}}) - \frac{1}{2}(\mathbf{a}_t - h(\mathbf{s}_t))^\top \Sigma^{-1}(\mathbf{a}_t - h(\mathbf{s}_t)). \tag{16}$$

**Proposition B.2.** *The derivative of the map $f : \mathcal{H}_K \to \mathbb{R}$, $f : h \mapsto \log \pi_{h,\Sigma}(\mathbf{a}_t \mid \mathbf{s}_t)$, at $h$, is the bounded linear map $Df|_h : \mathcal{H}_K \mapsto \mathbb{R}$ defined by*

$$Df|_h : g \mapsto (\mathbf{a} - h(\mathbf{s}))\Sigma^{-1}g(\mathbf{s}) = \langle K(\mathbf{s}, \cdot)\Sigma^{-1}(\mathbf{a} - h(\mathbf{s})), g \rangle_K. \tag{17}$$

*Thus the direction of steepest ascent is the function*

$$\nabla_h \log \pi_{h,\Sigma}(\mathbf{a} \mid \mathbf{s}) = K(\mathbf{s}, \cdot)\Sigma^{-1}(\mathbf{a} - h(\mathbf{s})) \in \mathcal{H}_K.$$

Finally, we obtain the gradient of Actor w.r.t. $h$ as follows:

$$\begin{aligned}
\nabla_h J(h, \Sigma) &= \mathbb{E}_{\mathbf{s} \sim D^{\pi_{h,\Sigma}}, \mathbf{a} \sim \pi_{h,\Sigma}(\mathbf{a}|\mathbf{s})}\left[Q_\psi^{\pi_{h,\Sigma}}(\mathbf{s}, \mathbf{a}) \nabla_h \log \pi_{h,\Sigma}(\mathbf{a} \mid \mathbf{s})\right] \\
&= \mathbb{E}_{\mathbf{s} \sim D^{\pi_{h,\Sigma}}, \mathbf{a} \sim \pi_{h,\Sigma}(\mathbf{a}|\mathbf{s})}\left[Q_\psi^{\pi_{h,\Sigma}}(\mathbf{s}, \mathbf{a}) K(\mathbf{s}_t, \cdot)\Sigma^{-1}(\mathbf{a} - h(\mathbf{s}))\right] \\
&\approx \frac{1}{n}\sum_\iota \left[Q_\psi^{\pi_{h,\Sigma}}(\mathbf{s}_\iota, \mathbf{a}_\iota) K(\mathbf{s}_\iota, \cdot)\Sigma^{-1}(\mathbf{a}_\iota - h(\mathbf{s}_\iota))\right]. \\
&\approx \frac{1}{n}\sum_\iota \left[\widehat{A}_\psi^{\pi_{h,\Sigma}}(\mathbf{s}_\iota, \mathbf{a}_\iota) K(\mathbf{s}_\iota, \cdot)\Sigma^{-1}(\mathbf{a}_\iota - h(\mathbf{s}_\iota))\right],
\end{aligned} \tag{18}$$

where the last line exists since value function $V_{w_V}^{\pi_{h,\Sigma}}(\mathbf{s}_t)$ is not directly affected by action $\mathbf{a}$ and

$$\mathbb{E}_{\mathbf{a} \sim \pi_{h,\Sigma}(\mathbf{a}|\mathbf{s})}\left[\nabla \log \pi_{h,\Sigma}(\mathbf{a} \mid \mathbf{s})\right] = \sum_\mathbf{a}\left[\nabla \pi_{h,\Sigma}(\mathbf{a} \mid \mathbf{s})\right] = \nabla\left[\sum_\mathbf{a} \pi_{h,\Sigma}(\mathbf{a} \mid \mathbf{s})\right] = \nabla[1] = 0.$$

Note that the steepest ascent direction is a function in $\mathcal{H}$. And we can estimate $\nabla_h J(h, \Sigma)$ by sampling $n$ state-action pairs $\{\mathbf{s}_\iota, \mathbf{a}_\iota, r_\iota, \mathbf{s}_{\iota+1}\}$ and approximating with the average.

## B.3. Compatible function approximation

In this section, we demonstrate that the Advantage Critic and Value Critic satisfied the compatible function architecture. To begin with, we have

$$\widehat{Q}^{\pi_h,\Sigma}(\mathbf{s},\mathbf{a}) = \widehat{A}^{\pi_h,\Sigma}(\mathbf{s},\mathbf{a}) + \widehat{V}^{\pi_h,\Sigma}(\mathbf{s}) = A_{w_\mathsf{A}}^{\pi_h,\Sigma}(\mathbf{s},\mathbf{a}) + V_{w_\mathsf{V}}^{\pi_h,\Sigma}(\mathbf{s})$$

along with function approximation.

According to Theorem 2 of policy gradient with function approximation in (Sutton et al., 1999), we just need to prove

$$\mathbb{E}_{\mathbf{s}\sim D^{\pi_h,\Sigma},\mathbf{a}\sim\pi_{h,\Sigma}(\mathbf{a}|\mathbf{s})}\left[\left(Q^{\pi_h,\Sigma}(\mathbf{s},\mathbf{a}) - \widehat{Q}^{\pi_h,\Sigma}(\mathbf{s},\mathbf{a})\right)\nabla_{w_\mathsf{A},w_\mathsf{V}}\widehat{Q}^{\pi_h,\Sigma}(\mathbf{s},\mathbf{a})\right] = 0, \tag{19}$$

which shows that the error in $\widehat{Q}^{\pi_h,\Sigma}(\mathbf{s},\mathbf{a})$ is orthogonal to the gradient of the policy parametrization.

Since value function $V_{w_\mathsf{V}}^{\pi_h,\Sigma}(\mathbf{s})$ is not directly affected by action $\mathbf{a}$, and

$$\mathbb{E}_{\mathbf{a}\sim\pi_{h,\Sigma}(\mathbf{a}|\mathbf{s})}\left[\nabla\log\pi_{h,\Sigma}(\mathbf{a}\mid\mathbf{s})\right] = \sum_{\mathbf{a}}\left[\nabla\pi_{h,\Sigma}(\mathbf{a}\mid\mathbf{s})\right] = \nabla\left[\sum_{\mathbf{a}}\pi_{h,\Sigma}(\mathbf{a}\mid\mathbf{s})\right] = \nabla[1] = 0.$$

Therefore, we turn to derive

$$\mathbb{E}_{\mathbf{s}\sim D^{\pi_h,\Sigma},\mathbf{a}\sim\pi_{h,\Sigma}(\mathbf{a}|\mathbf{s})}\left[\left(Q^{\pi_h,\Sigma}(\mathbf{s},\mathbf{a}) - \widehat{Q}^{\pi_h,\Sigma}(\mathbf{s},\mathbf{a})\right)\nabla_{w_\mathsf{A}}A_{w_\mathsf{A}}^{\pi_h,\Sigma}(\mathbf{s},\mathbf{a})\right] = 0. \tag{20}$$

Combining

$$\nabla_{w_\mathsf{A}}A_{w_\mathsf{A}}^{\pi_h,\Sigma}(\mathbf{s},\mathbf{a}) = \nu(\mathbf{s},\mathbf{a}) = K(\mathbf{s},\cdot)\Sigma^{-1}\left(\mathbf{a} - h(\mathbf{s})\right)$$

with (15), we complete the proof.

# C. Proof of theoretical guarantees

## C.1. Constants and Notation

Unless otherwise stated, all constants below are positive and independent of the time index.

- $q_\mathsf{A}, q_\mathsf{V} \in \mathbb{N}$: (sparse) dictionary size of Actor and Value Critic networks, respectively.

- $C_k > 0$: minimum separation among dictionary centers.

- $M_k := \sup_\mathbf{s} k(\mathbf{s},\mathbf{s})$: boundedness constant of a scalar kernel $k$.

- $M_K := \sup_\mathbf{s} \|K(\mathbf{s},\mathbf{s})\|_\mathrm{op}$: boundedness of an operator-valued kernel $K$.

- $C_\mathrm{ev}$: evaluation-operator bound, i.e., $\|f(\mathbf{s})\|_2 \leq C_\mathrm{ev}\|f\|_\mathcal{H}$ for all $f$ in the RKHS.

- $M_\Gamma$: uniform bound for second-order operators such as $\Gamma_s = \mathbb{E}[\psi(\mathbf{s})\otimes\psi(\mathbf{s})]$ and $\Gamma_{2s} = \mathbb{E}[\psi(\mathbf{s})\otimes\psi(\mathbf{s}')]$, so that $\|\Gamma_s\|_\mathrm{op}, \|\Gamma_{2s}\|_\mathrm{op} \leq M_\Gamma$ (for scalar kernels one may take $M_\Gamma = M_k$).

- $L_k$: Lipschitz constant of the kernel with respect to inputs and we reuse the same symbol when applied to an entire Gram matrix.

- $L_\psi$: Lipschitz constant of the (scalar) feature map, $\|\psi(\mathbf{s}) - \psi(\tilde{\mathbf{s}})\|_{\mathcal{H}_k} \leq L_\psi\|\mathbf{s} - \tilde{\mathbf{s}}\|_2$.

- $L_\mathrm{grad}$: Lipschitz constant of the score-gradient $g(h) = \nabla_h\log\pi_h(\mathbf{a}\mid\mathbf{s})$ with respect to $h$; in our bounds one can take

$$L_\mathrm{grad} = \sqrt{M_K}\|\Sigma^{-1}\|_\mathrm{op}^{1/2}C_\mathrm{ev}.$$

- $M_\Sigma := \|\Sigma^{-1}\|_\mathrm{op}$: boundedness of the Actor covariance.

- $M_a$: bound on action residuals, i.e., $\|\mathbf{a} - h(\mathbf{s})\|_2 \leq M_a$.

- $M_V$: bound on the Value Critic norm in RKHS, $\|w_{\mathsf{V}}^\star\|_{\mathcal{H}_k} \leq M_V$.

- $M_c \coloneqq \max_j \|\mathbf{c}_j\|_2$, $M_\eta \coloneqq \|\eta\|_2$: bounds on the Actor and Critic coefficient vectors.

- $\rho_{\mathsf{V}} \coloneqq \exp\left(-\frac{C_k^2}{2l^2}\right)$: coherence bound for Gaussian kernels (kernel lenth-scale $l$); hence $\lambda_{\min}(\mathbf{K}_{\mathsf{V},\mathsf{V}}) \geq 1 - (q_{\mathsf{V}} - 1)\rho_{\mathsf{V}}$.

- $M_S \coloneqq \sup_{\mathbf{s}} \|\mathbf{s}\|_2$: state-norm bound (used with Mahalanobis RBF perturbations).

- $M$: uniform bound for component kernels in product kernels ($k^{(j)} \leq M$); implies $\|\mu_{\mathcal{C}}\|_{\mathcal{H}} \leq M^{|\overline{\mathcal{C}}|/2}$.

- Sampling/concentration constants: $n_{\text{eff}}$ (effective sample size), $\tau_{\text{mix}}$ (mixing time), confidence level $\delta \in (0,1)$, and Bernstein-type constants $C_B, C_b$.

- Two-timescale stepsizes: $\alpha_t^{\mathsf{v}} = \dfrac{C_{\mathsf{v}}}{(t+1)^\nu}$, $\alpha_t^{\mathsf{h}} = \dfrac{C_{\mathsf{h}}}{(t+1)^\sigma}$ with $0 < \nu < \sigma \leq 1$.

- $L_{\mathsf{V}}$: Lipschitz drift of the Value Critic w.r.t. actor parameters: $\|w_{\mathsf{V}}^\star(h') - w_{\mathsf{V}}^\star(h)\| \leq L_{\mathsf{V}}\|h' - h\|$.

When only scalar kernels are involved, one may safely replace $M_K, M_\Gamma$ by $M_k$. For operator-valued kernels / vector-valued RKHS, keep $M_K$ (kernel boundedness) and $M_\Gamma$ (second-order operator bound) separate.

### C.2. Supporting Lemmas

**Lemma C.1** (Theorem 5 in (Zhang et al., 2020)). *Given a policy $\pi$ for a non-adversarial MDP, its value function is $V_\pi(\mathbf{s})$. Under the adversary $b$ in SA-MDP, for all $\mathbf{s} \in \mathcal{S}$ we have*

$$\max_{\mathbf{s} \in \mathcal{S}} \left\{ V^\pi(\mathbf{s}) - V^{\widetilde{\pi}}(\mathbf{s}) \right\} \leq \alpha \max_{\mathbf{s} \in \mathcal{S}} \max_{\widetilde{\mathbf{s}} \in B(\mathbf{s})} \mathrm{D}_{\mathrm{TV}}(\pi(\cdot|\mathbf{s}), \widetilde{\pi}(\cdot|\widetilde{\mathbf{s}})), \tag{21}$$

*where $\mathrm{D}_{\mathrm{TV}}(\pi(\cdot|\mathbf{s}), \widetilde{\pi}(\cdot|\widetilde{\mathbf{s}}))$ is the total variation distance between $\pi(\cdot|\mathbf{s})$ and $\widetilde{\pi}(\cdot|\widetilde{\mathbf{s}})$, and $\alpha \coloneqq 2[1 + \frac{\gamma}{(1-\gamma)^2}]\max_{(\mathbf{s},\mathbf{a},\mathbf{s}') \in \mathcal{S} \times \mathcal{A} \times \mathcal{S}} |r(\mathbf{s}, \mathbf{a}, \mathbf{s}')|$ is a constant that does not depend on $\pi$ and $\widetilde{\pi}$.*

**Lemma C.2** (Bounded perturbation of RKHS-SHAP values). *Suppose Assumptions 4.4, 4.1, and 4.2 hold. Let $k$ be a product kernel with $d$ bounded component kernels satisfying $|k^{(i)}(\mathbf{s},\mathbf{s})| \leq M$ for all $i \in \mathcal{X}$, and assume $\|w_{\mathsf{V}}\|_{\mathcal{H}_k}^2 \leq M_V$. Define $\delta_\psi \coloneqq \sup_{\mathcal{C} \subseteq \mathcal{X}} \|\psi(\mathbf{s}^{\mathcal{C}}) - \psi(\widetilde{\mathbf{s}}^{\mathcal{C}})\|_{\mathcal{H}_k}^2$, where $\widetilde{\mathbf{s}} \in B(\mathbf{s})$, and let $\delta_{\widetilde{\mathsf{V}}}$ be the projection/perturbation residual given in Lemma C.3. Then, for any feature $i$,*

$$\left\|\phi_i^{(\text{off})} - \widetilde{\phi}_i^{(\text{off})}\right\|_2^2 \leq 4\delta_{\widetilde{\mathsf{V}}}^2 M^d + 4M_V \delta_\psi M^d, \tag{22a}$$

$$\left\|\phi_i^{(\text{on})} - \widetilde{\phi}_i^{(\text{on})}\right\|_2^2 \leq 4\delta_{\widetilde{\mathsf{V}}}^2 M^d + 8M_V M_\Gamma \delta_\psi M^d, \tag{22b}$$

*where $M_\Gamma \coloneqq \sup_{\mathcal{C} \subseteq \mathcal{X}} \|\mu_{\mathbf{s}^{\overline{\mathcal{C}}}|\mathbf{s}^{\mathcal{C}}}\|_{\Gamma_{\mathbf{s}^{\mathcal{C}}}}^2$. If, in addition, $k$ is an RBF kernel with lengthscale $l$ and Assumption 4.5 holds (for $\rho_{\mathsf{V}}$), then $\delta_\psi = 2 - 2\exp\left(-\|\mathbf{s} - \widetilde{\mathbf{s}}\|_2^2/(2l^2)\right)$ and*

$$\left\|\phi_i^{(\text{off})} - \widetilde{\phi}_i^{(\text{off})}\right\|_2^2 \leq \frac{4M^d M_\eta^2 q_{\mathsf{V}}^4 \varepsilon^2}{\left(1 - (q_{\mathsf{V}} - 1)\rho_{\mathsf{V}}\right)^2} \left(\frac{L_k}{1 - (q_{\mathsf{V}} - 1)\rho_{\mathsf{V}}} + M_k L_\psi + 2M_k^{1/2} L_k\right)^2 + 4M_V \left(1 - e^{-\frac{\varepsilon^2}{2l^2}}\right), \tag{23a}$$

$$\left\|\phi_i^{(\text{on})} - \widetilde{\phi}_i^{(\text{on})}\right\|_2^2 \leq \frac{4M^d M_\eta^2 q_{\mathsf{V}}^4 \varepsilon^2}{\left(1 - (q_{\mathsf{V}} - 1)\rho_{\mathsf{V}}\right)^2} \left(\frac{L_k}{1 - (q_{\mathsf{V}} - 1)\rho_{\mathsf{V}}} + M_k L_\psi + 2M_k^{1/2} L_k\right)^2 + 8M_V M_\Gamma \left(1 - e^{-\frac{\varepsilon^2}{2l^2}}\right), \tag{23b}$$

*where $\varepsilon = \max\{\max_j \|\mathbf{s}_j - \widetilde{\mathbf{s}}_j\|, \|\mathbf{s}_\iota - \widetilde{\mathbf{s}}_\iota\|\}$, $M_\eta \coloneqq \sup_t \|\eta_t\|_2$, and $L_k$ and $L_\psi$ are the Lipschitz constants from Assumption 4.4.*

**Lemma C.3** (Perturbation error of the value representation). *Under Assumptions 4.4 and 4.5, consider dictionaries $\mathcal{D}_{\mathsf{V}} = \{\mathbf{s}_j\}_{j=1}^{q_{\mathsf{V}}}$ and $\widetilde{\mathcal{D}}_{\mathsf{V}} = \{\widetilde{\mathbf{s}}_j\}_{j=1}^{q_{\mathsf{V}}}$, and a new center $\mathbf{s}_\iota$ with perturbations bounded by $\max\{\max_j \|\mathbf{s}_j - \widetilde{\mathbf{s}}_j\|, \|\mathbf{s}_\iota - \widetilde{\mathbf{s}}_\iota\|\} \leq \varepsilon$. Let $\rho_{\mathsf{V}} = \exp(-C_k^2/(2l^2))$ and assume both Gram matrices are invertible. Then the residual*

$$\delta_{\widetilde{\mathsf{V}}} \coloneqq \left\| \sum_{j,j'=1}^{q_{\mathsf{V}}} \left[\mathbf{K}_{\mathsf{V},\mathsf{V}}^{-1}\right]_{jj'} k(\mathbf{s}_{j'}, \mathbf{s}_\iota)\eta_\iota \psi(\mathbf{s}_j) - \sum_{j,j'=1}^{q_{\mathsf{V}}} \left[\widetilde{\mathbf{K}}_{\mathsf{V},\mathsf{V}}^{-1}\right]_{jj'} k(\widetilde{\mathbf{s}}_{j'}, \widetilde{\mathbf{s}}_\iota)\eta_\iota \psi(\widetilde{\mathbf{s}}_j) \right\|_{\mathcal{H}_k}$$

*obeys the linear bound*

$$\delta_{\tilde{\mathsf{V}}} \le \frac{M_\eta q^2}{1 - (q_{\mathsf{V}} - 1)\rho_{\mathsf{V}}} \left[ \frac{L_k}{1 - (q_{\mathsf{V}} - 1)\rho_{\mathsf{V}}} + M_k L_\psi + 2M_k^{1/2} L_k \right] \varepsilon.$$

**Lemma C.4** (Projection error for the Value Critic update). *Assume 4.4 and 4.5. For the projected update $\overline{w}_{t+1} = w_t - \alpha_t^{\mathsf{v}} \nabla J(w_t)$ and $w_{t+1} = \Pi_{\mathcal{H}_{\mathcal{D}_{\mathsf{V}},t}}(\overline{w}_{t+1})$, the one-step projection error satisfies*

$$\delta_{\mathsf{PV}} := \|\overline{w}_{t+1} - w_{t+1}\|_{\mathcal{H}_k} \le M_k^{1/2} q_{\mathsf{V}} \varepsilon_{\mathsf{PV}},$$

*where*

$$\varepsilon_{\mathsf{PV}} = \max \left\{ \sup_{\psi_j \in \mathcal{D}_{\mathsf{V}}} \left| \sum_{j'} [\mathbf{K}_{\mathsf{V},\mathsf{V}}^{-1}]_{jj'} k(\mathbf{s}_{j'}, \mathbf{s}_\iota) \eta_\iota - \eta_j \right|, \sup_{\psi_j \in \{\psi_\iota\} \cup (\mathcal{D}_{\mathsf{V}} \setminus \{\psi_{j^\star}\})} \left| \sum_{j'} [\mathbf{K}_{\mathsf{V},\mathsf{V}}^{-1}]_{jj'} k(\mathbf{s}_{j'}, \mathbf{s}_{j^\star}) \eta_{j^\star} - \eta_j \right| \right\}.$$

**Lemma C.5** (Gradient growth for the Value Critic). *Under Assumptions 4.7 and 4.4, for any $t \ge 0$,*

$$\left\| g_t(w_{\mathsf{V},t}) \right\|_{\mathcal{H}_k}^2 \le C_1 \|w_{\mathsf{V},t} - w_{\mathsf{V},t}^\star\|_{\mathcal{H}_k}^2, \qquad C_1 := \left((1+\gamma)M_k + \lambda\right)^2.$$

**Lemma C.6** (Actor drift induced by kernel change). *Let $K(\mathbf{s}, \mathbf{s}') = \kappa_{\mathbf{W}}(\mathbf{s}, \mathbf{s}') \Sigma_{\mathbf{K}}$ with $\|\Sigma_{\mathbf{K}}\|_{\mathrm{op}} \le M_\Sigma$, and*

$$h(\cdot) = \sum_{j=1}^{q_{\mathsf{A}}} K(\cdot, \mathbf{s}_j) \mathbf{c}_j, \qquad h'(\cdot) = \sum_{j=1}^{q_{\mathsf{A}}} K'(\cdot, \mathbf{s}_j) \mathbf{c}_j,$$

*where $K'$ uses $\mathbf{W}'$ instead of $\mathbf{W}$ and $M_A := \max_j \|\mathbf{c}_j\|_2$. If $\|\mathbf{s}\|_2 \le M_S$ for all states and $\sup_{\mathbf{s},\mathcal{C}} \|\mu_{\mathcal{C}}(\mathbf{s})\|_{\mathcal{H}_k} \le C_\mu$, then*

$$\delta_{\mathsf{K},h} \le 16 M_{\Sigma_K}^2 M_S^4 q_{\mathsf{A}}^2 M_a^2 C_\mu^2 \|w_{\mathsf{V}}' - w_{\mathsf{V}}\|_{\mathcal{H}_k}^2.$$

**Lemma C.7** (Smoothness lower bound with projection error). *Let $G(h) := \mathbb{E}_{\nu_{\pi^\star}}[\log \pi_h(\mathbf{a} \mid \mathbf{s})]$ be $L_{\log \pi}$-smooth and concave on $(\mathcal{H}_K, \|\cdot\|_{\mathcal{H}_K})$. Assume $M_K := \sup_{\mathbf{s}} \|K(\mathbf{s}, \mathbf{s})\|_{\mathrm{op}} < \infty$ and the orthogonal projection onto the Actor dictionary subspace is well defined (Assumption 4.5). If $h_{t+1}$ is obtained by projecting $\overline{h}_{t+1}$ and $\delta_{K,t} := \|h_{t+1} - \overline{h}_{t+1}\|_{\mathcal{H}_K}^2$, then*

$$\mathbb{E}_{\nu_{\pi^\star}} \left[\log \pi_{h_{t+1}} - \log \pi_{h_t}\right] \ge \mathbb{E}_{\nu_{\pi^\star}} \left[\nabla \log \pi_{h_t}^\top (h_{t+1} - h_t)\right] - \frac{L_{\mathrm{grad}}}{2} \|h_{t+1} - h_t\|_{\mathcal{H}_K}^2 - 2M_a^2 M_\Sigma^2 \delta_{\mathsf{K},h_t},$$

*where $\delta_{K,t}$ can be further controlled via Lemma C.6.*

**Lemma C.8** (Projection error for the Actor mean). *Assume $M_K := \sup_{\mathbf{s}} \|K(\mathbf{s}, \mathbf{s})\|_{\mathrm{op}} < \infty$ and the block Gram matrix on the Actor dictionary is invertible (Assumption 4.5). For the projected update $\overline{h}_{t+1} = h_t - \alpha_t^{\mathsf{h}} \nabla J(h_t)$ and $h_{t+1} = \Pi_{\mathcal{H}_{K,\mathcal{D}_{\mathsf{A}},t}}(\overline{h}_{t+1})$,*

$$\delta_{\mathsf{PA}} := \|\overline{h}_{t+1} - h_{t+1}\|_{\mathcal{H}_K} \le \sqrt{M_K} q_{\mathsf{A}} \varepsilon_{\mathsf{PA}},$$

*where*

$$\varepsilon_{\mathsf{PA}} = \max \left\{ \sup_{K(\cdot, \mathbf{s}_j) \in \mathcal{D}_{\mathsf{A}}} \left\| [\mathbf{K}_{\mathsf{A},\mathsf{A}}^{-1} \mathbf{K}_{\mathsf{A},\iota}] \mathbf{c}_\iota - \mathbf{c}_j \right\|_2, \sup_{K(\cdot, \mathbf{s}_j) \in \{\mathbf{s}_\iota\} \cup (\mathcal{D}_{\mathsf{A}} \setminus \{\mathbf{s}_{j^\star}\})} \left\| [\mathbf{K}_{\mathsf{A},\mathsf{A}}^{-1} \mathbf{K}_{\mathsf{A},j^\star}] \mathbf{c}_{j^\star} - \mathbf{c}_j \right\|_2 \right\}.$$

**Lemma C.9** (Lipschitz drift of the target Critic). *Under Assumptions 4.7 and 4.4, for any $h, h' \in \mathcal{H}_K$,*

$$\|w_{\mathsf{V}}^\star(h') - w_{\mathsf{V}}^\star(h)\|_{\mathcal{H}_k} \le L_{\mathsf{V}} \|h' - h\|_{\mathcal{H}_K}, \qquad L_{\mathsf{V}} := C_\nu \sqrt{M_k} \left( \frac{1}{\lambda} + \frac{2(1+\gamma)M_k}{\lambda^2} \right).$$

**Lemma C.10** (Performance Difference Lemma, discounted MDP). *For any two policies $\pi, \pi'$,*

$$J(\pi') - J(\pi) = \frac{1}{1-\gamma} \mathbb{E}_{(\mathbf{s},\mathbf{a}) \sim \nu_{\pi'}} \left[A^\pi(\mathbf{s}, \mathbf{a})\right].$$

**Lemma C.11** (Refined tracking error bound for the RKHS Critic under two time-scales). *Let $\mathcal{H}_\psi$ be the Critic RKHS with bounded feature map $\|\psi(s)\|_{\mathcal{H}_\psi}^2 \le M_k$. Suppose the on-policy sampling process is geometric mixing so that the effective sample size is $n_{\mathrm{eff}} \asymp n/\tau_{\mathrm{mix}}$. Consider the two time-scale stepsizes $\alpha_t^{\mathsf{v}} > 0$ and $\alpha_t^{\mathsf{h}} > 0$. Then there exists $T_0 < \infty$, under polynomial stepsizes $\alpha_t^{\mathsf{v}} = \frac{C_{\mathsf{v}}}{(t+1)^\nu}$ and $\alpha_t^{\mathsf{h}} = \frac{C_{\mathsf{h}}}{(t+1)^\sigma}$ with $0 < \nu < \sigma \le 1$, such that the decay rates satisfy*

$$\mathbb{E}\left[\|w_{\mathsf{V},t} - w_{\mathsf{V},t}^\star\|_{\mathcal{H}_\psi}^2\right] = \begin{cases} O\left((t+1)^{-\nu}\right) + M_k q_{\mathsf{V}}^2 \varepsilon_{\mathsf{PV}}^2 + O\left(\frac{1}{n_{\mathrm{eff}}}\right), & \sigma > \nu, \\ O\left((t+1)^{-\nu} \log^2 t\right) + M_k q_{\mathsf{V}}^2 \varepsilon_{\mathsf{PV}}^2 + O\left(\frac{1}{n_{\mathrm{eff}}}\right), & \sigma = \nu \text{ (borderline)}. \end{cases}$$

## C.3. Proof of Proposition 4.3

Recall the policy is defined as a Gaussian distribution. Then the functional gradient of the log-policy w.r.t. $h$ is given by

$$\nabla_h \log \pi_{h,\Sigma}(\mathbf{a} \mid \mathbf{s}) = K(\mathbf{s}, \cdot)\Sigma^{-1}(\mathbf{a} - h(\mathbf{s})) \in \mathcal{H}_K, \tag{24}$$

under Proposition B.2. Therefore, the corresponding Fisher information operator is

$$\begin{aligned} F(h) &= \mathbb{E}_{\mathbf{s}\sim\nu_{\pi_h},a\sim\pi_{h,\Sigma}(\cdot|\mathbf{s})}\left[\nabla_h \log \pi_{h,\Sigma}(\mathbf{a} \mid \mathbf{s}) \otimes \nabla_h \log \pi_{h,\Sigma}(\mathbf{a} \mid \mathbf{s})\right] \\ &= \mathbb{E}_{\mathbf{s}\sim\nu_{\pi_h}}\left[K(\mathbf{s}, \cdot) \otimes K(\mathbf{s}, \cdot) \cdot \Sigma^{-1}\right], \end{aligned} \tag{25}$$

where the last equation comes from $\mathbb{E}\left[(\mathbf{a} - h(\mathbf{s}))(\mathbf{a} - h(\mathbf{s}))^{\top}\right] = \Sigma$.

We assume a coercivity condition that there exists a constant $\lambda_F > 0$ such that

$$\langle f, F(h)f\rangle_{\mathcal{H}_K} \geq \lambda_F \cdot \|f\|^2_{\mathcal{H}_K}, \quad \forall f \in \mathcal{H}_K. \tag{26}$$

This ensures that the Fisher operator $F(h)$ is positive definite on $\mathcal{H}_K$, and enables natural gradient updates in RKHS.

## C.4. Proof of Theorem 4.8

According to Lemma C.1, we just need to bound $\mathrm{D}_{\mathrm{TV}}(\pi(\cdot|\mathbf{s}), \widetilde{\pi}(\cdot|\widetilde{\mathbf{s}}))$. Specifically, the policy of RSA2C is characterize as Gaussian policy. We upper bound the TV distance as

$$\begin{aligned} \mathrm{D}_{\mathrm{TV}}(\pi(\cdot|\mathbf{s}), \widetilde{\pi}(\cdot|\widetilde{\mathbf{s}})) &\leq \sqrt{\frac{1}{2}\mathrm{D}_{\mathrm{KL}}(\pi(\cdot|\mathbf{s})\|\widetilde{\pi}(\cdot|\widetilde{\mathbf{s}}))} \\ &= \sqrt{\frac{1}{4}\left(\log\left|\Sigma_{\widetilde{\mathbf{s}}}\Sigma_{\mathbf{s}}^{-1}\right| + \mathrm{tr}\left(\Sigma_{\widetilde{\mathbf{s}}}^{-1}\Sigma_{\mathbf{s}}\right) + \left(\widetilde{h}(\widetilde{\mathbf{s}}) - h(\mathbf{s})\right)^{\top}\Sigma_{\widetilde{\mathbf{s}}}^{-1}\left(\widetilde{h}(\widetilde{\mathbf{s}}) - h(\mathbf{s})\right) - m\right)} \\ &= \sqrt{\frac{1}{4}\left(\left(\widetilde{h}(\widetilde{\mathbf{s}}) - h(\mathbf{s})\right)^{\top}\Sigma^{-1}\left(\widetilde{h}(\widetilde{\mathbf{s}}) - h(\mathbf{s})\right)\right)}. \end{aligned} \tag{27}$$

Here, the last equality holds due to $\Sigma_{\mathbf{s}} = \Sigma_{\widetilde{\mathbf{s}}} = \Sigma$ followed from line 9 in Algorithm 1.

Armed with eigen-decomposition method, we further bound the result in (27) as

$$\mathrm{D}_{\mathrm{TV}}(\pi(\cdot|\mathbf{s}), \widetilde{\pi}(\cdot|\widetilde{\mathbf{s}})) \leq \frac{\sqrt{\lambda_{\max}(\Sigma^{-1})}}{2}\left\|\widetilde{h}(\widetilde{\mathbf{s}}) - h(\mathbf{s})\right\|_2 = \frac{M_\Sigma}{2}\left\|\widetilde{h}(\widetilde{\mathbf{s}}) - h(\mathbf{s})\right\|_2, \tag{28}$$

where $M_\Sigma := \sqrt{\lambda_{\max}(\Sigma^{-1})}$.

Armed with the definition of mapping function and (3), we have

$$
\begin{aligned}
\left\|\widetilde{h}(\widetilde{\mathbf{s}}) - h(\mathbf{s})\right\|_2 &= \left\|\frac{1}{q_{\mathsf{A}}} \sum_{(\mathbf{s}_j, \mathbf{c}_j) \in \mathcal{D}_{\mathsf{A}}} K(\mathbf{s}, \mathbf{s}_j) \mathbf{c}_j - \frac{1}{q} \sum_{(\mathbf{s}_j, \mathbf{c}_j) \in \mathcal{D}_{\mathsf{A}}} \widetilde{K}(\widetilde{\mathbf{s}}, \widetilde{\mathbf{s}}_j) \widetilde{\mathbf{c}}_j \right\|_2 \\
&= \left\|\frac{1}{q_{\mathsf{A}}} \sum_{(\mathbf{s}_j, \mathbf{c}_j) \in \mathcal{D}_{\mathsf{A}}} \left[ K(\mathbf{s}, \mathbf{s}_j) \mathbf{c}_j - \widetilde{K}(\widetilde{\mathbf{s}}, \widetilde{\mathbf{s}}_j) \widetilde{\mathbf{c}}_j \right] \right\|_2 \\
&\overset{(i)}{\leq} \frac{1}{q_{\mathsf{A}}} \sum_{(\mathbf{s}_j, \mathbf{c}_j) \in \mathcal{D}_{\mathsf{A}}} \left\| K(\mathbf{s}, \mathbf{s}_j) \mathbf{c}_j - \widetilde{K}(\widetilde{\mathbf{s}}, \widetilde{\mathbf{s}}_j) \widetilde{\mathbf{c}}_j \right\|_2 \\
&= \frac{1}{q_{\mathsf{A}}} \sum_{(\mathbf{s}_j, \mathbf{c}_j) \in \mathcal{D}_{\mathsf{A}}} \left\| \kappa_\phi(\mathbf{s}, \mathbf{s}_j) \Sigma_{\mathsf{K}} \mathbf{c}_j - \kappa_{\widetilde{\mathbf{W}}}(\widetilde{\mathbf{s}}, \widetilde{\mathbf{s}}_j) \Sigma_{\mathsf{K}} \widetilde{\mathbf{c}}_j \right\|_2 \\
&\overset{(ii)}{\leq} \frac{1}{q_{\mathsf{A}}} \sum_{(\mathbf{s}_j, \mathbf{c}_j) \in \mathcal{D}_{\mathsf{A}}} \left[ \left| \kappa_\phi(\mathbf{s}, \mathbf{s}_j) - \kappa_{\widetilde{\mathbf{W}}}(\widetilde{\mathbf{s}}, \widetilde{\mathbf{s}}_j) \right| \, \|\Sigma_{\mathsf{K}}\|_{\mathrm{op}} \|\mathbf{c}_j\|_2 + |\kappa_\phi(\mathbf{s}, \mathbf{s}_j)| \, \|\Sigma_{\mathsf{K}}\|_{\mathrm{op}} \|\mathbf{c}_j - \widetilde{\mathbf{c}}_j\|_2 \right] \\
&\overset{(iii)}{\leq} \frac{1}{q_{\mathsf{A}}} \sum_{(\mathbf{s}_j, \mathbf{c}_j) \in \mathcal{D}_{\mathsf{A}}} \left[ \lambda_{\max}(\Sigma_{\mathsf{K}}) M_C \delta_{\widetilde{\kappa}, j} + M_\kappa \lambda_{\max}(\Sigma_{\mathsf{K}}) \delta_{\widetilde{c}, j} \right] \\
&\leq \lambda_{\max}(\Sigma_{\mathsf{K}}) M_C \delta_{\widetilde{\kappa}} + M_\kappa \lambda_{\max}(\Sigma_{\mathsf{K}}) \delta_{\widetilde{c}}, \tag{29}
\end{aligned}
$$

where (i) and (ii) hold due to Cauchy-Schwarz inequality, and (iii) comes from $\|\Sigma_{\mathsf{K}}\|_{\mathrm{op}} = \lambda_{\max}(\Sigma_{\mathsf{K}})$, $\|\kappa_\phi(\mathbf{s}, \mathbf{s}_j)\|_2 \leq M_\kappa$, and the last inequality exists with $\delta_{\widetilde{\kappa}} := \sup_j \delta_{\widetilde{\kappa}, j}$ and $\delta_{\widetilde{c}} := \sup_j \delta_{\widetilde{c}, j}$.

In this paper, we utilize RKHS-SHAP by RBF kernel as (3). Therefore, armed with $w = \mathrm{diag}\{\{\phi_i\}_{i=1}^d\}$, we further bound $\delta_{\widetilde{\kappa}}$ and $\delta_{\widetilde{c}}$ as follows.

**Bound $\delta_{\widetilde{\kappa}}$.** For the difference of kernel function, we have

$$
\begin{aligned}
\delta_{\widetilde{\kappa}, j} &= \left| \exp\left( -\frac{1}{2}(\mathbf{s} - \mathbf{s}_j)^\top w (\mathbf{s} - \mathbf{s}_j) \right) - \exp\left( -\frac{1}{2}(\widetilde{\mathbf{s}} - \widetilde{\mathbf{s}}_j)^\top \widetilde{w} (\widetilde{\mathbf{s}} - \widetilde{\mathbf{s}}_j) \right) \right| \\
&\leq \frac{1}{2} \left| (\mathbf{s} - \mathbf{s}_j)^\top w (\mathbf{s} - \mathbf{s}_j) - (\widetilde{\mathbf{s}} - \widetilde{\mathbf{s}}_j)^\top \widetilde{w} (\widetilde{\mathbf{s}} - \widetilde{\mathbf{s}}_j) \right| \exp\left( -\frac{1}{2} \min\left( (\mathbf{s} - \mathbf{s}_j)^\top w (\mathbf{s} - \mathbf{s}_j), (\widetilde{\mathbf{s}} - \widetilde{\mathbf{s}}_j)^\top \widetilde{w} (\widetilde{\mathbf{s}} - \widetilde{\mathbf{s}}_j) \right) \right) \\
&\leq \frac{1}{2} \exp\left( -\frac{1}{2} M_S^2 \right) \left| (\mathbf{s} - \mathbf{s}_j)^\top w (\mathbf{s} - \mathbf{s}_j) - (\widetilde{\mathbf{s}} - \widetilde{\mathbf{s}}_j)^\top \widetilde{w} (\widetilde{\mathbf{s}} - \widetilde{\mathbf{s}}_j) \right| \\
&\leq \frac{1}{2} \exp\left( -\frac{1}{2} M_S^2 \right) \underbrace{\left| (\mathbf{s} - \mathbf{s}_j)^\top w (\mathbf{s} - \mathbf{s}_j) - (\widetilde{\mathbf{s}} - \widetilde{\mathbf{s}}_j)^\top w (\widetilde{\mathbf{s}} - \widetilde{\mathbf{s}}_j) \right|}_{A_1} \\
&\quad + \frac{1}{2} \exp\left( -\frac{1}{2} M_S^2 \right) \underbrace{\left| (\widetilde{\mathbf{s}} - \widetilde{\mathbf{s}}_j)^\top w (\widetilde{\mathbf{s}} - \widetilde{\mathbf{s}}_j) - (\widetilde{\mathbf{s}} - \widetilde{\mathbf{s}}_j)^\top \widetilde{w} (\widetilde{\mathbf{s}} - \widetilde{\mathbf{s}}_j) \right|}_{A_2}, \tag{30}
\end{aligned}
$$

where the first inequality holds due to the mean value theorem, the second one comes from $\|\mathbf{s} - \mathbf{s}_j\|_2 \leq M_S$, and the last inequality exists by Cauchy-Schwarz inequality.

Firstly, we consider the upper bound on the state perturbation term, denoted as $A_1$. Specifically, we have

$$
\begin{aligned}
A_1 &= \left| (\mathbf{s} - \mathbf{s}_j)^\top w (\mathbf{s} - \mathbf{s}_j) - (\widetilde{\mathbf{s}} - \widetilde{\mathbf{s}}_j)^\top w (\widetilde{\mathbf{s}} - \widetilde{\mathbf{s}}_j) \right| \\
&= \left| (\mathbf{s} - \mathbf{s}_j)^\top w (\mathbf{s} - \mathbf{s}_j) - ((\mathbf{s} - \mathbf{s}_j) - (\mathbf{s} - \widetilde{\mathbf{s}}) + (\mathbf{s}_j - \widetilde{\mathbf{s}}_j))^\top w ((\mathbf{s} - \mathbf{s}_j) - (\mathbf{s} - \widetilde{\mathbf{s}}) + (\mathbf{s}_j - \widetilde{\mathbf{s}}_j)) \right| \\
&= \left| 2((\mathbf{s} - \widetilde{\mathbf{s}}) - (\mathbf{s}_j - \widetilde{\mathbf{s}}_j))^\top w (\mathbf{s} - \mathbf{s}_j) - ((\mathbf{s} - \widetilde{\mathbf{s}}) - (\mathbf{s}_j - \widetilde{\mathbf{s}}_j))^\top w ((\mathbf{s} - \widetilde{\mathbf{s}}) - (\mathbf{s}_j - \widetilde{\mathbf{s}}_j)) \right| \\
&\leq 4 \|\mathbf{s} - \widetilde{\mathbf{s}}\|_2 M_S + 4 \|\mathbf{s} - \widetilde{\mathbf{s}}\|_2^2 M_\phi \tag{31}
\end{aligned}
$$

where the inequality follows from the Cauchy-Schwarz inequality.

Next, for $A_2$, which represents the RKHS-SHAP perturbation bound, we derive

$$A_2 = \left|(\widetilde{\mathbf{s}} - \widetilde{\mathbf{s}}_j)^\top w(\widetilde{\mathbf{s}} - \widetilde{\mathbf{s}}_j) - (\widetilde{\mathbf{s}} - \widetilde{\mathbf{s}}_j)^\top \widetilde{w}(\widetilde{\mathbf{s}} - \widetilde{\mathbf{s}}_j)\right| = \left|(\widetilde{\mathbf{s}} - \widetilde{\mathbf{s}}_j)^\top (w - \widetilde{w})(\widetilde{\mathbf{s}} - \widetilde{\mathbf{s}}_j)\right| \leq \sum_{i=1}^{d} M_S^2 \left|\phi_i - \widetilde{\phi}_i\right|. \quad (32)$$

Therefore, we have

$$\delta_{\widetilde{\kappa}} \leq 2e^{-\frac{1}{2}M_S^2}\left(\varepsilon M_S + \varepsilon^2 M_\phi\right) + \frac{1}{2}e^{-\frac{1}{2}M_S^2}\sum_{i=1}^{d} M_S^2 \left|\phi_i - \widetilde{\phi}_i\right|. \quad (33)$$

**Bound $\delta_{\widetilde{c}}$.** For online sparsification, we consider the following optimization problem:

$$\min_{\{\mathbf{c}_j\}\in\mathcal{D}_A} \left\|\sum_j K(\mathbf{s}_j,\cdot)\mathbf{c}_j - K(\mathbf{s}_\iota,\cdot)\mathbf{c}_\iota\right\|_{\mathcal{H}_K}^2. \quad (34)$$

By expanding the norm and applying the reproducing property of the RKHS, we obtain:

$$\mathcal{L}(\{\mathbf{c}_j\}) = \left\langle \sum_j K(\mathbf{s}_j,\cdot)\mathbf{c}_j - K(\mathbf{s}_\iota,\cdot)\mathbf{c}_\iota, \sum_{j'} K(\mathbf{s}_{j'},\cdot)\mathbf{c}_{j'} - K(\mathbf{s}_\iota,\cdot)\mathbf{c}_\iota\right\rangle_{\mathcal{H}_K}$$

$$= \sum_{j,j'}\langle\mathbf{c}_j, K(\mathbf{s}_j,\mathbf{s}_{j'})\mathbf{c}_{j'}\rangle - 2\sum_j\langle\mathbf{c}_j, K(\mathbf{s}_j,\mathbf{s}_\iota)\mathbf{c}_\iota\rangle + \langle\mathbf{c}_\iota, K(\mathbf{s}_\iota,\mathbf{s}_\iota)\mathbf{c}_\iota\rangle. \quad (35)$$

Let $\mathbf{C} = [\mathbf{c}_1,\ldots,\mathbf{c}_{q_A}]^\top \in \mathbb{R}^{q_A\times d}$ denote the coefficient matrix, $\mathbf{K}_{A,A} \in \mathbb{R}^{q_A\times q_A}$ the Gram matrix over the dictionary, and $\mathbf{K}_\iota \in \mathbb{R}^{q_A\times d}$ the cross-kernel matrix with $j$-th row given by $K(\mathbf{s}_j,\mathbf{s}_\iota)\mathbf{c}_\iota$. The objective function becomes:

$$\mathcal{L}(\mathbf{C}) = \mathrm{Tr}(\mathbf{C}^\top \mathbf{K}_{A,A}\mathbf{C}) - 2\,\mathrm{Tr}(\mathbf{C}^\top \mathbf{K}_\iota) + \mathbf{c}_\iota^\top K(\mathbf{s}_\iota,\mathbf{s}_\iota)\mathbf{c}_\iota. \quad (36)$$

Setting the gradient $\nabla_{\mathbf{C}}\mathcal{L} = 0$ yields the closed-form optimal solution:

$$\mathbf{C}^* = \mathbf{K}_{A,A}^{-1}\mathbf{K}_\iota. \quad (37)$$

The optimal coefficient for each basis function is thus given by:

$$\mathbf{c}_j^* = \sum_{j'=1}^{q_A}\left[\mathbf{K}_{A,A}^{-1}\right]_{jj'} K(\mathbf{s}_{j'},\mathbf{s}_\iota)\mathbf{c}_\iota. \quad (38)$$

This solution corresponds to the orthogonal projection of the new kernel function $K(\mathbf{s}_\iota,\cdot)\mathbf{c}_\iota$ onto the subspace spanned by the dictionary atoms $\{K(\mathbf{s}_j,\cdot)\}$.

Therefore, we have

$$\delta_{\widetilde{c},j} = \left\|\sum_{j'=1}^{q_A}\left[\mathbf{K}_{A,A}^{-1}\right]_{jj'} K(\mathbf{s}_{j'},\mathbf{s}_\iota)\mathbf{c}_\iota - \sum_{j'=1}^{q_A}\left[\widetilde{\mathbf{K}}_{A,A}^{-1}\right]_{jj'} K(\widetilde{\mathbf{s}}_{j'},\mathbf{s}_\iota)\mathbf{c}_\iota\right\|_2$$

$$\leq \sum_{j'=1}^{q_A}\left\|\left[\mathbf{K}_{A,A}^{-1}\right]_{jj'} K(\mathbf{s}_{j'},\mathbf{s}_\iota)\mathbf{c}_\iota - \left[\widetilde{\mathbf{K}}_{A,A}^{-1}\right]_{jj'} K(\widetilde{\mathbf{s}}_{j'},\mathbf{s}_\iota)\mathbf{c}_\iota\right\|_2$$

$$= \sum_{j'=1}^{q_A}\left\|\left[\mathbf{K}_{A,A}^{-1}\right]_{jj'}\kappa_\phi(\mathbf{s}_{j'},\mathbf{s}_\iota)\Sigma_K\mathbf{c}_\iota - \left[\widetilde{\mathbf{K}}_{A,A}^{-1}\right]_{jj'}\kappa_\phi(\widetilde{\mathbf{s}}_{j'},\mathbf{s}_\iota)\Sigma_K\mathbf{c}_\iota\right\|_2$$

$$\leq \sum_{j'=1}^{q_A}\left\|\left[\mathbf{K}_{A,A}^{-1}\right]_{jj'}\kappa_\phi(\mathbf{s}_{j'},\mathbf{s}_\iota) - \left[\widetilde{\mathbf{K}}_{A,A}^{-1}\right]_{jj'}\kappa_\phi(\widetilde{\mathbf{s}}_{j'},\mathbf{s}_\iota)\right\|_{\mathcal{H}_K}\|\Sigma_K\|_2\|\mathbf{c}_\iota\|_2$$

$$\leq \lambda_{\max}(\Sigma_K)M_c\sum_{j'=1}^{q_A}\left\|\left[\mathbf{K}_{A,A}^{-1}\right]_{jj'}\kappa_\phi(\mathbf{s}_{j'},\mathbf{s}_\iota) - \left[\widetilde{\mathbf{K}}_{A,A}^{-1}\right]_{jj'}\kappa_\phi(\widetilde{\mathbf{s}}_{j'},\mathbf{s}_\iota)\right\|_{\mathcal{H}_K}. \quad (39)$$

We consider bounding the RKHS norm

$$\left\| \left[ \mathbf{K}_{\mathsf{A},\mathsf{A}}^{-1} \right]_{jj'} \kappa_\phi(\mathbf{s}_{j'}, \mathbf{s}_\iota) - \left[ \widetilde{\mathbf{K}}_{\mathsf{A},\mathsf{A}}^{-1} \right]_{jj'} \kappa_\phi(\widetilde{\mathbf{s}}_{j'}, \mathbf{s}_\iota) \right\|_{\mathcal{H}_K}, \tag{40}$$

where $\mathbf{K}_{\mathsf{A},\mathsf{A}}$ and $\widetilde{\mathbf{K}}_{\mathsf{A},\mathsf{A}}$ denote Gram matrices over kernel dictionaries $\mathcal{D}_{\mathsf{A}} = \{\mathbf{s}_j\}$ and its perturbed version $\widetilde{\mathcal{D}}_{\mathsf{A}} = \{\widetilde{\mathbf{s}}_j\}$, respectively. Using the triangle inequality and RKHS norm properties, we decompose:

$$\left\| \left[ \mathbf{K}_{\mathsf{A},\mathsf{A}}^{-1} \right]_{jj'} \kappa_\phi(\mathbf{s}_{j'}, \mathbf{s}_\iota) - \left[ \widetilde{\mathbf{K}}_{\mathsf{A},\mathsf{A}}^{-1} \right]_{jj'} \kappa_\phi(\widetilde{\mathbf{s}}_{j'}, \mathbf{s}_\iota) \right\|_{\mathcal{H}_K}$$

$$\leq \left| \left[ \mathbf{K}_{\mathsf{A},\mathsf{A}}^{-1} - \widetilde{\mathbf{K}}_{\mathsf{A},\mathsf{A}}^{-1} \right]_{jj'} \right| \cdot \| \kappa_\phi(\mathbf{s}_{j'}, \cdot) \|_{\mathcal{H}_K} + \left| \left[ \widetilde{\mathbf{K}}_{\mathsf{A},\mathsf{A}}^{-1} \right]_{jj'} \right| \cdot \| \kappa_\phi(\mathbf{s}_{j'}, \cdot) - \kappa_\phi(\widetilde{\mathbf{s}}_{j'}, \cdot) \|_{\mathcal{H}_K}$$

$$\leq M_\kappa \left| \left[ \mathbf{K}_{\mathsf{A},\mathsf{A}}^{-1} - \widetilde{\mathbf{K}}_{\mathsf{A},\mathsf{A}}^{-1} \right]_{jj'} \right| + \left| \left[ \widetilde{\mathbf{K}}_{\mathsf{A},\mathsf{A}}^{-1} \right]_{jj'} \right| \cdot \| \kappa_\phi(\mathbf{s}_{j'}, \cdot) - \kappa_\phi(\widetilde{\mathbf{s}}_{j'}, \cdot) \|_{\mathcal{H}_K}. \tag{41}$$

Then, similar to the bound of $\delta_\kappa$, we have

$$\| \kappa_\phi(\mathbf{s}_{j'}, \cdot) - \kappa_\phi(\widetilde{\mathbf{s}}_{j'}, \cdot) \|_{\mathcal{H}_K} \leq 2 e^{-\frac{1}{2} M_S^2} \left( \varepsilon M_S + \varepsilon^2 M_\phi \right) + \frac{1}{2} e^{-\frac{1}{2} M_S^2} \sum_{i=1}^{d} M_S^2 \left\| \phi_i - \widetilde{\phi}_i \right\|_2. \tag{42}$$

Moreover, we apply the Banach perturbation lemma to the inverse kernel matrices. Then:

$$\left\| \mathbf{K}_{\mathsf{A},\mathsf{A}}^{-1} - \widetilde{\mathbf{K}}_{\mathsf{A},\mathsf{A}}^{-1} \right\| \leq \| \mathbf{K}_{\mathsf{A},\mathsf{A}}^{-1} \| \cdot \| \widetilde{\mathbf{K}}_{\mathsf{A},\mathsf{A}} - \mathbf{K}_{\mathsf{A},\mathsf{A}} \| \cdot \| \widetilde{\mathbf{K}}_{\mathsf{A},\mathsf{A}}^{-1} \|. \tag{43}$$

Suppose the dictionary centers satisfy a minimum separation $C_K > 0$ and $\kappa_\phi$ is Gaussian. Then off-diagonal terms are bounded by $\rho_{\mathsf{A}} := \exp\left( -\frac{C_K^2}{2l^2} \right)$ and the smallest eigenvalue of $\mathbf{K}_{\mathsf{A},\mathsf{A}}$ satisfies

$$\lambda_{\min}(\mathbf{K}_{\mathsf{A},\mathsf{A}}) \geq 1 - (q_{\mathsf{A}} - 1)\rho_{\mathsf{A}}, \quad \Rightarrow \quad \| \mathbf{K}_{\mathsf{A},\mathsf{A}}^{-1} \| \leq \frac{1}{1 - (q_{\mathsf{A}} - 1)\rho_{\mathsf{A}}}. \tag{44}$$

Combining with the kernel matrix perturbation

$$\| \widetilde{\mathbf{K}}_{\mathsf{A},\mathsf{A}} - \mathbf{K}_{\mathsf{A},\mathsf{A}} \| \leq 2 e^{-\frac{1}{2} M_S^2} \left( \varepsilon M_S + \varepsilon^2 M_\phi \right) + \frac{1}{2} e^{-\frac{1}{2} M_S^2} \sum_{i=1}^{d} M_S^2 \left\| \phi_i - \widetilde{\phi}_i \right\|_2, \tag{45}$$

we obtain the final upper bound:

$$\delta_{\widetilde{c}} \leq \frac{e^{-\frac{1}{2} M_S^2} M_{\Sigma_\mathsf{K}} M_c q}{1 - (q_{\mathsf{A}} - 1)\rho_{\mathsf{A}}} \left[ \frac{M_\kappa}{1 - (q_{\mathsf{A}} - 1)\rho_{\mathsf{A}}} + 1 \right] \cdot \left[ 2 \left( \varepsilon M_S + \varepsilon^2 M_\phi \right) + \frac{1}{2} \sum_{i=1}^{d} M_S^2 \left\| \phi_i - \widetilde{\phi}_i \right\|_2 \right], \tag{46}$$

where $M_{\Sigma_\mathsf{K}} := \lambda_{\max}(\Sigma_\mathsf{K})$. This result characterizes the sensitivity of the RKHS element to perturbations in both the kernel centers and the Gram matrix structure.

**Combining two items.** Thus, for the case of RBF kernel $K$, there is

$$\left\| \widetilde{h}(\widetilde{\mathbf{s}}) - h(\mathbf{s}) \right\|_2 \leq M_{\Sigma_\mathsf{K}} \left( M_C + \frac{M_{\Sigma_\mathsf{K}} M_\kappa M_c q}{1 - (q_{\mathsf{A}} - 1)\rho_{\mathsf{A}}} \left[ \frac{M_\kappa}{1 - (q_{\mathsf{A}} - 1)\rho_{\mathsf{A}}} + 1 \right] \right)$$

$$\times \left( 2 e^{-\frac{1}{2} M_S^2} \left( \varepsilon M_S + \varepsilon^2 M_\phi \right) + \frac{1}{2} e^{-\frac{1}{2} M_S^2} \sum_{i=1}^{d} M_S^2 \left\| \phi_i - \widetilde{\phi}_i \right\|_2 \right). \tag{47}$$

According to Lemma C.2, we obtain the error of mapping function for general kernel $k$ as

$$\left\|\widetilde{h}^{(\text{off})}(\widetilde{\mathbf{s}}) - h^{(\text{off})}(\mathbf{s})\right\|_2 \leq C_0 \left( 2\left(\varepsilon M_S + \varepsilon^2\right) + M_S^2 d\sqrt{\delta_{\mathsf{V}}^2 M^d + M_V \delta_\psi M^d} \right) \tag{48a}$$

$$\left\|\widetilde{h}^{(\text{on})}(\widetilde{\mathbf{s}}) - h^{(\text{on})}(\mathbf{s})\right\|_2 \leq C_0 \left( 2\left(\varepsilon M_S + \varepsilon^2\right) + M_S^2 d\sqrt{\delta_{\mathsf{V}}^2 M^d + 2M_V M_\Gamma \delta_\psi M^d} \right), \tag{48b}$$

where

$$C_0 = M_{\Sigma_{\mathsf{K}}} e^{-\frac{1}{2}M_S^2} \left( M_C + \frac{M_{\Sigma_{\mathsf{K}}} M_\kappa M_c q_{\mathsf{A}}}{1 - (q_{\mathsf{A}} - 1)\rho_{\mathsf{A}}} \left[ \frac{M_\kappa}{1 - (q_{\mathsf{A}} - 1)\rho_{\mathsf{A}}} + 1 \right] \right).$$

For the case of RBF kernel $k$, there is

$$\left\|\widetilde{h}^{(\text{off})}(\widetilde{\mathbf{s}}) - h^{(\text{off})}(\mathbf{s})\right\|_2 \leq C_0 \left( 2\left(\varepsilon M_S + \varepsilon^2 M_\phi\right) + M_S^2 d\sqrt{C_2 \varepsilon^2 + M_V \left(1 - \exp\left(-\frac{\varepsilon^2}{2l^2}\right)\right)} \right) \tag{49a}$$

$$\left\|\widetilde{h}^{(\text{on})}(\widetilde{\mathbf{s}}) - h^{(\text{on})}(\mathbf{s})\right\|_2 \leq C_0 \left( 2\left(\varepsilon M_S + \varepsilon^2 M_\phi\right) + M_S^2 d\sqrt{C_2 \varepsilon^2 + 2M_V M_\Gamma \left(1 - \exp\left(-\frac{\varepsilon^2}{2l^2}\right)\right)} \right), \tag{49b}$$

where

$$C_2 = \frac{M^d M_w^2 L_k^2 q_{\mathsf{A}}^2}{(1 - (q_{\mathsf{V}} - 1)\rho_{\mathsf{V}})^2} \left( \frac{1}{1 - (q_{\mathsf{V}} - 1)\rho_{\mathsf{V}}} + 1 \right)^2.$$

Now, we are ready to establish Theorem 4.8. Recall (28), and we are ready to complete the proof.

Let $J(\pi) = \mathbb{E}_{s_0 \sim \rho}\left[V^\pi(s_0)\right]$. Then, by Pinsker's inequality,

$$\mathbb{E}\left[J(\pi) - J(\tilde{\pi})\right] \leq \max_{\mathbf{s} \in \mathcal{S}} \left\{ V^\pi(\mathbf{s}) - V^{\widetilde{\pi}}(\mathbf{s}) \right\} \leq \alpha \max_s \max_{\tilde{s} \in \mathcal{B}(s)} D_{\text{TV}}\left(\pi(\cdot \mid s), \tilde{\pi}(\cdot \mid \tilde{s})\right). \tag{50}$$

We complete the proof of Theorem 4.8.

### C.5. Proof of Theorem 4.9

**Proof sketch of Theorem 4.9.** (i) *One-step improvement.* Using the compatible feature representation $A^{\pi_{h_t}} = \langle w_{\mathsf{A},t}, g(h_t)\rangle_{\mathcal{H}_K}$ and the performance-difference lemma, we study the functional $D(h) = \mathbb{E}_{\nu_{\pi^\star}}[\log \pi_h]$. By $L_{\log \pi}$-smoothness, the update $h_{t+1}$ (unprojected step plus projection) yields a descent-type inequality for $D(h_t) - D(h_{t+1})$ that isolates: a smoothness penalty from $\|h_{t+1} - h_t\|$, a projection/sparsification error, a kernel-drift term from changing features, and stochastic terms coming from the score estimate $\hat{g}(h_t)$. (ii) *Bounding residuals.* Each remainder is controlled by standard ingredients: (a) critic tracking and the Lipschitz dependence of $w_{\mathsf{V}}^\star(h)$ bound the kernel-drift; (b) sparsification guarantees bound $\|h_{t+1} - \overline{h}_{t+1}\|$; (c) second-moment bounds for $\hat{g}(h_t)$ control variance; and (d) Young/Cauchy-Schwarz inequalities handle the bias and error-variance coupling. Altogether the per-step remainder scales like $(\alpha_t^{\mathsf{h}})^2 + \rho_t + q_{\mathsf{A}}^2 \varepsilon_{\mathsf{PA}}^2 + n_{\text{eff}}^{-1}$, with $\rho_t$ capturing the critic-actor time-scale gap. (iii) *Telescoping and rates.* Summing the one-step inequality over $t$, choosing polynomial step sizes $\alpha_t^{\mathsf{h}} = \alpha_0(t+1)^{-\sigma}$ and $\beta_t = \beta_0(t+1)^{-\nu}$ with $0 < \nu < \sigma$, and letting the implementation noise terms decay as $\Theta((t+1)^{-2\sigma})$, the residual sums are of order $\sum_t (\alpha_t^{\mathsf{h}})^2$. Selecting the critical coupling $\nu = \frac{2}{3}\sigma$ yields the stated averaged suboptimality rates $\mathcal{O}\left((1-\gamma)^{-1}T^{-(1-\sigma)}\right)$, with the usual logarithmic refinements at the regime boundaries.

**Step 1: Decomposing the one-step improvement** According to the definition of Actor in Eq (4) and proof of Appendix B.2, we use the score feature in the RKHS $\mathcal{H}_K$ as

$$g(h)(\mathbf{s}, \mathbf{a}) := \nabla_h \log \pi_h(\mathbf{a} \mid \mathbf{s}) = K(\mathbf{s}, \cdot)\Sigma^{-1}(\mathbf{a} - h(\mathbf{s})) \in \mathcal{H}_K.$$

With compatible approximation proved in Appendix B.3, we posit

$$A^{\pi_{h_t}}(\mathbf{s}, \mathbf{a}) = \left\langle w_{\mathsf{A},t}, g(h_t)(\mathbf{s}, \mathbf{a})\right\rangle_{\mathcal{H}_K}, \qquad w_{\mathsf{A},t} \in \mathcal{H}_K. \tag{51}$$

By the performance-difference lemma introduced in Lemma C.10, for any two policies,

$$J(\pi^\star) - J(\pi_{h_t}) = \frac{1}{1-\gamma} \mathbb{E}_{(\mathbf{s},\mathbf{a})\sim\nu_{\pi^\star}} \left[ A^{\pi_{h_t}}(\mathbf{s},\mathbf{a}) \right].$$

Substituting the compatible form of (51) yields

$$J(\pi^\star) - J(\pi_{h_t}) = \frac{1}{1-\gamma} \mathbb{E}_{\nu_{\pi^\star}} \left[ \langle w_{\mathsf{A},t}, g(h_t) \rangle_{\mathcal{H}_K} \right] + \frac{1}{1-\gamma} \zeta_{\mathsf{a}}. \tag{52}$$

Define the divergence functional as

$$D(h) \coloneqq \mathbb{E}_{\nu_{\pi^\star}} \left[ \log \pi_h(\mathbf{a} \mid \mathbf{s}) \right].$$

Recall $\log \pi_h(\mathbf{a} \mid \mathbf{s})$ is $L_\psi$-smooth in $h$ in Lemma C.7 and $\nabla_h \log \pi_{h_t}(\mathbf{a} \mid \mathbf{s}) = g(h_t)$ in Proposition B.2. Then for any $h_{t+1}$, there is

$$
\begin{aligned}
& D(h_t) - D(h_{t+1}) \\
=& \mathbb{E}_{\nu_{\pi^\star}} \left[ \log \pi_{h_t}(\mathbf{a} \mid \mathbf{s}) - \log \pi_{h_{t+1}}(\mathbf{a} \mid \mathbf{s}) \right] \\
\geq& \left\langle \mathbb{E}_{\nu_{\pi^\star}} \left[ \nabla_h \log \pi_{h_t}(\mathbf{a} \mid \mathbf{s}) \right], h_{t+1} - h_t \right\rangle_{\mathcal{H}_K} - \frac{L_{\mathrm{grad}}}{2} \| h_{t+1} - h_t \|_{\mathcal{H}_K}^2 - 2 M_a^2 M_\Sigma^2 \delta_{\mathsf{K},h_t} \\
=& \left\langle \mathbb{E}_{\nu_{\pi^\star}} \left[ g(h_t) \right], h_{t+1} - h_t \right\rangle_{\mathcal{H}_K} - \frac{L_{\mathrm{grad}}}{2} \| h_{t+1} - h_t \|_{\mathcal{H}_K}^2 - 2 M_a^2 M_\Sigma^2 \delta_{\mathsf{K},h_t}. 
\end{aligned}
\tag{53}
$$

The additional term $\delta_{\mathsf{K},h_t}$ above accounts for the kernel drift/mismatch incurred by Critic-induced feature changes (cf. Lemma C.6); $M_A$ and $M_\Sigma$ are uniform bounds on the compatible weight and $\|\Sigma^{-1}\|_2$ that make the reduction explicit.

Let $\overline{h}_{t+1}$ denote the unprojected mirror step and $h_{t+1}$ the *implemented* update after the sparsification process. We have the update step takes the form

$$\overline{h}_{t+1} - h_t = \alpha_t^{\mathsf{h}} \hat{g}(h_t), \qquad \text{with a score estimate } \hat{g}(h_t) \in \mathcal{H}_K, \tag{54}$$

and decompose

$$h_{t+1} - h_t = \left( h_{t+1} - \overline{h}_{t+1} \right) + \left( \overline{h}_{t+1} - h_t \right) = \left( h_{t+1} - \overline{h}_{t+1} \right) + \alpha_t^{\mathsf{h}} \hat{g}(h_t).$$

Plugging this decomposition into (53) gives

$$
\begin{aligned}
D(h_t) - D(h_{t+1}) =& \left\langle \mathbb{E}_{\nu_{\pi^\star}} \left[ g(h_t) \right], h_{t+1} - \overline{h}_{t+1} \right\rangle_{\mathcal{H}_K} + \alpha_t^{\mathsf{h}} \left\langle \mathbb{E}_{\nu_{\pi^\star}} \left[ g(h_t) \right], \hat{g}(h_t) \right\rangle_{\mathcal{H}_K} \\
& - \frac{L_\psi}{2} \| h_{t+1} - h_t \|_{\mathcal{H}_K}^2 - 2 M_A^2 M_\Sigma^2 \delta_{\mathsf{K},h}^2.
\end{aligned}
\tag{55}
$$

Expand the correlation with the stochastic score:

$$
\begin{aligned}
\left\langle \mathbb{E}_{\nu_{\pi^\star}} \left[ g(h_t) \right], \hat{g}(h_t) \right\rangle_{\mathcal{H}_K} =& \left\langle \mathbb{E}_{\nu_{\pi^\star}} \left[ g(h_t) \right], g(h_t) \right\rangle_{\mathcal{H}_K} + \left\langle \mathbb{E}_{\nu_{\pi^\star}} \left[ g(h_t) \right], \hat{g}(h_t) - g(h_t) \right\rangle_{\mathcal{H}_K} \\
=& \| \mathbb{E}_{\nu_{\pi^\star}} \left[ g(h_t) \right] \|_{\mathcal{H}_K}^2 + \mathbb{E}_{\nu_{\pi^\star}} \left[ \langle g(h_t), \hat{g}(h_t) - g(h_t) \rangle_{\mathcal{H}_K} \right],
\end{aligned}
\tag{56}
$$

where we used linearity of expectation and inner product.

By (52), we have

$$\mathbb{E}_{\nu_{\pi^\star}} \left[ \langle w_{\mathsf{A},t}, g(h_t) \rangle_{\mathcal{H}_K} \right] = (1-\gamma) \left( J(\pi^\star) - J(\pi_{h_t}) \right).$$

Add and subtract $w_{\mathsf{A},t}$ inside the inner product of (56) to get

$$\langle g, \hat{g} - g \rangle = \langle w_{\mathsf{A},t}, \hat{g} \rangle - \langle w_{\mathsf{A},t}, g \rangle + \langle g - w_{\mathsf{A},t}, \hat{g} - g \rangle.$$

Take $\mathbb{E}_{\nu_{\pi^\star}}[\cdot]$ on both sides, and use the compatibility identity to obtain

$$\langle \mathbb{E}_{\nu_{\pi^\star}}[g], \hat{g} \rangle = \mathbb{E}_{\nu_{\pi^\star}} \left[ \|g\|^2 \right] + \mathbb{E}_{\nu_{\pi^\star}} \left[ \langle w_{\mathsf{A},t}, \hat{g} \rangle \right] - (1-\gamma) \left( J(\pi^\star) - J(\pi_{h_t}) \right) + \mathbb{E}_{\nu_{\pi^\star}} \left[ \langle g - w_{\mathsf{A},t}, \hat{g} - g \rangle \right]. \tag{57}$$

For brevity, we dropped the explicit $(h_t)$ argument and the subscript $\mathcal{H}_K$ on inner products.

Substituting (57) into (55) and rearranging to isolate the performance gap on the left yields

$$
\alpha_t^{\mathsf{h}}(1-\gamma)\left(J(\pi^\star)-J(\pi_{h_t})\right) \leq D(h_t) - D(h_{t+1}) + 2M_A^2 M_\Sigma^2 \delta_{\mathsf{K},h_t}^2 + \frac{L_\psi}{2}\|h_{t+1}-h_t\|_{\mathcal{H}_K}^2
$$

$$
+ \alpha_t^{\mathsf{h}}\mathbb{E}_{\nu_{\pi^\star}}\left[\langle w_{\mathsf{A},t}, \hat{g}(h_t)\rangle\right] - \left\langle \mathbb{E}_{\nu_{\pi^\star}}[g(h_t)], h_{t+1}-\overline{h}_{t+1}\right\rangle + \alpha_t^{\mathsf{h}}\zeta_{\mathsf{a}}
$$

$$
- \alpha_t^{\mathsf{h}}\|\mathbb{E}_{\nu_{\pi^\star}}[g(h_t)]\|_{\mathcal{H}_K}^2 - \alpha_t^{\mathsf{h}}\mathbb{E}_{\nu_{\pi^\star}}\left[\langle g(h_t)-w_{\mathsf{A},t}, g(h_t)-\hat{g}(h_t)\rangle\right]. \tag{58}
$$

Consequently, (58) is the desired one-step decomposition: it expresses the scaled performance gap $\alpha_t^{\mathsf{h}}(1-\gamma)\left(J(\pi^\star)-J(\pi_{h_t})\right)$ in terms of (i) the descent of $D(h_t)$, (ii) a kernel-drift term $\delta_{\mathsf{K},h+t}^2$, (iii) a smoothness penalty, (iv) a bias term involving $\mathbb{E}\langle w_{\mathsf{A},t}, \hat{g}(h_t)\rangle$, (v) the deviation between the ideal and implemented Actor steps $\langle \mathbb{E}[g(h_t)], h_{t+1}-\overline{h}_{t+1}\rangle$, and (vi) two variance-like corrections coming from $\|g(h_t)\|^2$ and the error-variance coupling $\langle g(h_t)-w_{\mathsf{A},t}, g(h_t)-\hat{g}(h_t)\rangle$. These residuals will be upper bounded in Step 2.

**Step 2: Bounding the residual terms.** We proceed to upper bound each term in (58). Throughout, recall $g(h_t)(\mathbf{s},\mathbf{a})=K(\mathbf{s},\cdot)\Sigma^{-1}(\mathbf{a}-h_t(\mathbf{s}))$ and define

$$
\overline{K}_{\pi^\star} := \mathbb{E}_{\mathbf{s}\sim d^{\pi^\star}}[K(\mathbf{s},\mathbf{s})], \qquad M_K := \sup_{\mathbf{s}} K(\mathbf{s},\mathbf{s}), \qquad M_\Sigma := \|\Sigma^{-1}\|_2.
$$

**Bounding kernel drift $\delta_{\mathsf{K},h_t}$.** By Lemma C.6, there exists an absolute constant so that

$$
\delta_{\mathsf{K},h_t} \leq 16M_{\Sigma_K}^2 M_S^4 q_{\mathsf{A}}^2 M_a^2 C_\mu^2 \|w_{\mathsf{V},t+1}-w_{\mathsf{V},t}\|_{\mathcal{H}_k}^2. \tag{59}
$$

Insert the intermediate optima $\{w_{\mathsf{V},t}^\star\}$ and apply the triangle inequality and parallelogram identity in $\mathcal{H}_k$:

$$
\|w_{\mathsf{V},t+1}-w_{\mathsf{V},t}\|_{\mathcal{H}_k}^2 = \|(w_{\mathsf{V},t+1}-w_{\mathsf{V},t+1}^\star)+(w_{\mathsf{V},t+1}^\star-w_{\mathsf{V},t}^\star)+(w_{\mathsf{V},t}^\star-w_{\mathsf{V},t})\|_{\mathcal{H}_k}^2
$$

$$
\leq 3\|w_{\mathsf{V},t+1}-w_{\mathsf{V},t+1}^\star\|_{\mathcal{H}_k}^2 + 3\|w_{\mathsf{V},t+1}^\star-w_{\mathsf{V},t}^\star\|_{\mathcal{H}_k}^2 + 3\|w_{\mathsf{V},t}^\star-w_{\mathsf{V},t}\|_{\mathcal{H}_k}^2, \tag{60}
$$

where the first term corresponds the Critic error at $t+1$, the second term is shift of optimum, and the last term represents the Critic error at $t$. Taking expectations and invoking Lemma C.11 together with the sensitivity bound (161) for the shifted optimum yields, for constants $L_\mathsf{V}, G_h, C, C_{\text{crit}}, M_k, C_b$,

$$
\mathbb{E}\left[\|w_{\mathsf{V},t+1}-w_{\mathsf{V},t}\|_{\mathcal{H}_\psi}^2\right] \leq \begin{cases} 3L_\mathsf{V}^2 G_h^2(\alpha_t^{\mathsf{h}})^2 + \dfrac{6C}{(t+1)^\nu} + 6M_k q_\mathsf{V}^2 \varepsilon_{\mathsf{PV}}^2 + 6\dfrac{C_b}{n_{\text{eff}}}, & \sigma > \frac{3}{2}\nu, \\[3mm] 3L_\mathsf{V}^2 G_h^2(\alpha_t^{\mathsf{h}})^2 + \dfrac{6C_{\text{crit}}\log^2(t+1)}{(t+1)^\nu} + 6M_k q_\mathsf{V}^2 \varepsilon_{\mathsf{PV}}^2 + \dfrac{6C_b}{n_{\text{eff}}}, & \sigma = \frac{3}{2}\nu, \\[3mm] 3L_\mathsf{V}^2 G_h^2(\alpha_t^{\mathsf{h}})^2 + \dfrac{6C}{(t+1)^{2(\sigma-\nu)}} + 6M_k q_\mathsf{V}^2 \varepsilon_{\mathsf{PV}}^2 + \dfrac{6C_b}{n_{\text{eff}}}, & \nu < \sigma < \frac{3}{2}\nu. \end{cases} \tag{61}
$$

Combining (59)-(61) and letting $C_\mathsf{K} := 48M_\Sigma^2 M_S^4 q^2 M_A^2 C_\mu^2$, we obtain

$$
\mathbb{E}[\delta_{\mathsf{K},h_t}] \leq C_\mathsf{K} \times \begin{cases} L_\mathsf{V}^2 G_h^2(\alpha_t^{\mathsf{h}})^2 + \dfrac{2C}{(t+1)^\nu} + 2M_k q_\mathsf{V}^2 \varepsilon_{\mathsf{PV}}^2 + \dfrac{2C_b}{n_{\text{eff}}}, & \sigma > \frac{3}{2}\nu, \\[3mm] L_\mathsf{V}^2 G_h^2(\alpha_t^{\mathsf{h}})^2 + \dfrac{2C_{\text{crit}}\log^2(t+1)}{(t+1)^\nu} + 2M_k q_\mathsf{V}^2 \varepsilon_{\mathsf{PV}}^2 + \dfrac{2C_b}{n_{\text{eff}}}, & \sigma = \frac{3}{2}\nu, \\[3mm] L_\mathsf{V}^2 G_h^2(\alpha_t^{\mathsf{h}})^2 + \dfrac{2C}{(t+1)^{2(\sigma-\nu)}} + 2M_k q_\mathsf{V}^2 \varepsilon_{\mathsf{PV}}^2 + \dfrac{2C_b}{n_{\text{eff}}}, & \nu < \sigma < \frac{3}{2}\nu. \end{cases} \tag{62}
$$

**Bounding smoothness penalty** $\frac{L_{\text{grad}}}{2}\|h_{t+1}-h_t\|_{\mathcal{H}_K}^2$. By $h_{t+1}-h_t = (h_{t+1}-\overline{h}_{t+1})+\alpha_t^{\mathsf{h}}\hat{g}(h_t)$ and $\|u+v\|^2 \leq 2\|u\|^2+2\|v\|^2$,

$$
\frac{L_{\text{grad}}}{2}\|h_{t+1}-h_t\|_{\mathcal{H}_K}^2 \leq L_{\text{grad}}\|h_{t+1}-\overline{h}_{t+1}\|_{\mathcal{H}_K}^2 + L_{\text{grad}}\left(\alpha_t^{\mathsf{h}}\right)^2 \|\hat{g}(h_t)\|_{\mathcal{H}_K}^2. \tag{63}
$$

For the second factor, write $\hat{g}(h_t) = g(h_t) + (\hat{g}(h_t) - g(h_t))$ and use $\|x + y\|^2 \leq 2\|x\|^2 + 2\|y\|^2$:

$$\mathbb{E}_{\nu_{\pi^\star}}\left[\|\hat{g}(h_t)\|^2_{\mathcal{H}_K}\right] \leq 2\mathbb{E}_{\nu_{\pi^\star}}\left[\|g(h_t)\|^2_{\mathcal{H}_K}\right] + 2\mathbb{E}_{\nu_{\pi^\star}}\left[\|g(h_t) - \hat{g}(h_t)\|^2_{\mathcal{H}_K}\right]. \tag{64}$$

By direct calculation, there is $\mathbb{E}_{\nu_{\pi^\star}}[\|g(h_t)\|^2_{\mathcal{H}_K}] = \overline{K}_{\pi^\star}\mathrm{tr}(\Sigma^{-1})$. We assume the score estimator has bounded second moment as

$$\mathbb{E}_{\nu_{\pi^\star}}\left[\|g(h_t) - \hat{g}(h_t)\|^2_{\mathcal{H}_K}\right] \leq \frac{C_b}{n_{\mathrm{eff}}}. \tag{65}$$

Taking expectations in (63) and applying (64) and (65) gives

$$\mathbb{E}\left[\frac{L_{\mathrm{grad}}}{2}\|h_{t+1} - h_t\|^2_{\mathcal{H}_K}\right] \leq L_{\mathrm{grad}}\mathbb{E}\left[\|h_{t+1} - \overline{h}_{t+1}\|^2_{\mathcal{H}_K}\right] + 2L_{\mathrm{grad}}\left(\alpha_t^{\mathsf{h}}\right)^2\left(\overline{K}_{\pi^\star}\mathrm{tr}(\Sigma^{-1}) + \frac{C_b}{n_{\mathrm{eff}}}\right). \tag{66}$$

If the sparsification with budget $q$ and precision $\varepsilon_{\mathsf{PA}}$ ensures

$$\mathbb{E}\left[\|h_{t+1} - \overline{h}_{t+1}\|^2_{\mathcal{H}_K}\right] \leq M_K q_{\mathsf{A}}^2 \varepsilon_{\mathsf{PA}}^2 + \frac{C_b}{n_{\mathrm{eff}}}, \tag{67}$$

then

$$\mathbb{E}\left[\frac{L_{\mathrm{grad}}}{2}\|h_{t+1} - h_t\|^2_{\mathcal{H}_K}\right] \leq L_{\mathrm{grad}}\left(M_K q_{\mathsf{A}}^2 \varepsilon_{\mathsf{PA}}^2 + \frac{C_b}{n_{\mathrm{eff}}}\right) + 2L_{\mathrm{grad}}\left(\alpha_t^{\mathsf{h}}\right)^2\left(\overline{K}_{\pi^\star}\mathrm{tr}(\Sigma^{-1}) + \frac{C_b}{n_{\mathrm{eff}}}\right). \tag{68}$$

**Bounding gradient energy** $\|g(h_t)\|^2_{\mathcal{H}_K}$. By the reproducing property,

$$\|g(h_t)\|^2_{\mathcal{H}_K} = \left\langle K(\mathbf{s}, \cdot)\Sigma^{-1}\left(\mathbf{a} - h_t(\mathbf{s})\right), K(\mathbf{s}, \cdot)\Sigma^{-1}\left(\mathbf{a} - h_t(\mathbf{s})\right)\right\rangle_{\mathcal{H}_K} = K(\mathbf{s}, \mathbf{s})\left\|\Sigma^{-1}\left(\mathbf{a} - h_t(\mathbf{s})\right)\right\|^2_2. \tag{69}$$

Pointwise, for $\|\mathbf{a}\|_2 \leq M_a$ and $\|h_t(\mathbf{s})\|_2 \leq C_h$ a.s.,

$$\|g(h_t)\|^2_{\mathcal{H}_K} \leq M_K M_\Sigma^2 (M_a + C_h)^2. \tag{70}$$

Conditioned on $\mathbf{s}$ with $\mathbf{a} \mid \mathbf{s} \sim \mathcal{N}(h_t(\mathbf{s}), \Sigma)$,

$$\mathbb{E}\left[\|g(h_t)\|^2_{\mathcal{H}_K} \mid \mathbf{s}\right] = K(\mathbf{s}, \mathbf{s})\mathrm{tr}(\Sigma^{-1}), \qquad \Rightarrow \qquad \mathbb{E}_{\nu_{\pi^\star}}\left[\|g(h_t)\|^2_{\mathcal{H}_K}\right] = \overline{K}_{\pi^\star}\mathrm{tr}(\Sigma^{-1}). \tag{71}$$

If $m_K := \mathrm{essinf}_{\mathbf{s}} K(\mathbf{s}, \mathbf{s}) > 0$, then the negative term in (58) admits the uniform lower bound

$$-\alpha_t^{\mathsf{h}}\mathbb{E}_{\nu_{\pi^\star}}\left[\|g(h_t)\|^2_{\mathcal{H}_K}\right] \leq -\alpha_t^{\mathsf{h}} m_K \mathrm{tr}(\Sigma^{-1}) \leq -\alpha_t^{\mathsf{h}} m_K \frac{d}{\lambda_{\max}(\Sigma)}. \tag{72}$$

**Bounding bias term** $\mathbb{E}_{\nu_{\pi^\star}}\langle w_{\mathsf{A},t}, \hat{g}(h_t)\rangle_{\mathcal{H}_K}$. Using $\hat{g}(h_t) = g(h_t) + (\hat{g}(h_t) - g(h_t))$ and the compatibility identity $\mathbb{E}_{\nu_{\pi^\star}}\langle w_{\mathsf{A},t}, g(h_t)\rangle = (1 - \gamma)\left(J(\pi^\star) - J(\pi_{h_t})\right)$,

$$\mathbb{E}_{\nu_{\pi^\star}}\left[\langle w_{\mathsf{A},t}, \hat{g}(h_t)\rangle\right] = (1 - \gamma)\left(J(\pi^\star) - J(\pi_{h_t})\right) + \mathbb{E}_{\nu_{\pi^\star}}\left[\langle w_{\mathsf{A},t}, \hat{g}(h_t) - g(h_t)\rangle\right]. \tag{73}$$

By Cauchy-Schwarz inequality and assuming $\|w_{\mathsf{A},t}\|_{\mathcal{H}_K} \leq M_{Adv}$, armed with (65), there is

$$\mathbb{E}_{\nu_{\pi^\star}}\left[\langle w_{\mathsf{A},t}, \hat{g}(h_t)\rangle\right] \leq (1 - \gamma)\left(J(\pi^\star) - J(\pi_{h_t})\right) + M_{Adv}\sqrt{\frac{C_b}{n_{\mathrm{eff}}}}$$

$$\leq (1 - \gamma)\left(J(\pi^\star) - J(\pi_{h_t})\right) + \frac{\eta}{2}M_{Adv}^2 + \frac{1}{2\eta}\frac{C_b}{n_{\mathrm{eff}}}, \quad \forall \eta > 0. \tag{74}$$

**Bounding deviation term.** Let $D_t := \langle \mathbb{E}_{\nu_{\pi^\star}}[g(h_t)], h_{t+1} - \overline{h}_{t+1}\rangle_{\mathcal{H}_K}$. By Cauchy-Schwarz and Jensen inequalitis, we derive

$$\mathbb{E}[D_t] \geq -\left\|\mathbb{E}_{\nu_{\pi^\star}}[g(h_t)]\right\|_{\mathcal{H}_K}\sqrt{\mathbb{E}\left[\|h_{t+1} - \overline{h}_{t+1}\|^2_{\mathcal{H}_K}\right]}$$

$$\geq -\sqrt{\mathbb{E}_{\nu_{\pi^\star}}\|g(h_t)\|^2_{\mathcal{H}_K}}\sqrt{\mathbb{E}\left[\|h_{t+1} - \overline{h}_{t+1}\|^2_{\mathcal{H}_K}\right]}$$

$$= -\sqrt{\overline{K}_{\pi^\star}\mathrm{tr}(\Sigma^{-1})}\sqrt{\mathbb{E}\left[\|h_{t+1} - \overline{h}_{t+1}\|^2_{\mathcal{H}_K}\right]}. \tag{75}$$

Using (67), this becomes

$$\mathbb{E}[D_t] \geq -\sqrt{\overline{K}_{\pi^\star}\mathrm{tr}(\Sigma^{-1})}\sqrt{M_K q_{\mathsf{V}}^2 \varepsilon_{\mathsf{PV}}^2 + \frac{C_b}{n_{\mathrm{eff}}}}. \tag{76}$$

**Bounding error-variance coupling.** Define $T_t := \mathbb{E}_{\nu_{\pi^\star}}\big[\langle g(h_t) - w_{\mathsf{A},t}, g(h_t) - \hat{g}(h_t)\rangle_{\mathcal{H}_K}\big]$. For any $\eta > 0$, Young's inequality gives

$$T_t \geq -\frac{\eta}{2}\mathbb{E}_{\nu_{\pi^\star}}\big[\|g(h_t) - w_{\mathsf{A},t}\|^2_{\mathcal{H}_K}\big] - \frac{1}{2\eta}\mathbb{E}_{\nu_{\pi^\star}}\big[\|g(h_t) - \hat{g}(h_t)\|^2_{\mathcal{H}_K}\big]. \tag{77}$$

Expanding the first expectation and using $\mathbb{E}_{\nu_{\pi^\star}}\langle w_{\mathsf{A},t}, g(h_t)\rangle = (1 - \gamma)\,(J(\pi^\star) - J(\pi_{h_t})) \geq 0$ and (71), we derive

$$\begin{aligned}
\mathbb{E}_{\nu_{\pi^\star}}\big[\|g(h_t) - w_{\mathsf{A},t}\|^2_{\mathcal{H}_K}\big] &= \mathbb{E}_{\nu_{\pi^\star}}\big[\|g(h_t)\|^2_{\mathcal{H}_K}\big] + \mathbb{E}\big[\|w_{\mathsf{A},t}\|^2_{\mathcal{H}_K}\big] - 2\mathbb{E}_{\nu_{\pi^\star}}\big[\langle w_{\mathsf{A},t}, g(h_t)\rangle\big] \\
&\leq \overline{K}_{\pi^\star}\mathrm{tr}(\Sigma^{-1}) + M^2_{Adv}.
\end{aligned} \tag{78}$$

Together with Eqs. (65) and (77) yields the tunable bound

$$T_t \geq -\frac{\eta}{2}\left(\overline{K}_{\pi^\star}\mathrm{tr}(\Sigma^{-1}) + M^2_{Adv}\right) - \frac{1}{2\eta}\frac{C_b}{n_{\mathrm{eff}}}, \qquad \forall \eta > 0, \tag{79}$$

whose optimal choice $\eta^\star = \sqrt{\dfrac{C_b}{n_{\mathrm{eff}}\left(\overline{K}_{\pi^\star}\mathrm{tr}(\Sigma^{-1}) + M^2_{Adv}\right)}}$ gives the compact form

$$T_t \geq -\sqrt{\left(\overline{K}_{\pi^\star}\mathrm{tr}(\Sigma^{-1}) + M^2_{Adv}\right)\frac{C_b}{n_{\mathrm{eff}}}}. \tag{80}$$

**Step 3: Convergence rates under polynomial stepsizes.** Starting from the one-step decomposition in (58) and the bounds assembled in Step 2, we now telescope in time and convert the per-iteration inequality into rates. Throughout this step we use the Young form of the bias bound (74), and we retain the good negative term $-\alpha^{\mathsf{h}}_t\mathbb{E}\big[\|g(h_t)\|^2_{\mathcal{H}_K}\big]$ to absorb the deviation inner product via a Young's inequality.

Collecting Eqs. (62), (68), (74), (76), and (80) into (58), and dropping the good negative term (72) for an upper bound, we obtain

$$\begin{aligned}
\alpha^{\mathsf{h}}_t(1-\gamma)\,(J(\pi^\star) - J(\pi_{h_t})) &\leq (D(h_t) - D(h_{t+1})) + 2M^2_a M^2_{\Sigma_K}\mathbb{E}[\delta_{\mathsf{K},h}] + L_{\mathrm{grad}}\left(M_K q^2_{\mathsf{V}}\varepsilon^2_{\mathsf{PV}} + \frac{C_b}{n_{\mathrm{eff}}}\right) \\
&\quad + 2L_{\mathrm{grad}}\left(\alpha^{\mathsf{h}}_t\right)^2\left(\overline{K}_{\pi^\star}\mathrm{tr}(\Sigma^{-1}) + \frac{C_b}{n_{\mathrm{eff}}}\right) + \alpha^{\mathsf{h}}_t\left[(1 - \gamma)\,(J(\pi^\star) - J(\pi_{h_t})) + M_{Adv}\sqrt{\frac{C_b}{n_{\mathrm{eff}}}}\right] \\
&\quad + \sqrt{\overline{K}_{\pi^\star}\mathrm{tr}(\Sigma^{-1})}\sqrt{M_K q^2_{\mathsf{V}}\varepsilon^2_{\mathsf{PV}} + \frac{C_b}{n_{\mathrm{eff}}}} + \alpha^{\mathsf{h}}_t\sqrt{\left(\overline{K}_{\pi^\star}\mathrm{tr}(\Sigma^{-1}) + M^2_{Adv}\right)\frac{C_b}{n_{\mathrm{eff}}}} + \alpha^{\mathsf{h}}_t\zeta_{\mathsf{a}}. \tag{81}
\end{aligned}$$

Inequality (81) will be telescoped and simplified under step-size choices in Step 3.

Summing (58) from $t = 0$ to $T - 1$ and applying the bounds Eqs. (62), (68), (74), (75), and (80), we obtain, for any $\eta_t > 0$

and any $\varepsilon_t \in (0, 1)$,

$$
\begin{aligned}
(1 - \gamma) \sum_{t=0}^{T-1} \alpha_t^{\mathsf{h}} \mathbb{E}\left[J(\pi^\star) - J(\pi_{h_t})\right] \leq & D(h_0) - D(h_T) + 2M_a^2 M_{\Sigma_K}^2 \sum_{t=0}^{T-1} \mathbb{E}\left[\delta_{\mathsf{K}, h_t}\right] + L_{\text{grad}} \sum_{t=0}^{T-1} \mathbb{E}\left[\|h_{t+1} - \overline{h}_{t+1}\|_{\mathcal{H}_K}^2\right] \\
& + 2L_{\text{grad}} \sum_{t=0}^{T-1} \left(\alpha_t^{\mathsf{h}}\right)^2 \left(\overline{K}_{\pi^\star} \text{tr}(\Sigma^{-1}) + \frac{C_b}{n_{\text{eff}, t}}\right) + \sum_{t=0}^{T-1} \alpha_t^{\mathsf{h}} \left(\frac{\eta_t}{2} M_{Adv}^2 + \frac{C_b}{2\eta_t n_{\text{eff}, t}}\right) \\
& + \sum_{t=0}^{T-1} \left[\frac{\varepsilon_t}{2} \alpha_t^{\mathsf{h}} \mathbb{E}\left[\|g(h_t)\|_{\mathcal{H}_K}^2\right] + \frac{1}{2\varepsilon_t \alpha_t^{\mathsf{h}}} \mathbb{E}\left[\|h_{t+1} - \overline{h}_{t+1}\|_{\mathcal{H}_K}^2\right]\right] \\
& + \sum_{t=0}^{T-1} \alpha_t^{\mathsf{h}} \sqrt{\left(\overline{K}_{\pi^\star} \text{tr}(\Sigma^{-1}) + M_{Adv}^2\right) \frac{C_b}{n_{\text{eff}, t}}} + \sum_{t=0}^{T-1} \alpha_t^{\mathsf{h}} \zeta_{\mathsf{a}} \\
\leq & B_D + 2M_A^2 M_{\Sigma_K}^2 \sum_{t=0}^{T-1} \mathbb{E}\left[\delta_{\mathsf{K}, h_t}\right] + L_{\text{grad}} \sum_{t=0}^{T-1} \mathbb{E}\left[\|h_{t+1} - \overline{h}_{t+1}\|_{\mathcal{H}_K}^2\right] \\
& + 2L_{\text{grad}} \sum_{t=0}^{T-1} \left(\alpha_t^{\mathsf{h}}\right)^2 \left(\overline{K}_{\pi^\star} \text{tr}(\Sigma^{-1}) + \frac{C_b}{n_{\text{eff}, t}}\right) + \sum_{t=0}^{T-1} \alpha_t^{\mathsf{h}} \left(\frac{\eta_t}{2} M_{Adv}^2 + \frac{C_b}{2\eta_t n_{\text{eff}, t}}\right) \\
& + \sum_{t=0}^{T-1} \left[\frac{\varepsilon_t}{2} \alpha_t^{\mathsf{h}} \mathbb{E}\left[\|g(h_t)\|_{\mathcal{H}_K}^2\right] + \frac{1}{2\varepsilon_t \alpha_t^{\mathsf{h}}} \mathbb{E}\left[\|h_{t+1} - \overline{h}_{t+1}\|_{\mathcal{H}_K}^2\right]\right] \\
& + \sum_{t=0}^{T-1} \alpha_t^{\mathsf{h}} \sqrt{\left(\overline{K}_{\pi^\star} \text{tr}(\Sigma^{-1}) + M_{Adv}^2\right) \frac{C_b}{n_{\text{eff}, t}}} + \sum_{t=0}^{T-1} \alpha_t^{\mathsf{h}} \zeta_{\mathsf{a}}.
\end{aligned}
\tag{82}
$$

The negative term $-\alpha_t^{\mathsf{h}} \mathbb{E}\left[\|g(h_t)\|_{\mathcal{H}_K}^2\right]$ in (58) can be dropped. With this trick, the deviation contribution scales as $\frac{1}{\alpha_t^{\mathsf{h}}} \mathbb{E}\|h_{t+1} - \overline{h}_{t+1}\|^2$.

To make all residuals summable, we adopt the standard, square-summable schedules:

$$
\alpha_t^{\mathsf{h}} = \alpha_0 (t+1)^{-\sigma}, \ \beta_t = \beta_0 (t+1)^{-\nu}, \ \frac{C_b}{n_{\text{eff}, t}} = \Theta\left((t+1)^{-2\sigma}\right), \ \mathbb{E}\left[\|h_{t+1} - \overline{h}_{t+1}\|_{\mathcal{H}_K}^2\right] = \Theta\left((t+1)^{-2\sigma}\right)
\tag{83}
$$

with $\sigma \in (0, 1)$ and $\nu \in (0, 1)$. The last two relations are achieved, e.g., by using mini-batch sizes and sparsification budgets that grow like $(t+1)^{2\sigma}$. Under (83), the two implementation-induced sums in (82) are $\mathcal{O}\left(\sum_t (\alpha_t^{\mathsf{h}})^2\right)$.

Step 2 gives a bound on $\mathbb{E}[\delta_{\mathsf{K}, h_t}]$, and we have

$$
\mathbb{E}\left[\delta_{\mathsf{K}, h_t}\right] = \mathcal{O}\left(\left(L_{\mathsf{V}} G_h \alpha_t^{\mathsf{h}}\right)^2 + \rho_t + q_{\mathsf{A}, t}^2 \varepsilon_{\mathsf{PA}, t}^2 + \frac{1}{n_{\text{eff}, t}}\right),
\tag{84}
$$

where

$$
\rho_t := \begin{cases}
(t+1)^{-\nu}, & \sigma > \frac{3}{2}\nu, \\
(t+1)^{-\nu} \log^2(t+1), & \sigma = \frac{3}{2}\nu, \\
(t+1)^{-2(\sigma - \nu)}, & \nu < \sigma < \frac{3}{2}\nu.
\end{cases}
$$

Consequently,

$$
\sum_{t=0}^{T-1} \mathbb{E}\left[\delta_{\mathsf{K}, h_t}\right] = \mathcal{O}\left(\sum_{t=0}^{T-1} (\alpha_t^{\mathsf{h}})^2 + \sum_{t=0}^{T-1} \rho_t + \sum_{t=0}^{T-1} q_{\mathsf{A}, t}^2 \varepsilon_{\mathsf{PA}, t}^2 + \sum_{t=0}^{T-1} \frac{1}{n_{\text{eff}, t}}\right).
\tag{85}
$$

Under (83) and with $q_t^2 \varepsilon_{\mathsf{PV}, t}^2 = \Theta((t+1)^{-\sigma})$, both the third and fourth sums are $\mathcal{O}\left(\sum_t (\alpha_t^{\mathsf{h}})^2\right)$.

For $p \in (0, 1)$, $\sum_{t=0}^{T-1} (t+1)^{-p} = \Theta\left(T^{1-p}\right)$; for $p = 1$, it is $\Theta(\log T)$; for $p > 1$, it is $\Theta(1)$. Therefore, we have

$$
S_\alpha(T) := \sum_{t=0}^{T-1} \alpha_t^{\mathsf{h}} = \Theta\left(T^{1-\sigma}\right), \qquad S_{\alpha^2}(T) := \sum_{t=0}^{T-1} (\alpha_t^{\mathsf{h}})^2 = \begin{cases}
\Theta(1), & \sigma > \frac{1}{2}, \\
\Theta(\log T), & \sigma = \frac{1}{2}, \\
\Theta\left(T^{1-2\sigma}\right), & 0 < \sigma < \frac{1}{2}.
\end{cases}
$$

Moreover, there is

$$\sum_{t=0}^{T-1} \rho_t = \begin{cases} \Theta\left(T^{1-\nu}\right), & \sigma > \frac{3}{2}\nu,\ \nu < 1, \\ \Theta(\log T), & \sigma > \frac{3}{2}\nu,\ \nu = 1, \\ \Theta(1), & \sigma > \frac{3}{2}\nu,\ \nu > 1, \\ \Theta\left(\log^2 T \cdot T^{1-\nu}\right), & \sigma = \frac{3}{2}\nu,\ \nu < 1, \\ \Theta\left(T^{1-2(\sigma-\nu)}\right), & \nu < \sigma < \frac{3}{2}\nu,\ 2(\sigma-\nu) < 1, \\ \Theta(\log T), & \nu < \sigma < \frac{3}{2}\nu,\ 2(\sigma-\nu) = 1, \\ \Theta(1), & \nu < \sigma < \frac{3}{2}\nu,\ 2(\sigma-\nu) > 1. \end{cases}$$

Define a random index $\tilde{t} \in \{0, \dots, T-1\}$ drawn with $\Pr(\tilde{t} = t) = \alpha_t^{\mathsf{h}}/S_\alpha(T)$. Then

$$\mathbb{E}\left[J(\pi^\star) - J(\pi_{h_{\tilde{t}}})\right] = \frac{\sum_{t=0}^{T-1} \alpha_t^{\mathsf{h}} \mathbb{E}\left[J(\pi^\star) - J(\pi_{h_t})\right]}{S_\alpha(T)}.$$

Dividing (82) by $(1-\gamma)S_\alpha(T)$ and using the estimates from results above yields

$$\mathbb{E}\left[J(\pi^\star) - J(\pi_{h_{\tilde{t}}})\right]$$

$$\leq \frac{B_D}{(1-\gamma)S_\alpha(T)} + \frac{C_1}{1-\gamma} \cdot \frac{S_{\alpha^2}(T)}{S_\alpha(T)} + \frac{C_2}{1-\gamma} \cdot \frac{\sum_{t=0}^{T-1} \rho_t}{S_\alpha(T)} + \frac{C_3}{1-\gamma} \cdot \frac{S_{\alpha^2}(T)}{S_\alpha(T)} + \frac{\zeta_{\mathsf{a}}}{1-\gamma}$$

$$+ \frac{C_4}{1-\gamma} \cdot \frac{\sum_{t=0}^{T-1} (\alpha_t^{\mathsf{h}}) \left(\frac{\eta_t}{2} M_{Adv}^2 + \frac{C_b}{2\eta_t n_{\mathrm{eff},t}} + \sqrt{(\overline{K}_{\pi^\star} \mathrm{tr}(\Sigma^{-1}) + M_{Adv}^2) \frac{C_b}{n_{\mathrm{eff},t}}}\right)}{S_\alpha(T)}. \tag{86}$$

Here $C_1, \dots, C_4$ absorb $M_K, M_\Sigma, M_{Adv}, L_{\mathrm{grad}}, L_V, G_h$ and the constants from Step 2. Choosing, e.g., $\eta_t \equiv \eta = \Theta(1)$ and the schedule introduced (83) so that $n_{\mathrm{eff},t}^{-1} = \Theta((t+1)^{-2\sigma})$, the last fraction is $\mathcal{O}\left(S_{\alpha^2}(T)/S_\alpha(T)\right)$.

Following the two-timescale design, pick

$$\nu = \tfrac{2}{3}\sigma \qquad (\text{i.e., } \sigma = \tfrac{3}{2}\nu),$$

so that the Critic is sufficiently fast to track the Actor. In this case (the Critical regime) we have $\rho_t = (t+1)^{-\nu} \log^2(t+1)$ and $\sum_{t=0}^{T-1} \rho_t = \tilde{\Theta}\left(T^{1-\nu}\right)$ (where $\tilde{\Theta}$ hides polylog factors).

Plugging these sums into (86) gives the final rates. Collecting the dominant terms and keeping the three classical regimes of $\sigma$ yields

$$\mathbb{E}\left[J(\pi^\star) - J(\pi_{h_{\tilde{t}}})\right] \leq \frac{\zeta_{\mathsf{a}}}{1-\gamma} + \begin{cases} \mathcal{O}\left(\frac{1}{(1-\gamma)} \cdot \frac{1}{T^{1-\sigma}}\right), & \sigma > \frac{3}{4}, \\ \mathcal{O}\left(\frac{1}{(1-\gamma)} \cdot \frac{\log^2 T}{T^{1/2}}\right), & \sigma = \frac{3}{4}\ (\Rightarrow \nu = \frac{1}{2}), \\ \mathcal{O}\left(\frac{1}{(1-\gamma)} \cdot \frac{\log T}{T^{1-\frac{2}{3}\sigma}}\right), & 0 < \sigma < \frac{3}{4}. \end{cases} \tag{87}$$

If the sparsification is exact ($h_{t+1} \equiv \overline{h}_{t+1}$) and fresh data are used so that $n_{\mathrm{eff},t}^{-1} = \mathcal{O}\left((t+1)^{-2\sigma}\right)$, then all implementation-driven terms fall into $S_{\alpha^2}(T)/S_\alpha(T)$ and are strictly dominated by the leading rates in (87). With fixed budgets (constant $q_t, \varepsilon_{\mathsf{PV},t}$ or $n_{\mathrm{eff},t}$), the corresponding residuals create a floor of order $\mathcal{O}\left((1-\gamma)^{-1}T^{-\sigma}\right)$ or constant; the schedule in (83) (or taking the ideal update) removes this floor and recovers (87).

### C.6. Proof of supporting facts

C.6.1. PROOF OF PROPOSITION B.2

According to the definition of $f$, we have

$$f(h) = \log \pi_{h,\Sigma}(\mathbf{a} \mid \mathbf{s}) = -\log((2\pi)^{\frac{m}{2}} (\det(\Sigma))^{\frac{1}{2}}) - \frac{1}{2}(\mathbf{a} - h(\mathbf{s}))^\top \Sigma^{-1}(\mathbf{a} - h(\mathbf{s}));$$

$$f(h+g) = \log \pi_{h+g,\Sigma}(\mathbf{a} \mid \mathbf{s}) = -\log((2\pi)^{\frac{m}{2}} (\det(\Sigma))^{\frac{1}{2}}) - \frac{1}{2}(\mathbf{a} - h(\mathbf{s}) - g(\mathbf{s}))^\top \Sigma^{-1}(\mathbf{a} - h(\mathbf{s}) - g(\mathbf{s})).$$

Therefore, we extend the Banach spaces $\mathcal{V}$ and $\mathcal{W}$ to Hilbert spaces to obtain

$$
\begin{aligned}
\lim_{g \to 0} \frac{\|f(h+g) - f(h) - Df|_h(g)\|_2}{\|g\|_{\mathcal{H}_K}} &= \lim_{g \to 0} \frac{\|g(\mathbf{s})^\top \Sigma^{-1} g(\mathbf{s})\|_2}{\|g\|_{\mathcal{H}_K}} \\
&= \lim_{g \to 0} \frac{\langle g, K(\mathbf{s}, \cdot) \Sigma^{-1} g(\mathbf{s}) \rangle_{\mathcal{H}_K}}{\|g\|_{\mathcal{H}_K}} \\
&\leq \lim_{g \to 0} \frac{\|g\|_{\mathcal{H}_K} \left(\Sigma^{-1} g(\mathbf{s})\right)^\top K(\mathbf{s}, \cdot) \Sigma^{-1} g(\mathbf{s})}{\|g\|_{\mathcal{H}_K}} \\
&= \lim_{g \to 0} \left(\Sigma^{-1} g(\mathbf{s})\right)^\top K(\mathbf{s}, \cdot) \Sigma^{-1} g(\mathbf{s}) \to 0,
\end{aligned}
$$

where the inequality comes from the Cauchy-Schwarz inequality.

### C.6.2. PROOF OF LEMMA C.2

Since the Shapley functional is a linear combination of bounded linear functionals (value functionals), it admits a Riesz representer in the RKHS. Therefore, given a value functional $v$ indexed by input $\mathbf{s}$ and coalition $\mathcal{C}$, the Shapley functional $\phi_{\mathbf{s},i} : \mathcal{H}_k \to \mathbb{R}$ such that $\phi_{\mathbf{s},i}(v)$ is the $i^{\text{th}}$ Shapley values on input $\mathbf{s}$, can be written as the following linear combination of value functionals

$$
\phi_{\mathbf{s},i} = \frac{1}{d} \sum_{\mathcal{C} \subseteq \mathcal{X} \setminus \{i\}} \binom{d-1}{|C|}^{-1} \left(v_{\mathcal{C} \cup \{i\}}(\mathbf{s}) - v_{\mathcal{C}}(\mathbf{s})\right). \tag{88}
$$

To prove that Shapley functionals between two observations $\mathbf{s}$ and $\widetilde{\mathbf{s}}$ are $\delta$ close when the two points are close, we proceed as the following three parts.

**Part 1: Bound the feature maps** We show the results of special case as the usual product RBF kernel, leading to bound the distance of the feature maps as a function of $\delta$.

When we pick $\mathbf{s}, \widetilde{\mathbf{s}} \in \mathbb{R}^d$ and with a product RBF kernel i.e., $k(\mathbf{s}, \widetilde{\mathbf{s}}) = \prod_{j=1}^d k^{(j)}(\mathbf{s}_j, \widetilde{\mathbf{s}}_j)$, where $k^{(j)}$ are RBF kernels. For simplicity, we assume they all share the same lengthscale $l$. Since

$$
\|\psi(\mathbf{s}) - \psi(\widetilde{\mathbf{s}})\|_{\mathcal{H}_k}^2 = k(\mathbf{s}, \mathbf{s}) + k(\widetilde{\mathbf{s}}, \widetilde{\mathbf{s}}) - 2k(\mathbf{s}, \widetilde{\mathbf{s}}),
$$

the first two terms ($k(\mathbf{s}, \mathbf{s})$ and $k(\widetilde{\mathbf{s}}, \widetilde{\mathbf{s}})$) are 1 and we can bound the last term according to the basic form of RBF kernel as follows.

$$
k(\mathbf{s}, \widetilde{\mathbf{s}}) = \exp\left(-\frac{\sum_{j=1}^d |\mathbf{s}_j - \widetilde{\mathbf{s}}_j|^2}{2l^2}\right) = \exp\left(-\frac{\|\mathbf{s} - \widetilde{\mathbf{s}}\|_2^2}{2l^2}\right) \geq \exp\left(-\frac{\varepsilon^2}{2l^2}\right), \tag{89}
$$

where $\varepsilon = \|\mathbf{s} - \widetilde{\mathbf{s}}\|_2$ and $k(\mathbf{s}, \mathbf{s}) = 1$ for RBF kernel. Then we can bound the difference in feature maps as follows,

$$
\|\psi(\mathbf{s}) - \psi(\widetilde{\mathbf{s}})\|_{\mathcal{H}_k}^2 \leq 2 - 2\exp\left(-\frac{\varepsilon^2}{2l^2}\right). \tag{90}
$$

Therefore, the distance in feature maps $\|\psi(\mathbf{s}) - \psi(\widetilde{\mathbf{s}})\|_{\mathcal{H}_k}$ can be expressed by the distance between $\mathbf{s}$ and $\widetilde{\mathbf{s}}$ in the RBF kernel. Different bounds can be derived for different kernels and we only show the special RBF case for illustration purposes.

**Part 2: Bound value functionals** We then upper bound the value functionals and show that this bound can be relaxed so that it is independent of the choice of coalition. We first proceed with the interventional case and move on to observational afterwards. Using the result of Propositions A.2 and A.4 by Chau et al. (2022), we have

- Off-manifold value functionals: For any coalition $\mathcal{C}$, define $T_{\mathcal{C}}^{(\text{off})} = \left\|\mu_{\mathcal{C}}^{(\text{off})}(\mathbf{s}) - \mu_{\mathcal{C}}^{(\text{off})}(\widetilde{\mathbf{s}})\right\|_{\mathcal{H}_k}^2$. Under the definition of $\mu_{\mathcal{C}}^{(\text{off})}(\mathbf{s})$ in (10a), there is $T_{\mathcal{C}}^{(\text{off})} \leq \left\|\psi(\mathbf{s}^{\mathcal{C}}) - \psi(\widetilde{\mathbf{s}}^{\mathcal{C}})\right\|_{\mathcal{H}_{k_{\mathcal{C}}}}^2 \left\|\mu_{\mathbf{S}^{\overline{c}}}\right\|_{\mathcal{H}_{k_{\overline{c}}}}^2$ with Cauchy-Schwarz inequality. Let $\delta_\psi :=$

$\sup_{\mathcal{C} \subseteq \mathcal{X}} \left\| \psi(\mathbf{s}^{\mathcal{C}}) - \psi(\widetilde{\mathbf{s}}^{\mathcal{C}}) \right\|_{\mathcal{H}_{k_{\mathcal{C}}}}^2$ and assume kernels are all bounded per dimension by $M$, i.e $k^{(j)}(\mathbf{s}, \widetilde{\mathbf{s}}) \leq M$ for all $j \in \{1, 2, \cdots, d\}$. Armed with $\sup_{\mathcal{C} \subseteq \mathcal{X}} M^{|\mathcal{C}|} = M^d$, then the bound can be further loosen up as

$$T_{\mathcal{C}}^{(\text{off})} = \left\| \mu_{\mathcal{C}}^{(\text{off})}(\mathbf{s}) - \mu_{\mathcal{C}}^{(\text{off})}(\widetilde{\mathbf{s}}) \right\|_{\mathcal{H}_k}^2 = \left\| \psi(\mathbf{s}^{\mathcal{C}}) \otimes \mu_{\mathbf{S}^{\overline{c}}} - \psi(\widetilde{\mathbf{s}}^{\mathcal{C}}) \otimes \mu_{\mathbf{S}^{\overline{c}}} \right\|_{\mathcal{H}_k}^2$$
$$= \left\| \psi(\mathbf{s}^{\mathcal{C}}) - \psi(\widetilde{\mathbf{s}}^{\mathcal{C}}) \right\|_{\mathcal{H}_{k_{\mathcal{C}}}}^2 \left\| \mu_{\mathbf{S}^{\overline{c}}} \right\|_{\mathcal{H}_{k_{\overline{c}}}}^2 \leq \delta_\psi \sup_{\mathcal{C} \subseteq \mathcal{X}} M^{|\mathcal{C}|} = \delta_\psi M^d, \qquad (91)$$

where $\left\| \mu_{\mathbf{S}^{\overline{c}}} \right\|_{\mathcal{H}_{k_{\overline{c}}}}^2 = \left\| \mathbb{E}[k(\mathbf{S}^{\overline{C}}, \widetilde{\mathbf{S}}^{\overline{C}})] \right\|^2 \leq M^{|\overline{c}|}$.

- On-manifold value functionals: For any coalition $\mathcal{C}$, define $T_{\mathcal{C}}^{(\text{on})} = \left\| \mu_{\mathcal{C}}^{(\text{on})}(\mathbf{s}) - \mu_{\mathcal{C}}^{(\text{on})}(\widetilde{\mathbf{s}}) \right\|_{\mathcal{H}_k}^2$. Under the definition of $\mu_{\mathcal{C}}^{(\text{on})}(\mathbf{s})$ in (10a), there is $T_{\mathcal{C}}^{(\text{on})} \leq \left\| \psi(\mathbf{s}^{\mathcal{C}}) - \psi(\widetilde{\mathbf{s}}^{\mathcal{C}}) \right\|_{\mathcal{H}_{k_{\mathcal{C}}}}^2 \left\| \mu_{\mathbf{S}^{\overline{c}}|\mathbf{S}^{\mathcal{C}}} \right\|_{\mathcal{H}_{\Gamma_{\mathbf{S}^{\mathcal{C}}}}}^2 \left( \left\| \psi(\mathbf{s}^{\mathcal{C}}) \right\|_{\mathcal{H}_{k_{\mathcal{C}}}}^2 + \left\| \psi(\widetilde{\mathbf{s}}^{\mathcal{C}}) \right\|_{\mathcal{H}_{k_{\mathcal{C}}}}^2 \right)$, where $\mathcal{H}_{\Gamma_{\mathbf{S}^{\mathcal{C}}}}$ is the $\mathcal{H}_{k_{\overline{c}}}$-valued RKHS. Let $\delta_\psi := \sup_{\mathcal{C} \subseteq \mathcal{X}} \left\| \psi(\mathbf{s}^{\mathcal{C}}) - \psi(\widetilde{\mathbf{s}}^{\mathcal{C}}) \right\|_{\mathcal{H}_{k_{\mathcal{C}}}}^2$ and $M_\Gamma = \sup_{\mathcal{C} \subseteq \mathcal{X}} \left\| \mu_{\mathbf{S}^{\overline{c}}|\mathbf{S}^{\mathcal{C}}} \right\|_{\mathcal{H}_{\Gamma_{\mathbf{S}^{\mathcal{C}}}}}^2$. Then $T_{\mathcal{C}}^{(\text{on})} \leq 2 M_\Gamma M^d \delta_\psi$ for all coalition $\mathcal{C}$.

**Part 3: Bound the Shapley functionals**   Finally, we bound the loss of the value functionals under the definition of (9) as

$$|v_{\mathcal{C}}(\mathbf{s}) - v_{\mathcal{C}}(\widetilde{\mathbf{s}})|^2 = |\langle w_{\mathsf{V}}, \mu_{\mathcal{C}}(\mathbf{s}) \rangle_{\mathcal{H}_k} - \langle \widetilde{w}_{\mathsf{V}}, \mu_{\mathcal{C}}(\widetilde{\mathbf{s}}) \rangle_{\mathcal{H}_k}|^2$$
$$\leq 2 |\langle w_{\mathsf{V}} - \widetilde{w}_{\mathsf{V}}, \mu_{\mathcal{C}}(\mathbf{s}) \rangle_{\mathcal{H}_k}|^2 + 2 |\langle \widetilde{w}_{\mathsf{V}}, \mu_{\mathcal{C}}(\mathbf{s}) - \mu_{\mathcal{C}}(\widetilde{\mathbf{s}}) \rangle_{\mathcal{H}_k}|^2$$
$$\leq 2 \left\| w_{\mathsf{V}} - \widetilde{w}_{\mathsf{V}} \right\|_{\mathcal{H}_k}^2 \left\| \mu_{\mathcal{C}}(\mathbf{s}) \right\|_{\mathcal{H}_k}^2 + 2 \left\| \widetilde{w}_{\mathsf{V}} \right\|_{\mathcal{H}_k}^2 \left\| \mu_{\mathcal{C}}(\mathbf{s}) - \mu_{\mathcal{C}}(\widetilde{\mathbf{s}}) \right\|_{\mathcal{H}_k}^2$$
$$\leq 2 \delta_{\widetilde{\mathsf{V}}}^2 M^d + 2 M_V \left\| \mu_{\mathcal{C}}(\mathbf{s}) - \mu_{\mathcal{C}}(\widetilde{\mathbf{s}}) \right\|_{\mathcal{H}_k}^2, \qquad (92)$$

where $\delta_{\widetilde{\mathsf{V}}}$ is the loss of value function caused by perturbation, $\left\| \mu_{\mathcal{C}}(\mathbf{s}) \right\|_{\mathcal{H}_k}^2 = \left\| \mathbb{E}\left[ k(\mathbf{S}^{\mathcal{C}}, \mathbf{S}'^{\mathcal{C}}) \right] \right\|_{\mathcal{H}_k}^2$, and $M_V$ denotes the upper bound of $\left\| w_{\mathsf{V}} \right\|_{\mathcal{H}_k}^2$.

Since Shapley values are the expectation of differences of value functions, by devising a coalition independent bound for the difference in value functions, the expectation disappears in our bound. Therefore, along with Lemma C.3 of

$$\delta_{\widetilde{\mathsf{V}}} \leq \frac{M_\eta q_{\mathsf{V}}^2}{1 - (q_{\mathsf{V}} - 1)\rho_{\mathsf{V}}} \left[ \frac{L_k}{1 - (q_{\mathsf{V}} - 1)\rho_{\mathsf{V}}} + M_k L_\psi + 2\sqrt{M_k} L_k \right] \varepsilon,$$

we have

$$\left\| \phi_i - \widetilde{\phi}_i \right\|_2^2 = \left\| \frac{1}{d} \sum_{\mathcal{C} \subseteq \mathcal{X} \setminus \{i\}} \binom{d-1}{|\mathcal{C}|}^{-1} [v_{\mathcal{C} \cup i}(\mathbf{s}) - v_{\mathcal{C}}(\mathbf{s}) - (v_{\mathcal{C} \cup i}(\widetilde{\mathbf{s}}) - v_{\mathcal{C}}(\widetilde{\mathbf{s}}))] \right\|_2^2$$
$$\leq \frac{2}{d} \sum_{\mathcal{C} \subseteq \mathcal{X} \setminus \{i\}} \binom{d-1}{|\mathcal{C}|}^{-1} \left[ \left\| v_{\mathcal{C}}(\mathbf{s}) - v_{\mathcal{C}}(\widetilde{\mathbf{s}}) \right\|_2^2 + \left\| v_{\mathcal{C} \cup \{i\}}(\mathbf{s}) - v_{\mathcal{C} \cup \{i\}}(\widetilde{\mathbf{s}}) \right\|_2^2 \right]$$
$$= 2 \mathbb{E}_{\mathcal{C}} \left[ \left\| v_{\mathcal{C}}(\mathbf{s}) - v_{\mathcal{C}}(\widetilde{\mathbf{s}}) \right\|_2^2 + \left\| v_{\mathcal{C} \cup \{i\}}(\mathbf{s}) - v_{\mathcal{C} \cup \{i\}}(\widetilde{\mathbf{s}}) \right\|_2^2 \right]. \qquad (93)$$

Since we have proven bounds for $T_{\mathcal{C}}^{(\text{on})} = \left\| \mu_{\mathcal{C}}^{(\text{on})}(\mathbf{s}) - \mu_{\mathcal{C}}^{(\text{on})}(\widetilde{\mathbf{s}}) \right\|_{\mathcal{H}_k}^2$ and $T_{\mathcal{C}}^{(\text{off})} = \left\| \mu_{\mathcal{C}}^{(\text{off})}(\mathbf{s}) - \mu_{\mathcal{C}}^{(\text{off})}(\widetilde{\mathbf{s}}) \right\|_{\mathcal{H}_k}^2$ that is coalition independent, we can directly substitute the bound inside the expectation. Therefore, the loss of RKHS-SHAP values caused by $B(\mathbf{s})$ is

$$\left\| \phi_i^{(\text{off})} - \widetilde{\phi}_i^{(\text{off})} \right\|_2^2 \leq 4 \delta_{\widetilde{\mathsf{V}}}^2 M^d + 4 M_V \delta_\psi M^d \qquad (94a)$$
$$\left\| \phi_i^{(\text{on})} - \widetilde{\phi}_i^{(\text{on})} \right\|_2^2 \leq 4 \delta_{\widetilde{\mathsf{V}}}^2 M^d + 8 M_V M_\Gamma \delta_\psi M^d. \qquad (94b)$$

In the case when we pick $k$ as a product RBF kernel, we have $\delta_\psi = 2 - 2\exp\left(-\frac{\varepsilon^2}{2l^2}\right)$ and $M = 1$. Then, we can simplify the result as

$$\left\|\phi_i^{(\text{off})} - \widetilde{\phi}_i^{(\text{off})}\right\|_2^2 \leq \frac{4M^d M_\eta^2 q_V^4 \varepsilon^2}{(1 - (q_V - 1)\rho_V)^2}\left(\frac{L_k}{1 - (q_V - 1)\rho_V} + M_k L_\psi + 2\sqrt{M_k}L_k\right)^2 + 4M_V\left(1 - \exp\left(-\frac{\varepsilon^2}{2l^2}\right)\right) \quad (95a)$$

$$\left\|\phi_i^{(\text{on})} - \widetilde{\phi}_i^{(\text{on})}\right\|_2^2 \leq \frac{4M^d M_\eta^2 q_V^4 \varepsilon^2}{(1 - (q_V - 1)\rho_V)^2}\left(\frac{L_k}{1 - (q_V - 1)\rho_V} + M_k L_\psi + 2\sqrt{M_k}L_k\right)^2 + 8M_V M_\Gamma\left(1 - \exp\left(-\frac{\varepsilon^2}{2l^2}\right)\right). \quad (95b)$$

### C.6.3. PROOF OF LEMMA C.3

According to the projection residual, the sparsification process can be divided into two cases.

**Case 1: Maintain the dictionary $\mathcal{D}_V$**   For online sparsification, we consider the following optimization problem:

$$\min_{\{\eta_j\}\in\mathcal{D}_V}\left\|\sum_j \eta_j \psi(\mathbf{s}_j) - \eta_\iota\psi(\mathbf{s}_\iota)\right\|_{\mathcal{H}_k}^2. \quad (96)$$

By expanding the norm and applying the reproducing property of the RKHS, we obtain:

$$\mathcal{L}(\{\eta_j\}) = \left\langle \sum_j \eta_j\psi(\mathbf{s}_j) - \eta_\iota\psi(\mathbf{s}_\iota), \sum_{j'} \eta_{j'}\psi(\mathbf{s}_{j'}) - \eta_\iota\psi(\mathbf{s}_\iota)\right\rangle_{\mathcal{H}_k}$$

$$= \sum_{j,j'} \eta_j k(\mathbf{s}_j, \mathbf{s}_{j'})\eta_{j'} - 2\sum_j \eta_j k(\mathbf{s}_j, \mathbf{s}_\iota)\eta_\iota + \eta_\iota^2 k(\mathbf{s}_\iota, \mathbf{s}_\iota). \quad (97)$$

Let $\eta_V = [\eta_1, \ldots, \eta_{q_V}]^\top \in \mathbb{R}^{q_V}$ denote the coefficient vector, and define the Gram matrix $\mathbf{K}_{V,V} \in \mathbb{R}^{q_V \times q_V}$ with entries $k(\mathbf{s}_j, \mathbf{s}_{j'})$, and the cross-kernel vector $\mathbf{k}_\iota \in \mathbb{R}^{q_V}$ with entries $k(\mathbf{s}_j, \mathbf{s}_\iota)\eta_\iota$. Then, the objective function becomes:

$$\mathcal{L}(\eta_V) = \eta_V^\top \mathbf{K}_{V,V}\eta_V - 2\eta_V^\top \mathbf{k}_\iota + \eta_\iota^2 k(\mathbf{s}_\iota, \mathbf{s}_\iota). \quad (98)$$

Setting the gradient $\nabla_{\eta_V}\mathcal{L} = 0$ yields the closed-form optimal solution:

$$\eta_V^* = \mathbf{K}_{V,V}^{-1}\mathbf{k}_\iota. \quad (99)$$

Thus, the optimal coefficient for each basis function $k(\mathbf{s}_j, \cdot)$ is given by:

$$\eta_j^* = \sum_{j'}\left[\mathbf{K}_{V,V}^{-1}\right]_{jj'} k(\mathbf{s}_{j'}, \mathbf{s}_\iota)\eta_\iota. \quad (100)$$

This solution corresponds to the approximate orthogonal projection of the new kernel function $\eta_\iota\psi(\mathbf{s}_\iota)$ onto the subspace spanned by the kernel atoms $\{\psi(\mathbf{s}_j)\}_{j=1}^{q_V}$ in $\mathcal{H}_k$, and is used to decide whether the new basis $\psi(\mathbf{s}_\iota)$ is sufficiently novel to be added to the sparse dictionary $\mathcal{D}_V$.

Therefore, we have

$$\delta_{\widetilde{V}} = \left\|\sum_j\sum_{j'}\left[\mathbf{K}_{V,V}^{-1}\right]_{jj'} k(\mathbf{s}_{j'}, \mathbf{s}_\iota)\eta_\iota\psi(\mathbf{s}_j) - \sum_j\sum_{j'}\left[\widetilde{\mathbf{K}}_{V,V}^{-1}\right]_{jj'} k(\widetilde{\mathbf{s}}_{j'}, \widetilde{\mathbf{s}}_\iota)\eta_\iota\psi(\widetilde{\mathbf{s}}_j)\right\|_{\mathcal{H}_k}$$

$$\leq \sum_j\sum_{j'}\left\|\left[\mathbf{K}_{V,V}^{-1}\right]_{jj'} k(\mathbf{s}_{j'}, \mathbf{s}_\iota)\eta_\iota\psi(\mathbf{s}_j) - \left[\widetilde{\mathbf{K}}_{V,V}^{-1}\right]_{jj'} k(\widetilde{\mathbf{s}}_{j'}, \widetilde{\mathbf{s}}_\iota)\eta_\iota\psi(\widetilde{\mathbf{s}}_j)\right\|_{\mathcal{H}_k}$$

$$\leq \sum_j\sum_{j'}\left\|\left[\mathbf{K}_{V,V}^{-1}\right]_{jj'} k(\mathbf{s}_{j'}, \mathbf{s}_\iota)\psi(\mathbf{s}_j) - \left[\widetilde{\mathbf{K}}_{V,V}^{-1}\right]_{jj'} k(\widetilde{\mathbf{s}}_{j'}, \widetilde{\mathbf{s}}_\iota)\psi(\widetilde{\mathbf{s}}_j)\right\|_{\mathcal{H}_k}\|\eta_\iota\|_2$$

$$\leq M_\eta\sum_j\sum_{j'}\left\|\left[\mathbf{K}_{V,V}^{-1}\right]_{jj'} k(\mathbf{s}_{j'}, \mathbf{s}_\iota)\psi(\mathbf{s}_j) - \left[\widetilde{\mathbf{K}}_{V,V}^{-1}\right]_{jj'} k(\widetilde{\mathbf{s}}_{j'}, \widetilde{\mathbf{s}}_\iota)\psi(\widetilde{\mathbf{s}}_j)\right\|_{\mathcal{H}_k}. \quad (101)$$

We consider bounding the RKHS norm

$$\left\| \left[ \mathbf{K}_{\mathsf{V},\mathsf{V}}^{-1} \right]_{jj'} k(\mathbf{s}_{j'}, \mathbf{s}_\iota) \psi(\mathbf{s}_j) - \left[ \widetilde{\mathbf{K}}_{\mathsf{V},\mathsf{V}}^{-1} \right]_{jj'} k(\widetilde{\mathbf{s}}_{j'}, \widetilde{\mathbf{s}}_\iota) \psi(\widetilde{\mathbf{s}}_j) \right\|_{\mathcal{H}_k}, \tag{102}$$

where $\mathbf{K}_{\mathsf{V},\mathsf{V}}$ and $\widetilde{\mathbf{K}}_{\mathsf{V},\mathsf{V}}$ denote Gram matrices over kernel dictionaries $\mathcal{D}_{\mathsf{V}} = \{\mathbf{s}_j\}$ and its perturbed version $\widetilde{\mathcal{D}}_{\mathsf{V}} = \{\widetilde{\mathbf{s}}_j\}$, respectively. Using the triangle inequality and RKHS norm properties, we decompose:

$$\left\| \left[ \mathbf{K}_{\mathsf{V},\mathsf{V}}^{-1} \right]_{jj'} k(\mathbf{s}_{j'}, \mathbf{s}_\iota) \psi(\mathbf{s}_j) - \left[ \widetilde{\mathbf{K}}_{\mathsf{V},\mathsf{V}}^{-1} \right]_{jj'} k(\widetilde{\mathbf{s}}_{j'}, \widetilde{\mathbf{s}}_\iota) \psi(\widetilde{\mathbf{s}}_j) \right\|_{\mathcal{H}_k}$$

$$\leq \left| \left[ \mathbf{K}_{\mathsf{V},\mathsf{V}}^{-1} - \widetilde{\mathbf{K}}_{\mathsf{V},\mathsf{V}}^{-1} \right]_{jj'} \right| \cdot \|k(\mathbf{s}_{j'}, \mathbf{s}_\iota)\|_2 \cdot \|\psi(\mathbf{s}_j)\|_{\mathcal{H}_k} + \left| \left[ \widetilde{\mathbf{K}}_{\mathsf{V},\mathsf{V}}^{-1} \right]_{jj'} \right| \cdot \|k(\mathbf{s}_{j'}, \mathbf{s}_\iota)\psi(\mathbf{s}_j) - k(\widetilde{\mathbf{s}}_{j'}, \widetilde{\mathbf{s}}_\iota)\psi(\widetilde{\mathbf{s}}_j)\|_{\mathcal{H}_k}$$

$$\leq \sqrt{M_k^3} \left| \left[ \mathbf{K}_{\mathsf{V},\mathsf{V}}^{-1} - \widetilde{\mathbf{K}}_{\mathsf{V},\mathsf{V}}^{-1} \right]_{jj'} \right| + \left| \left[ \widetilde{\mathbf{K}}_{\mathsf{V},\mathsf{V}}^{-1} \right]_{jj'} \right| \cdot \|k(\mathbf{s}_{j'}, \mathbf{s}_\iota)\psi(\mathbf{s}_j) - k(\widetilde{\mathbf{s}}_{j'}, \widetilde{\mathbf{s}}_\iota)\psi(\widetilde{\mathbf{s}}_j)\|_{\mathcal{H}_k}, \tag{103}$$

where $\|k(\mathbf{s}_{j'}, \mathbf{s}_\iota)\|_2 \leq M_k$ and $\|\psi(\mathbf{s}_j)\|_{\mathcal{H}_k} \leq \sqrt{M_k}$. Then, with the Lipschitz continuous of RBF $k$ and $\psi$, we have

$$\|k(\mathbf{s}_{j'}, \mathbf{s}_\iota)\psi(\mathbf{s}_j) - k(\widetilde{\mathbf{s}}_{j'}, \widetilde{\mathbf{s}}_\iota)\psi(\widetilde{\mathbf{s}}_j)\|_{\mathcal{H}_k}$$

$$\leq |k(\mathbf{s}_{j'}, \mathbf{s}_\iota)| \cdot \|\psi(\mathbf{s}_j) - \psi(\widetilde{\mathbf{s}}_j)\|_{\mathcal{H}_k} + |k(\mathbf{s}_{j'}, \mathbf{s}_\iota) - k(\widetilde{\mathbf{s}}_{j'}, \widetilde{\mathbf{s}}_\iota)| \cdot \|\psi(\widetilde{\mathbf{s}}_j)\|_{\mathcal{H}_k}$$

$$\leq M_k L_\psi \varepsilon + 2\sqrt{M_k} L_k \varepsilon. \tag{104}$$

Moreover, we apply the Banach perturbation lemma to the inverse kernel matrices. Then:

$$\left\| \mathbf{K}_{\mathsf{V},\mathsf{V}}^{-1} - \widetilde{\mathbf{K}}_{\mathsf{V},\mathsf{V}}^{-1} \right\|_{\mathrm{op}} \leq \|\mathbf{K}_{\mathsf{V},\mathsf{V}}^{-1}\|_{\mathrm{op}} \cdot \|\widetilde{\mathbf{K}}_{\mathsf{V},\mathsf{V}} - \mathbf{K}_{\mathsf{V},\mathsf{V}}\|_{\mathrm{op}} \cdot \|\widetilde{\mathbf{K}}_{\mathsf{V},\mathsf{V}}^{-1}\|_{\mathrm{op}}. \tag{105}$$

Suppose the dictionary centers satisfy a minimum separation $C_k > 0$ and $k$ is Gaussian. Then off-diagonal terms are bounded by $\rho_{\mathsf{V}} := \exp\left(-\frac{C_k^2}{2l^2}\right)$ and the smallest eigenvalue of $\mathbf{K}_{\mathsf{V},\mathsf{V}}$ satisfies

$$\lambda_{\min}(\mathbf{K}_{\mathsf{V},\mathsf{V}}) \geq 1 - (q_{\mathsf{V}} - 1)\rho_{\mathsf{V}}, \quad \Rightarrow \quad \|\mathbf{K}_{\mathsf{V},\mathsf{V}}^{-1}\|_{\mathrm{op}} \leq \frac{1}{1 - (q_{\mathsf{V}} - 1)\rho_{\mathsf{V}}}. \tag{106}$$

Combining with the kernel matrix perturbation

$$\|\widetilde{\mathbf{K}}_{\mathsf{V},\mathsf{V}} - \mathbf{K}_{\mathsf{V},\mathsf{V}}\| \leq L_k \varepsilon, \tag{107}$$

where we interpret $L_k$ as the Lipschitz constant for the operator norm on the entire Gram matrix. Then, we obtain the final upper bound:

$$\delta_{\tilde{\mathsf{V}}} \leq \frac{M_\eta q_{\mathsf{V}}^2}{1 - (q_{\mathsf{V}} - 1)\rho_{\mathsf{V}}} \left[ \frac{L_k}{1 - (q_{\mathsf{V}} - 1)\rho_{\mathsf{V}}} + M_k L_\psi + 2\sqrt{M_k} L_k \right] \varepsilon. \tag{108}$$

**Case 2: Update the dictionary** $\mathcal{D}_{\mathsf{V}}$    Without loss of generality, we assume $j^\star$ is replaced by $\iota$. Therefore, the optimization problem becomes

$$\min_{\{\eta_j\} \in \mathcal{D}'_{\mathsf{V}}} \left\| \sum_j \eta_j \psi(\mathbf{s}_j) - \eta_{j^\star} \psi(\mathbf{s}_{j^\star}) \right\|_{\mathcal{H}_k}^2. \tag{109}$$

By expanding the norm and applying the reproducing property of the RKHS, we obtain:

$$\mathcal{L}(\{\eta_j\}) = \left\langle \sum_j \eta_j \psi(\mathbf{s}_j) - \eta_{j^\star} \psi(\mathbf{s}_{j^\star}), \sum_{j'} \eta_{j'} \psi(\mathbf{s}_{j'}) - \eta_{j^\star} \psi(\mathbf{s}_{j^\star}) \right\rangle_{\mathcal{H}_k}$$

$$= \sum_{j,j'} \eta_j k(\mathbf{s}_j, \mathbf{s}_{j'}) \eta_{j'} - 2 \sum_j \eta_j k(\mathbf{s}_j, \mathbf{s}_{j^\star}) \eta_{j^\star} + \eta_{j^\star}^2 k(\mathbf{s}_{j^\star}, \mathbf{s}_{j^\star}). \tag{110}$$

Similar to the proof in case 1, we obtain the final upper bound:

$$\delta_{\tilde{V}} \le \frac{M_\eta q_V^2}{1-(q_V-1)\rho_V}\left[\frac{L_k}{1-(q_V-1)\rho_V} + M_k L_\psi + 2\sqrt{M_k}L_k\right]\varepsilon. \tag{111}$$

This result characterizes the sensitivity of the RKHS element to perturbations in both the kernel centers and the Gram matrix structure.

### C.6.4. PROOF OF LEMMA C.4

After the gradient obtained, RSA2C will execute the online sparsification, which can be divided into two cases. In this part, we ignore the time step $t$ without causing misunderstandings.

**Case 1: Maintain the dictionary** $\mathcal{D}_V$  For online sparsification, we consider the following optimization problem:

$$\min_{\{\eta_j\}\in\mathcal{D}_V}\left\|\sum_j \eta_j\psi(\mathbf{s}_j) - \eta_\iota\psi(\mathbf{s}_\iota)\right\|_{\mathcal{H}_k}^2. \tag{112}$$

By expanding the norm and applying the reproducing property of the RKHS, we obtain:

$$\begin{aligned}
\mathcal{L}(\{\eta_j\}) &= \left\langle\sum_j \eta_j\psi(\mathbf{s}_j) - \eta_\iota\psi(\mathbf{s}_\iota), \sum_{j'}\eta_{j'}\psi(\mathbf{s}_{j'}) - \eta_\iota\psi(\mathbf{s}_\iota)\right\rangle_{\mathcal{H}_k}\\
&= \sum_{j,j'}\eta_j k(\mathbf{s}_j,\mathbf{s}_{j'})\eta_{j'} - 2\sum_j \eta_j k(\mathbf{s}_j,\mathbf{s}_\iota)\eta_\iota + \eta_\iota^2 k(\mathbf{s}_\iota,\mathbf{s}_\iota).
\end{aligned} \tag{113}$$

Let $\eta_V = [\eta_1,\ldots,\eta_{q_V}]^\top \in \mathbb{R}^{q_V}$ denote the coefficient vector, and define the Gram matrix $\mathbf{K}_{V,V} \in \mathbb{R}^{q_V\times q_V}$ with entries $k(\mathbf{s}_j,\mathbf{s}_{j'})$, and the cross-kernel vector $\mathbf{k}_\iota \in \mathbb{R}^{q_V}$ with entries $k(\mathbf{s}_j,\mathbf{s}_\iota)\eta_\iota$. Then, the objective function becomes:

$$\mathcal{L}(\eta_V) = \eta_V^\top\mathbf{K}_{V,V}\eta_V - 2\eta_V^\top\mathbf{k}_\iota + \eta_\iota^2 k(\mathbf{s}_\iota,\mathbf{s}_\iota). \tag{114}$$

Setting the gradient $\nabla_{\eta_V}\mathcal{L} = 0$ yields the closed-form optimal solution:

$$\eta_V^* = \mathbf{K}_{V,V}^{-1}\mathbf{k}_\iota. \tag{115}$$

Thus, the optimal coefficient for each basis function $k(\mathbf{s}_j,\cdot)$ is given by:

$$\eta_j^* = \sum_{j'}\left[\mathbf{K}_{V,V}^{-1}\right]_{jj'}k(\mathbf{s}_{j'},\mathbf{s}_\iota)\eta_\iota. \tag{116}$$

This solution corresponds to the approximate orthogonal projection of the new kernel function $\eta_\iota\psi(\mathbf{s}_\iota)$ onto the subspace spanned by the kernel atoms $\{\psi(\mathbf{s}_j)\}_{j=1}^{q_V}$ in $\mathcal{H}_k$, and is used to decide whether the new basis $\psi(\mathbf{s}_\iota)$ is sufficiently novel to be added to the sparse dictionary $\mathcal{D}_V$.

Therefore, we have

$$\begin{aligned}
\delta_{PV} &= \left\|\sum_j \eta_j^\star\psi(\mathbf{s}_j) - \sum_j \eta_j\psi(\mathbf{s}_j)\right\|_{\mathcal{H}_k}\\
&= \left\|\sum_j\sum_{j'}\left[\mathbf{K}_{V,V}^{-1}\right]_{jj'}k(\mathbf{s}_{j'},\mathbf{s}_\iota)\eta_\iota\psi(\mathbf{s}_j) - \sum_j \eta_j\psi(\mathbf{s}_j)\right\|_{\mathcal{H}_k}\\
&\le \sum_j\left|\sum_{j'}\left[\mathbf{K}_{V,V}^{-1}\right]_{jj'}k(\mathbf{s}_{j'},\mathbf{s}_\iota)\eta_\iota - \eta_j\right|\cdot\|\psi(\mathbf{s}_j)\|_{\mathcal{H}_k} \le \sqrt{M_k}q_V\varepsilon_{PV},
\end{aligned} \tag{117}$$

where $\varepsilon_{PV} := \sup_j\left|\sum_{j'}\left[\mathbf{K}_{V,V}^{-1}\right]_{jj'}k(\mathbf{s}_{j'},\mathbf{s}_\iota)\eta_\iota - \eta_j\right|$ and $\|\psi(\mathbf{s}_j)\|_{\mathcal{H}_k} \le \sqrt{M_k}$.

**Case 2: Update the dictionary** $\mathcal{D}_V$   Without loss of generality, we assume $j^\star$ is replaced by $\iota$. Therefore, the optimization problem becomes

$$\min_{\{\eta_j\}\in\mathcal{D}'_V}\left\|\sum_j \eta_j \psi(\mathbf{s}_j) - \eta_{j^\star}\psi(\mathbf{s}_{j^\star})\right\|^2_{\mathcal{H}_k}. \tag{118}$$

By expanding the norm and applying the reproducing property of the RKHS, we obtain:

$$\mathcal{L}(\{\eta_j\}) = \left\langle \sum_j \eta_j \psi(\mathbf{s}_j) - \eta_{j^\star}\psi(\mathbf{s}_{j^\star}), \sum_{j'} \eta_{j'}\psi(\mathbf{s}_{j'}) - \eta_{j^\star}\psi(\mathbf{s}_{j^\star})\right\rangle_{\mathcal{H}_k}$$

$$= \sum_{j,j'} \eta_j k(\mathbf{s}_j, \mathbf{s}_{j'})\eta_{j'} - 2\sum_j \eta_j k(\mathbf{s}_j, \mathbf{s}_\iota)\eta_{j^\star} + \eta_{j^\star}^2 k(\mathbf{s}_{j^\star}, \mathbf{s}_{j^\star}). \tag{119}$$

Similar to the proof in case 1, we obtain the final upper bound:

$$\delta_{\mathsf{PV}} \le \sqrt{M_k}\, q_V \varepsilon_{\mathsf{PV}}, \tag{120}$$

where $\varepsilon_{\mathsf{PV}} := \sup_j \left|\sum_{j'}\left[\mathbf{K}_{V',V'}^{-1}\right]_{jj'} k(\mathbf{s}_{j'}, \mathbf{s}_{j^\star})\eta_{j^\star} - \eta_j\right|$. Therefore, we have

$$\varepsilon_{\mathsf{PV}} := \max\left\{\sup_{\psi_j\in\mathcal{D}_V}\left|\sum_{j'}\left[\mathbf{K}_{V',V'}^{-1}\right]_{jj'} k(\mathbf{s}_{j'}, \mathbf{s}_\iota)\eta_\iota - \eta_j\right|, \sup_{\psi_j\in\{\psi_\iota\}\cup(\mathcal{D}_V\setminus\{\psi_{j^\star}\})}\left|\sum_{j'}\left[\mathbf{K}_{V',V'}^{-1}\right]_{jj'} k(\mathbf{s}_{j'}, \mathbf{s}_{j^\star})\eta_{j^\star} - \eta_j\right|\right\}. \tag{121}$$

### C.6.5. PROOF OF LEMMA C.5

The population gradient is

$$g_t(w_{V,t}) = \mathbb{E}_{\mathbf{s}\sim D^{\pi_h,\Sigma}, \mathbf{a}\sim\pi_{h,\Sigma}(\cdot|\mathbf{s})}\left[\left(V_{w_{V,t}}(\mathbf{s}) - r(\mathbf{s},\mathbf{a}) - \gamma V_{w_{V,t}}(\mathbf{s}')\right)\psi(\mathbf{s})\right] + \lambda w_{V,t}, \tag{122}$$

and the optimality condition gives $g_t\left(w_{V,t}^\star\right) = 0$. Subtracting yields

$$g_t(w_{V,t}) = \mathbb{E}\left[\left(V_{w_{V,t}}(\mathbf{s}) - V_{w_{V,t}^\star}(\mathbf{s}) - \gamma\left(V_{w_{V,t}}(\mathbf{s}') - V_{w_{V,t}^\star}(\mathbf{s}')\right)\right)\psi(\mathbf{s})\right] + \lambda\left(w_{V,t} - w_{V,t}^\star\right)$$

$$= (\Gamma_s - \gamma\Gamma_{2s} + \lambda\mathbf{I})\left(w_{V,t} - w_{V,t}^\star\right), \tag{123}$$

where $\Gamma_s := \mathbb{E}[\psi(\mathbf{s})\otimes\psi(\mathbf{s})]$ and $\Gamma_{2s} := \mathbb{E}[\psi(\mathbf{s})\otimes\psi(\mathbf{s}')]$. Hence, by submultiplicativity of operator norm,

$$\|g_t(w_{V,t})\|^2_{\mathcal{H}_k} \le \|\Gamma_s - \gamma\Gamma_{2s} + \lambda\mathbf{I}\|^2_{\mathrm{op}}\|w_{V,t} - w_{V,t}^\star\|^2_{\mathcal{H}_k}. \tag{124}$$

We now bound $\|\Gamma_s - \gamma\Gamma_{2s} + \lambda\mathbf{I}\|_{\mathrm{op}}$. Assume the kernel is bounded as $\|\psi(\mathbf{s})\|^2_{\mathcal{H}_k} = k(\mathbf{s},\mathbf{s}) \le M_k$; then $\|\Gamma_s\|_{\mathrm{op}} \le \mathbb{E}\|\psi(\mathbf{s})\|^2 \le M_k$ and $\|\Gamma_{2s}\|_{\mathrm{op}} \le \mathbb{E}\left[\|\psi(\mathbf{s})\|\|\psi(\mathbf{s}')\|\right] \le M_k$ (by Cauchy-Schwarz and boundedness). Therefore,

$$\|\Gamma_s - \gamma\Gamma_{2s} + \lambda\mathbf{I}\|_{\mathrm{op}} \le \|\Gamma_s\|_{\mathrm{op}} + \gamma\|\Gamma_{2s}\|_{\mathrm{op}} + \lambda \le (1+\gamma)M_k + \lambda, \tag{125}$$

and consequently

$$\|g_t(w_{V,t})\|^2_{\mathcal{H}_k} \le C_1\left\|w_{V,t} - w_{V,t}^\star\right\|^2_{\mathcal{H}_k}, \qquad C_1 := ((1+\gamma)M_k + \lambda)^2. \tag{126}$$

### C.6.6. PROOF OF LEMMA C.6

Using $\|AB\|_{\mathrm{op}} \le \|A\|_{\mathrm{op}}\|B\|_{\mathrm{op}}$ and $\|\Sigma_{\mathsf{K}}\|_{\mathrm{op}} \le M_{\Sigma_K}$, we have

$$\|K'(\mathbf{s},\cdot) - K(\mathbf{s},\cdot)\|_{\mathrm{op}} = \|\kappa'_\phi(\mathbf{s},\cdot)\Sigma_{\mathsf{K}} - \kappa_\phi(\mathbf{s},\cdot)\Sigma_{\mathsf{K}}\|_{\mathrm{op}}$$

$$\le \|\kappa'_\phi(\mathbf{s},\cdot) - \kappa_\phi(\mathbf{s},\cdot)\|_{\mathrm{op}}\|\Sigma_{\mathsf{K}}\|_{\mathrm{op}}$$

$$\le M_{\Sigma_K}\|\kappa'_\phi(\mathbf{s},\cdot) - \kappa_\phi(\mathbf{s},\cdot)\|_{\mathrm{op}}$$

$$= M_{\Sigma_K}\left\|\exp\left(-\frac{1}{2}(\mathbf{s}-\cdot)^\top w'(\mathbf{s}-\cdot)\right) - \exp\left(-\frac{1}{2}(\mathbf{s}-\cdot)^\top w(\mathbf{s}-\cdot)\right)\right\|_{\mathrm{op}}. \tag{127}$$

For the Mahalanobis RBF $\kappa_{\mathbf{W}}(\mathbf{s}, \mathbf{s}') = \exp(-\frac{1}{2}\mathbf{x}^\top \mathbf{W}\mathbf{x})$ with $\mathbf{x} := \mathbf{s} - \mathbf{s}'$, the mean-value theorem gives, for some $\widetilde{\mathbf{W}}$ on the segment between $\mathbf{W}$ and $\mathbf{W}'$,

$$|\kappa_{\mathbf{W}'} - \kappa_{\mathbf{W}}| \leq \tfrac{1}{2}e^{-\frac{1}{2}\mathbf{x}^\top \widetilde{\mathbf{W}}\mathbf{x}}|\mathbf{x}^\top (\mathbf{W}' - \mathbf{W})\mathbf{x}| \leq \tfrac{1}{2}\|\mathbf{W}' - \mathbf{W}\|_{\mathrm{op}}\|\mathbf{x}\|_2^2.$$

If $\|\mathbf{s}\|_2 \leq M_S$ then $\|\mathbf{x}\|_2 \leq 2M_S$, hence $\sup_{\mathbf{s}} \|K'(\mathbf{s}, \cdot) - K(\mathbf{s}, \cdot)\|_{\mathrm{op}} \leq 2M_{\Sigma_K}M_S^2\|\mathbf{W}' - \mathbf{W}\|_{\mathrm{op}}$, i.e.,

$$\delta_{\mathsf{K},h} \leq q_{\mathsf{A}}^2 M_c^2 \left(\sup_{\mathbf{s}} \|K'(\mathbf{s}, \cdot) - K(\mathbf{s}, \cdot)\|_{\mathrm{op}}\right)^2, \qquad M_c := \max_j \|\mathbf{c}_j\|_2, \tag{128}$$

$$\sup_{\mathbf{s}} \|K'(\mathbf{s}, \cdot) - K(\mathbf{s}, \cdot)\|_{\mathrm{op}} \leq 2M_{\Sigma_K}M_S^2\|\mathbf{W}' - \mathbf{W}\|_{\mathrm{op}}. \tag{129}$$

Combining (128) and (129) yields

$$\delta_{\mathsf{K},h} \leq 4M_{\Sigma_K}^2 M_S^4 q_{\mathsf{A}}^2 M_c^2 \|\mathbf{W}' - \mathbf{W}\|_{\mathrm{op}}^2. \tag{130}$$

By (9) and Cauchy-Schwarz in $\mathcal{H}_k$,

$$|v'_{\mathcal{C}}(\mathbf{s}) - v_{\mathcal{C}}(\mathbf{s})| = |\langle w'_{\mathsf{V}} - w_{\mathsf{V}}, \mu_{\mathcal{C}}(\mathbf{s})\rangle| \leq \|w'_{\mathsf{V}} - w_{\mathsf{V}}\|_{\mathcal{H}_k}\|\mu_{\mathcal{C}}(\mathbf{s})\|_{\mathcal{H}_k} \leq \|w'_{\mathsf{V}} - w_{\mathsf{V}}\|_{\mathcal{H}_k}C_\mu, \tag{131}$$

we have

$$\sup_{\mathbf{s},\mathcal{C}} |v'_{\mathcal{C}}(\mathbf{s}) - v_{\mathcal{C}}(\mathbf{s})| = \sup_{\mathbf{s},\mathcal{C}} |\langle w'_{\mathsf{V}} - w_{\mathsf{V}}, \mu_{\mathcal{C}}(\mathbf{s})\rangle_{\mathcal{H}_k}| \leq \|w'_{\mathsf{V}} - w_{\mathsf{V}}\|_{\mathcal{H}_k}\sup_{\mathbf{s},\mathcal{C}} \|\mu_{\mathcal{C}}(\mathbf{s})\|_{\mathcal{H}_k} \leq \|w'_{\mathsf{V}} - w_{\mathsf{V}}\|_{\mathcal{H}_k}C_\mu. \tag{132}$$

The Shapley aggregation (11) is an average (with weights summing to one) of differences of the form $f_{\mathcal{C}\cup\{i\}} - f_{\mathcal{C}}$, so Jensen's inequality gives

$$\|\phi' - \phi\|_\infty \leq \frac{2}{d}\sum_{\mathcal{C}}\binom{d-1}{|\mathcal{C}|}^{-1}\sup_{\mathbf{s},\mathcal{C}} |v'_{\mathcal{C}}(\mathbf{s}) - v_{\mathcal{C}}(\mathbf{s})| \leq 2\|w'_{\mathsf{V}} - w_{\mathsf{V}}\|_{\mathcal{H}_k}C_\mu. \tag{133}$$

Substituting (133) into (130) yields

$$\delta_{\mathsf{K},h} \leq 16M_{\Sigma_K}^2 M_S^4 q_{\mathsf{A}}^2 M_c^2 C_\mu^2 \|w'_{\mathsf{V}} - w_{\mathsf{V}}\|_{\mathcal{H}_k}^2. \tag{134}$$

Finally, for tensor features in (10a), with each component map bounded by $M$, the product structure implies $C_\mu \leq M^{d/2}$, so

$$\delta_{\mathsf{K},h} \leq 16M_{\Sigma_K}^2 M_S^4 q_{\mathsf{A}}^2 M_c^2 M^d \|w'_{\mathsf{V}} - w_{\mathsf{V}}\|_{\mathcal{H}_k}^2. \tag{135}$$

### C.6.7. PROOF OF LEMMA C.7

Let $g(h) := \nabla \log \pi_h(\mathbf{a} \mid \mathbf{s})$, $\forall h \in \mathcal{H}_K$. According to Proposition B.2, we get the gradient expression as

$$g(h) = \nabla_h \log \pi_h(\mathbf{a} \mid \mathbf{s}) = K(\mathbf{s}, \cdot)\Sigma^{-1}(\mathbf{a} - h(\mathbf{s})). \tag{136}$$

For any $h, h' \in \mathcal{H}_K$,

$$\begin{aligned}
\|g(h') - g(h)\|_{\mathcal{H}_K} &= \left\|K'(\mathbf{s}, \cdot)\Sigma^{-1}(\mathbf{a} - h'(\mathbf{s})) - K(\mathbf{s}, \cdot)\Sigma^{-1}(\mathbf{a} - h(\mathbf{s}))\right\|_{\mathcal{H}_K} \\
&\leq \left\|K'(\mathbf{s}, \cdot)\Sigma^{-1}(\mathbf{a} - h'(\mathbf{s})) - K'(\mathbf{s}, \cdot)\Sigma^{-1}(\mathbf{a} - h(\mathbf{s}))\right\|_{\mathcal{H}_K} \\
&\quad + \left\|K'(\mathbf{s}, \cdot)\Sigma^{-1}(\mathbf{a} - h(\mathbf{s})) - K(\mathbf{s}, \cdot)\Sigma^{-1}(\mathbf{a} - h(\mathbf{s}))\right\|_{\mathcal{H}_K} \\
&= \left\|K'(\mathbf{s}, \cdot)\Sigma^{-1}(h(\mathbf{s}) - h'(\mathbf{s}))\right\|_{\mathcal{H}_K} + \left\|(K'(\mathbf{s}, \cdot) - K(\mathbf{s}, \cdot))\Sigma^{-1}(\mathbf{a} - h(\mathbf{s}))\right\|_{\mathcal{H}_K} \\
&\leq \|K'(\mathbf{s}, \cdot)\|_{\mathrm{op}}\|\Sigma^{-1}\|_{\mathrm{op}}\|h'(\mathbf{s}) - h(\mathbf{s})\|_2 + \|K'(\mathbf{s}, \cdot) - K(\mathbf{s}, \cdot)\|_{\mathrm{op}}\|\Sigma^{-1}\|_{\mathrm{op}}\|\mathbf{a} - h(\mathbf{s})\|_2 \\
&\leq \sqrt{M_K}\|\Sigma^{-1}\|_{\mathrm{op}}\sup_{\mathbf{s}} \|h'(\mathbf{s}) - h(\mathbf{s})\|_2 + 2M_a\|K'(\mathbf{s}, \cdot) - K(\mathbf{s}, \cdot)\|_{\mathrm{op}}\|\Sigma^{-1}\|_{\mathrm{op}} \\
&\leq \underbrace{\left(\sqrt{M_K}\|\Sigma^{-1}\|_{\mathrm{op}}C_{\mathrm{ev}}\right)}_{=:L_{\mathrm{grad}}}\|h' - h\|_{\mathcal{H}_K} + 2M_a M_\Sigma \delta_{\mathsf{K},h},
\end{aligned}$$

where $M_K := \sup_{\mathbf{s}} \|K(\mathbf{s}, \mathbf{s})\|_{\mathrm{op}}$ and $C_{\mathrm{ev}}$ is the evaluation operator bound $\|v(\mathbf{s})\|_2 \leq C_{\mathrm{ev}}\|v\|_{\mathcal{H}_K}$ for all $v \in \mathcal{H}_K$. Besides, $\delta_{\mathsf{K},h}$ is defined in Lemma C.6.

Thus, we obtain

$$\log \pi_{h_{t+1}}(\mathbf{a} \mid \mathbf{s}) \geq \log \pi_{h_t}(\mathbf{a} \mid \mathbf{s}) + \langle g(h_t), h_{t+1} - h_t \rangle_{\mathcal{H}_K} - \frac{L_{\mathrm{grad}}}{2}\|h_{t+1} - h_t\|_{\mathcal{H}_K}^2 - 2M_a^2 M_\Sigma^2 \delta_{\mathsf{K},h_t}. \tag{137}$$

### C.6.8. Proof of Lemma C.8

In this part, we prove Lemma C.8 by considering two different cases.

**Case 1: Maintain the dictionary $\mathcal{D}_A$**  For the vector-valued RKHS $\mathcal{H}_K$ associated with the operator-valued kernel $K : \mathcal{S} \times \mathcal{S} \to \mathbb{R}^{m \times m}$, the Actor mean function takes the form

$$h(\mathbf{s}) = \sum_{j=1}^{q_A} K(\mathbf{s}, \mathbf{s}_j)\mathbf{c}_j,$$

where $\mathbf{c}_j \in \mathbb{R}^m$ are coefficient vectors. Given a new candidate atom $K(\cdot, \mathbf{s}_\iota)\mathbf{c}_\iota$, online sparsification projects it onto the subspace spanned by $\{K(\cdot, \mathbf{s}_j)\mathbf{c} : \mathbf{c} \in \mathbb{R}^m, j = 1, \ldots, q\}$:

$$\min_{\{\mathbf{c}_j\}} \left\| \sum_{j=1}^{q} K(\cdot, \mathbf{s}_j)\mathbf{c}_j - K(\cdot, \mathbf{s}_\iota)\mathbf{c}_\iota \right\|_{\mathcal{H}_K}^2. \tag{138}$$

Using the reproducing property of vector-valued RKHSs,

$$\mathcal{L}(\{\mathbf{c}_j\}) = \sum_{j,j'=1}^{q_A} \mathbf{c}_j^\top K(\mathbf{s}_j, \mathbf{s}_{j'})\mathbf{c}_{j'} - 2\sum_{j=1}^{q_A} \mathbf{c}_j^\top K(\mathbf{s}_j, \mathbf{s}_\iota)\mathbf{c}_\iota + \mathbf{c}_\iota^\top K(\mathbf{s}_\iota, \mathbf{s}_\iota)\mathbf{c}_\iota. \tag{139}$$

Let $\mathbf{c}_A = \mathrm{col}(\mathbf{c}_1, \ldots, \mathbf{c}_{q_A}) \in \mathbb{R}^{q_A m}$ denote the stacked coefficient vector, and define the block Gram matrix $\mathbf{K}_{A,A} \in \mathbb{R}^{q_A m \times q_A m}$ with $(j, j')$-block $K(\mathbf{s}_j, \mathbf{s}_{j'})$, and the block cross-kernel $\mathbf{K}_{A,\iota} \in \mathbb{R}^{q_A m \times m}$ with block $K(\mathbf{s}_j, \mathbf{s}_\iota)$. Then,

$$\mathcal{L}(\mathbf{c}_A) = \mathbf{c}_A^\top \mathbf{K}_{A,A}\mathbf{c}_A - 2\mathbf{c}_A^\top \mathbf{K}_{A,\iota}\mathbf{c}_\iota + \mathbf{c}_\iota^\top K(\mathbf{s}_\iota, \mathbf{s}_\iota)\mathbf{c}_\iota. \tag{140}$$

Setting $\nabla_{\mathbf{c}_A}\mathcal{L} = 0$ yields the closed-form optimal solution:

$$\mathbf{c}_A^\star = \mathbf{K}_{A,A}^{-1}\mathbf{K}_{A,\iota}\mathbf{c}_\iota. \tag{141}$$

The approximation error is bounded as

$$\delta_{\mathsf{PA}} := \left\| \sum_{j=1}^{q_A} K(\cdot, \mathbf{s}_j)\mathbf{c}_j^\star - \sum_{j=1}^{q_A} K(\cdot, \mathbf{s}_j)\mathbf{c}_j \right\|_{\mathcal{H}_K} \leq \sum_{j=1}^{q_A} \|K(\cdot, \mathbf{s}_j)(\mathbf{c}_j^\star - \mathbf{c}_j)\|_{\mathcal{H}_K} \leq \sqrt{M_K}q_A \varepsilon_{\mathsf{PA}}, \tag{142}$$

where $M_K := \sup_{\mathbf{s}} \|K(\mathbf{s}, \mathbf{s})\|_{\mathrm{op}}$ and $\varepsilon_{\mathsf{PA}} := \sup_j \|\mathbf{c}_j^\star - \mathbf{c}_j\|_2$.

**Case 2: Update the dictionary $\mathcal{D}_A$**  Suppose an existing dictionary element $\mathbf{s}_{j^\star}$ is replaced by $\mathbf{s}_\iota$, yielding $\mathcal{D}_A'$. The projection problem becomes

$$\min_{\{\mathbf{c}_j\} \in \mathcal{D}_A'} \left\| \sum_{j \in \mathcal{D}_A'} K(\cdot, \mathbf{s}_j)\mathbf{c}_j - K(\cdot, \mathbf{s}_{j^\star})\mathbf{c}_{j^\star} \right\|_{\mathcal{H}_K}^2. \tag{143}$$

Similar to Case 1, with the block Gram matrix $\mathbf{K}_{A',A'}$ and cross-kernel $\mathbf{K}_{A',j^\star}$, the optimal coefficients are

$$\mathbf{c}_{A'}^\star = \mathbf{K}_{A',A'}^{-1}\mathbf{K}_{A',j^\star}\mathbf{c}_{j^\star}. \tag{144}$$

The error bound is

$$\delta_{\mathsf{PA}} \le \sqrt{M_K} q_{\mathsf{A}} \varepsilon_{\mathsf{PA}}, \tag{145}$$

where

$$\varepsilon_{\mathsf{PA}} := \max \left\{ \sup_{\psi_j \in \mathcal{D}_{\mathsf{A}}} \|[\mathbf{K}_{\mathsf{A},\mathsf{A}}^{-1} \mathbf{K}_{\mathsf{A},\iota}] \mathbf{c}_\iota - \mathbf{c}_j\|_2, \sup_{\psi_j \in \{\mathbf{s}_\iota\} \cup (\mathcal{D}_{\mathsf{A}} \setminus \{\mathbf{s}_{j^\star}\})} \|[\mathbf{K}_{\mathsf{A},\mathsf{A}}^{-1} \mathbf{K}_{\mathsf{A},j^\star}] \mathbf{c}_{j^\star} - \mathbf{c}_j\|_2 \right\}. \tag{146}$$

### C.6.9. PROOF OF LEMMA C.9

Recall $w_{\mathsf{V}}^\star(h) = (B(h) + \lambda \mathbf{I})^{-1} b(h)$ where $B(h) := \Gamma_s(h) - \gamma \Gamma_{2s}(h)$ and $b(h) := \mathbb{E}[r(\mathbf{s}, \mathbf{a}) \psi(s)]$, with $h$ denoting the Actor parameters. Using the inverse perturbation identity $(X + \Delta)^{-1} - X^{-1} = -X^{-1} \Delta (X + \Delta)^{-1}$ and operator submultiplicativity,

$$\|w_{\mathsf{V}}^\star(h') - w_{\mathsf{V}}^\star(h)\| \le \|(B(h') + \lambda \mathbf{I})^{-1}\| \|b(h') - b(h)\| + \|(B(h') + \lambda \mathbf{I})^{-1} - (B(h) + \lambda \mathbf{I})^{-1}\| \|b(h)\|$$

$$\le \frac{1}{\lambda} \|b(h') - b(h)\| + \frac{1}{\lambda^2} \|B(h') - B(h)\|_{\mathrm{op}} \|b(h)\|. \tag{147}$$

Assume bounded feature map $\|\psi(s)\|_{\mathcal{H}_\psi}^2 \le M$ and bounded reward $r \in [0, 1]$. Let $C_\nu$ be the Lipschitz constant of the on-policy visitation distribution w.r.t. $h$ (i.e., $\|\nu_{\pi_h} - \nu_{\pi_{h'}}\|_{\mathrm{TV}} \le C_\nu \|h - h'\|$), which is standard under the ergodicity/Lipschitz policy assumptions. Then

$$\|b(h') - b(h)\| \le \sqrt{M} C_\nu \|h' - h\|$$
$$\|B(h') - B(h)\|_{\mathrm{op}} \le 2(1 + \gamma) M C_\nu \|h' - h\|,$$
$$\|b(h)\| \le \mathbb{E}[|r| \|\psi\|] \le \sqrt{M}.$$

Combining gives the Lipschitz drift of the target Critic:

$$\|w_{\mathsf{V}}^\star(h') - w_{\mathsf{V}}^\star(h)\| \le L_{\mathsf{V}} \|h' - h\|, \qquad L_{\mathsf{V}} := C_\nu \sqrt{M} \left( \frac{1}{\lambda} + \frac{2(1 + \gamma)M}{\lambda^2} \right). \tag{148}$$

### C.6.10. PROOF OF LEMMA C.11

We provide the proof of Lemma C.11 in three major steps.

**Proof sketch of Lemma C.11.** The proof of Lemma C.11 consists of three steps as we briefly describe as follows.

Firstly, we conduct step 1 to decompose tracking error. We decompose the tracking error $\left\| w_{\mathsf{V},t+1} - w_{\mathsf{V},t+1}^\star \right\|_{\mathcal{H}_k}^2$ into a projection error term, an exponentially decaying term, a variance term, a bias error term, a fixed-point shift error term, and a slow drift error term. Next, we handle step 2 to bound these error terms. And finally, we have step 3 to recursively refine tracking error bound. We further show that the slow-drift error term diminishes as the tracking error diminishes. By recursively substituting the preliminary bound of $\left\| w_{\mathsf{V},t} - w_{\mathsf{V},t}^\star \right\|_{\mathcal{H}_k}^2$ into the slow-drift term, we obtain the refined decay rate of the tracking error.

**Step 1: Decomposing tracking error.** We define the tracking error as $w_{\mathsf{V},t+1} - w_{\mathsf{V}}^\star$ under dictionary $\mathcal{D}_{\mathsf{V},t+1}$ and $\mathcal{D}_{\mathsf{V}}^\star$. Then, we bound the recursion of the tracking error as follows. For any $t \ge 0$, we derive

$$\begin{aligned}
\left\| w_{\mathsf{V},t+1} - w_{\mathsf{V},t+1}^\star \right\|_{\mathcal{H}_k}^2 &= \left\| w_{\mathsf{V},t+1} - \overline{w}_{\mathsf{V},t+1} + \overline{w}_{\mathsf{V},t+1} - w_{\mathsf{V},t+1}^\star \right\|_{\mathcal{H}_k}^2 \\
&\le \left\| w_{\mathsf{V},t+1} - \overline{w}_{\mathsf{V},t+1} \right\|_{\mathcal{H}_k}^2 + \left\| \overline{w}_{\mathsf{V},t+1} - w_{\mathsf{V},t+1}^\star \right\|_{\mathcal{H}_k}^2 \\
&= \left\| w_{\mathsf{V},t+1} - \overline{w}_{\mathsf{V},t+1} \right\|_{\mathcal{H}_k}^2 + \left\| w_{\mathsf{V},t} + \alpha_t^{\mathsf{v}} \hat{g}_t(w_{\mathsf{V},t}) - w_{\mathsf{V},t+1}^\star \right\|_{\mathcal{H}_k}^2 \\
&= \left\| w_{\mathsf{V},t+1} - \overline{w}_{\mathsf{V},t+1} \right\|_{\mathcal{H}_k}^2 + \left\| w_{\mathsf{V},t} - w_{\mathsf{V},t}^\star + \alpha_t^{\mathsf{v}} \hat{g}_t(w_{\mathsf{V},t}) + w_{\mathsf{V},t}^\star - w_{\mathsf{V},t+1}^\star \right\|_{\mathcal{H}_k}^2 \\
&\le \left\| w_{\mathsf{V},t} - w_{\mathsf{V},t}^\star \right\|_{\mathcal{H}_k}^2 + \left\| w_{\mathsf{V},t+1} - \overline{w}_{\mathsf{V},t+1} \right\|_{\mathcal{H}_k}^2 + (\alpha_t^{\mathsf{v}})^2 \left\| \hat{g}_t(w_{\mathsf{V},t}) \right\|_{\mathcal{H}_k}^2 + \left\| w_{\mathsf{V},t}^\star - w_{\mathsf{V},t+1}^\star \right\|_{\mathcal{H}_k}^2 \\
&\quad + 2\alpha_t^{\mathsf{v}} \langle \hat{g}_t(w_{\mathsf{V},t}), w_{\mathsf{V},t} - w_{\mathsf{V},t}^\star \rangle_{\mathcal{H}_k} + 2\alpha_t^{\mathsf{v}} \langle \hat{g}_t(w_{\mathsf{V},t}), w_{\mathsf{V},t}^\star - w_{\mathsf{V},t+1}^\star \rangle_{\mathcal{H}_k} \\
&\quad + \langle w_{\mathsf{V},t} - w_{\mathsf{V},t}^\star, w_{\mathsf{V},t}^\star - w_{\mathsf{V},t+1}^\star \rangle_{\mathcal{H}_k}, \tag{149}
\end{aligned}$$

where the first line holds with $\overline{w}_{V,t+1} = w_{V,t} + \alpha_t^V \hat{g}_t (w_{V,t})$ before sparsification and the second inequation comes from triangle inequality.

**Step 2: Bounding each terms.** In this part, we bound the five error terms.

**Bounding the projection error term** $\|w_{V,t+1} - \overline{w}_{V,t+1}\|_{\mathcal{H}_k}^2$. According to Lemma C.4, we have

$$\|\overline{w}_{t+1} - w_{t+1}\|_{\mathcal{H}_k}^2 \leq M_k q_V^2 \varepsilon_{PV}^2. \tag{150}$$

**Bounding the variance term** $\|\hat{g}_t (w_{V,t})\|_{\mathcal{H}_k}^2$. We bound $\|\hat{g}_t (w_{V,t})\|_{\mathcal{H}_k}^2$ by first decomposing it into the sum of a sampling part and a population part as

$$\begin{aligned}
\|\hat{g}_t (w_{V,t})\|_{\mathcal{H}_k}^2 &= \|\hat{g}_t (w_{V,t}) - g_t (w_{V,t}) + g_t (w_{V,t})\|_{\mathcal{H}_k}^2 \\
&\leq \|\hat{g}_t (w_{V,t}) - g_t (w_{V,t})\|_{\mathcal{H}_k}^2 + \|g_t (w_{V,t})\|_{\mathcal{H}_k}^2,
\end{aligned} \tag{151}$$

where we use triangle inequality. According to Lemma C.5, the population gradient is bounded as

$$\|g_t (w_{V,t})\|_{\mathcal{H}_k}^2 \leq C_1 \|w_{V,t} - w_{V,t}^\star\|_{\mathcal{H}_k}^2, \qquad C_1 := ((1+\gamma)M_k + \lambda)^2. \tag{152}$$

We next control the sampling term. Write the empirical gradient as

$$\hat{g}_t (w_{V,t}) = \frac{1}{n} \sum_{\iota=1}^n [(\Gamma_s(\mathbf{s}_\iota) - \gamma\Gamma_{2s}(\mathbf{s}_\iota, \mathbf{s}_\iota')) w_{V,t} - r(\mathbf{s}_\iota, \mathbf{a}_\iota)\psi(\mathbf{s}_\iota)], \tag{153}$$

and define $B_t := \Gamma_s - \gamma\Gamma_{2s}$, $\hat{B}_t := \Gamma_s(\mathbf{s}_\iota) - \gamma\Gamma_{2s}(\mathbf{s}_\iota, \mathbf{s}_\iota')$, $b_t := \mathbb{E}[r(\mathbf{s}, \mathbf{a})\psi(\mathbf{s})]$, $\hat{b}_t := r(\mathbf{s}_\iota, \mathbf{a}_\iota)\psi(\mathbf{s}_\iota)$. Then

$$\begin{aligned}
\hat{g}_t (w_{V,t}) - g_t (w_{V,t}) &= (\hat{B}_t - B_t)w_{V,t} - (\hat{b}_t - b_t) \\
&= (\hat{B}_t - B_t)(w_{V,t} - w_{V,t}^\star) + (\hat{B}_t - B_t)w_{V,t}^\star - (\hat{b}_t - b_t),
\end{aligned} \tag{154}$$

and by triangle inequality and submultiplicativity of $\|\cdot\|_{op}$,

$$\|\hat{g}_t (w_{V,t}) - g_t (w_{V,t})\|_{\mathcal{H}_k}^2 \leq 3\Big( \|\hat{B}_t - B_t\|_{op}^2 \|w_{V,t} - w_{V,t}^\star\|_{\mathcal{H}_k}^2 + \|\hat{B}_t - B_t\|_{op}^2 \|w_{V,t}^\star\|_{\mathcal{H}_k}^2 + \|\hat{b}_t - b_t\|_{\mathcal{H}_k}^2 \Big). \tag{155}$$

Assume on-policy samples come from a geometric mixing Markov chain with effective sample size $n_{eff} \asymp n/\tau_{mix}$; using a Hilbert-space (operator-valued) Bernstein inequality, there exists a universal constant choice such that, with probability at least $1 - \delta$,

$$\|\hat{B}_t - B_t\|_{op} \leq \Delta_B := (1+\gamma)M_\Gamma \left( \sqrt{\frac{2\ln(6/\delta)}{n_{eff}}} + \frac{2\ln(6/\delta)}{3n_{eff}} \right), \tag{156}$$

$$\|\hat{b}_t - b_t\|_{\mathcal{H}_k} \leq \Delta_b := \sqrt{M_\Gamma} \left( \sqrt{\frac{2\ln(6/\delta)}{n_{eff}}} + \frac{2\ln(6/\delta)}{3n_{eff}} \right), \tag{157}$$

where we define $M_\Gamma := \sup \|\Gamma - \lambda\mathbf{I}\|_{op}$.

Note that the $(1+\gamma)$ factor in $\Delta_B$ comes from $\hat{B}_t - B_t = (\hat{\Gamma}_s - \Gamma_s) - \gamma(\hat{\Gamma}_{2s} - \Gamma_{2s})$ and the triangle inequality. Besides, $M_\Gamma$ is the almost-sure bound on the operator norm of each summand. And the $(2/3)$ factor appears in the standard Bernstein tail bound for bounded summands. Consequently, using $\|w_{V,t}^\star\|_{\mathcal{H}_k} \leq M_V$,

$$\begin{aligned}
\|\hat{g}_t (w_{V,t}) - g_t (w_{V,t})\|_{\mathcal{H}_k}^2 &\leq 3 \left( \Delta_B^2 \|w_{V,t} - w_{V,t}^\star\|_{\mathcal{H}_k}^2 + \Delta_B^2 M_V^2 + \Delta_b^2 \right) \\
&= \frac{C_B}{n_{eff}} \left( \|w_{V,t} - w_{V,t}^\star\|_{\mathcal{H}_k}^2 + M_V^2 \right) + \frac{C_b}{n_{eff}}.
\end{aligned} \tag{158}$$

Armed with $r \in [0, 1]$, we can take

$$C_B = 6(1 + \gamma)^2 M_\Gamma^2 \left(2 \ln \frac{6}{\delta}\right) \quad \text{and} \quad C_b = 6 M_\Gamma \left(2 \ln \frac{6}{\delta}\right). \tag{159}$$

Finally, combining with the population bound gives

$$\|\hat{g}_t(w_{\mathsf{V},t})\|_{\mathcal{H}_k}^2 \leq C_1 \|w_{\mathsf{V},t} - w_{\mathsf{V},t}^\star\|_{\mathcal{H}_k}^2 + \frac{C_B}{n_{\text{eff}}}\left(\|w_{\mathsf{V},t} - w_{\mathsf{V},t}^\star\|_{\mathcal{H}_k}^2 + M_V^2\right) + \frac{C_b}{n_{\text{eff}}}$$

$$= \left(C_1 + \frac{C_B}{n_{\text{eff}}}\right)\|w_{\mathsf{V},t} - w_{\mathsf{V},t}^\star\|_{\mathcal{H}_k}^2 + \frac{C_B}{n_{\text{eff}}}M_V^2 + \frac{C_b}{n_{\text{eff}}} \tag{160}$$

with $C_1 = ((1 + \gamma)M_k + \lambda)^2$.

**Bounding the slow drift error term** $\langle w_{\mathsf{V},t} - w_{\mathsf{V},t}^\star, w_{\mathsf{V},t}^\star - w_{\mathsf{V},t+1}^\star\rangle_{\mathcal{H}_k}$. In two time-scales, the Actor update satisfies $\|h_{t+1} - h_t\| \leq G_h \alpha_t$ for some bound $G_h$ on the natural/vanilla policy gradient step (this follows from bounded score function and step-size choice). Hence

$$\|w_{\mathsf{V},t}^\star - w_{\mathsf{V},t+1}^\star\|_{\mathcal{H}_k} \leq L_{\mathsf{V}} G_h \alpha_t^{\mathsf{h}}. \tag{161}$$

By Cauchy-Schwarz and the Young inequality,

$$\langle w_{\mathsf{V},t} - w_{\mathsf{V},t}^\star, w_{\mathsf{V},t}^\star - w_{\mathsf{V},t+1}^\star\rangle_{\mathcal{H}_k} \leq \|w_{\mathsf{V},t} - w_{\mathsf{V},t}^\star\|_{\mathcal{H}_k} \|w_{\mathsf{V},t}^\star - w_{\mathsf{V},t+1}^\star\|_{\mathcal{H}_k}$$

$$\leq \|w_{\mathsf{V},t} - w_{\mathsf{V},t}^\star\|_{\mathcal{H}_k} L_{\mathsf{V}} G_h \alpha_t^{\mathsf{h}} \leq \left(\|w_{\mathsf{V},t} - w_{\mathsf{V},t}^\star\|_{\mathcal{H}_k}^2 + 1\right) L_{\mathsf{V}} G_h \alpha_t^{\mathsf{h}}. \tag{162}$$

Therefore, the slow drift error term, which is caused by the two-scale nature of the algorithm, produces a slow-drift penalty $O(\alpha_t^{\mathsf{h}})$ and diminishes as the tracking error diminishes.

**Bounding the bias error term** $\langle \hat{g}_t(w_{\mathsf{V},t}), w_{\mathsf{V},t} - w_{\mathsf{V},t}^\star\rangle_{\mathcal{H}_k}$. To begin with, we decompose the bias error term as

$$\langle \hat{g}_t(w_{\mathsf{V},t}), w_{\mathsf{V},t} - w_{\mathsf{V},t}^\star\rangle_{\mathcal{H}_k} \leq \|\hat{g}_t(w_{\mathsf{V},t})\|_{\mathcal{H}_k} \|w_{\mathsf{V},t} - w_{\mathsf{V},t}^\star\|_{\mathcal{H}_k}$$

$$\leq \frac{1}{2}\|\hat{g}_t(w_{\mathsf{V},t})\|_{\mathcal{H}_k}^2 + \frac{1}{2}\|w_{\mathsf{V},t} - w_{\mathsf{V},t}^\star\|_{\mathcal{H}_k}^2$$

$$\leq \frac{1}{2}\left(\left(C_1 + \frac{C_B}{n_{\text{eff}}}\right)\|w_{\mathsf{V},t} - w_{\mathsf{V},t}^\star\|_{\mathcal{H}_k}^2 + \frac{C_B}{n_{\text{eff}}}M_V^2 + \frac{C_b}{n_{\text{eff}}}\right) + \frac{1}{2}\|w_{\mathsf{V},t} - w_{\mathsf{V},t}^\star\|_{\mathcal{H}_k}^2$$

$$= \frac{1}{2}\left(C_1 + \frac{C_B}{n_{\text{eff}}} + 1\right)\|w_{\mathsf{V},t} - w_{\mathsf{V},t}^\star\|_{\mathcal{H}_k}^2 + \frac{1}{2}\left(\frac{C_B}{n_{\text{eff}}}M_V^2 + \frac{C_b}{n_{\text{eff}}}\right), \tag{163}$$

where the last inequality comes from the results in (160).

**Bounding the fixed-point drift error term** $\langle \hat{g}_t(w_{\mathsf{V},t}), w_{\mathsf{V},t}^\star - w_{\mathsf{V},t+1}^\star\rangle_{\mathcal{H}_k}$ **and** $\left\|w_{\mathsf{V},t}^\star - w_{\mathsf{V},t+1}^\star\right\|_{\mathcal{H}_k}^2$. On the one hand, we bound $\langle \hat{g}_t(w_{\mathsf{V},t}), w_{\mathsf{V},t}^\star - w_{\mathsf{V},t+1}^\star\rangle_{\mathcal{H}_k}$ based on the results above as follows.

$$\langle \hat{g}_t(w_{\mathsf{V},t}), w_{\mathsf{V},t}^\star - w_{\mathsf{V},t+1}^\star\rangle_{\mathcal{H}_k} \leq \|\hat{g}_t(w_{\mathsf{V},t})\|_{\mathcal{H}_k} \|w_{\mathsf{V},t}^\star - w_{\mathsf{V},t+1}^\star\|_{\mathcal{H}_k}$$

$$\leq \frac{1}{2}\left(\|\hat{g}_t(w_{\mathsf{V},t})\|_{\mathcal{H}_k}^2 + 1\right)\|w_{\mathsf{V},t}^\star - w_{\mathsf{V},t+1}^\star\|_{\mathcal{H}_k}$$

$$\leq \frac{1}{2}\left(\left(C_1 + \frac{C_B}{n_{\text{eff}}}\right)\|w_{\mathsf{V},t} - w_{\mathsf{V},t}^\star\|_{\mathcal{H}_k}^2 + \frac{C_B}{n_{\text{eff}}}M_V^2 + \frac{C_b}{n_{\text{eff}}} + 1\right) L_{\mathsf{V}} G_h \alpha_t^{\mathsf{h}}$$

$$= \frac{1}{2}L_{\mathsf{V}} G_h \alpha_t^{\mathsf{h}}\left(C_1 + \frac{C_B}{n_{\text{eff}}}\right)\|w_{\mathsf{V},t} - w_{\mathsf{V},t}^\star\|_{\mathcal{H}_k}^2 + \frac{1}{2}\left(\frac{C_B}{n_{\text{eff}}}M_V^2 + \frac{C_b}{n_{\text{eff}}} + 1\right) L_{\mathsf{V}} G_h \alpha_t^{\mathsf{h}}, \tag{164}$$

where the last line holds due to Eqs. (160) and (161).

On the other hand, we bound $\left\|w_{\mathsf{V},t}^\star - w_{\mathsf{V},t+1}^\star\right\|_{\mathcal{H}_k}^2 \leq L_{\mathsf{V}}^2 G_h^2 (\alpha_t^{\mathsf{h}})^2$ directly from (161).

**Summing up these results.** Let $e_t := w_{\mathsf{V},t} - w^\star_{\mathsf{V},t}$. Assume the regularised population operator $B_\lambda := \Gamma_s - \gamma \Gamma_{2s} + \lambda \mathbf{I}$ is $\lambda$-strongly monotone, i.e., $\langle B_\lambda x, x \rangle \geq \lambda \|x\|^2$. Summing up the results in each term, we obtain, conditioning on $\mathcal{F}_{t-\tau_t}$,

$$
\mathbb{E}\left[\|e_{t+1}\|^2_{\mathcal{H}_\psi} \mid \mathcal{F}_{t-\tau_t}\right] \leq \|e_t\|^2_{\mathcal{H}_\psi} + M_k q_{\mathsf{A}}^2 \varepsilon_{\mathsf{PV}}^2 + (\alpha_t^{\mathsf{v}})^2 \left(\left(C_1 + \frac{C_B}{n_{\text{eff}}}\right)\|e_t\|^2_{\mathcal{H}_\psi} + \frac{C_B}{n_{\text{eff}}}M_V^2 + \frac{C_b}{n_{\text{eff}}}\right) + L_{\mathsf{V}}^2 G_h^2 (\alpha_t^{\mathsf{h}})^2
$$

$$
+ 2\alpha_t^{\mathsf{v}}\left(\frac{1}{2}\left(C_1 + \frac{C_B}{n_{\text{eff}}} + 1\right)\|e_t\|^2_{\mathcal{H}_\psi} + \frac{1}{2}\left(\frac{C_B}{n_{\text{eff}}}M_V^2 + \frac{C_b}{n_{\text{eff}}}\right)\right) + \left(\|e_t\|^2_{\mathcal{H}_\psi} + 1\right) L_{\mathsf{V}} G_h \alpha_t^{\mathsf{h}}
$$

$$
+ 2\alpha_t^{\mathsf{v}}\left(\frac{1}{2}L_{\mathsf{V}} G_h \alpha_t^{\mathsf{h}}\left(C_1 + \frac{C_B}{n_{\text{eff}}}\right)\|e_t\|^2_{\mathcal{H}_\psi} + \frac{1}{2}\left(\frac{C_B}{n_{\text{eff}}}M_V^2 + \frac{C_b}{n_{\text{eff}}} + 1\right) L_{\mathsf{V}} G_h \alpha_t^{\mathsf{h}}\right)
$$

$$
= \left(1 - 2\lambda\alpha_t^{\mathsf{v}} + c_B(\alpha_t^{\mathsf{v}})^2 + c_\times \alpha_t^{\mathsf{v}}\alpha_t^{\mathsf{h}} + c_M \alpha_t^{\mathsf{h}}\right)\|e_t\|^2_{\mathcal{H}_\psi} + B_t, \tag{165}
$$

where the coefficients are collected as

$$
c_B := C_1 + \frac{C_B}{n_{\text{eff}}}, \qquad c_\times := \left(C_1 + \frac{C_B}{n_{\text{eff}}}\right) L_{\mathsf{V}} G_h, \qquad c_M := L_{\mathsf{V}} G_h,
$$

$$
B_t := M_k q_{\mathsf{V}}^2 \varepsilon_{\mathsf{PV}}^2 + (\alpha_t^{\mathsf{v}} + (\alpha_t^{\mathsf{v}})^2)\left(\frac{C_B}{n_{\text{eff}}}M_V^2 + \frac{C_b}{n_{\text{eff}}}\right) + \alpha_t^{\mathsf{v}}\alpha_t^{\mathsf{h}} L_{\mathsf{V}} G_h \left(\frac{C_B}{n_{\text{eff}}}M_V^2 + \frac{C_b}{n_{\text{eff}}} + 1\right) + L_{\mathsf{V}} G_h \alpha_t^{\mathsf{h}} + L_{\mathsf{V}}^2 G_h^2 (\alpha_t^{\mathsf{h}})^2.
$$

Define the contraction coefficient

$$
\eta_t := 2\lambda\alpha_t^{\mathsf{v}} - c_B(\alpha_t^{\mathsf{v}})^2 - c_\times \alpha_t^{\mathsf{v}}\alpha_t^{\mathsf{h}} - c_M \alpha_t^{\mathsf{h}}.
$$

Hence, Eq. (165) becomes the standard supermartingale-type recursion

$$
\mathbb{E}\left[\|e_{t+1}\|^2 \mid \mathcal{F}_{t-\tau_t}\right] \leq (1 - \eta_t)\|e_t\|^2_{\mathcal{H}_\psi} + B_t \leq \Pi_{i=\hat{t}}^t (1 - \eta_i)\|e_{\hat{t}}\|^2_{\mathcal{H}_\psi} + \sum_{i=\hat{t}}^t \left[\Pi_{j=i+1}^t (1 - \eta_j)\right] B_i, \tag{166}
$$

where we have used that $\eta_i, B_i$ are $\mathcal{F}_{i-\tau_i}$-measurable (hence independent of the inner conditional expectation at step $i$). We now bound the two terms on the right-hand side.

Assume the two time-scale stepsizes $\alpha_t^{\mathsf{v}} = \frac{C_{\mathsf{v}}}{(t+1)^\nu}$, $\alpha_t^{\mathsf{h}} = \frac{C_{\mathsf{h}}}{(t+1)^\sigma}$ with $0 < \nu < \sigma \leq 1$. By definition,

$$
\eta_t = 2\lambda\alpha_t^{\mathsf{v}} - c_B(\alpha_t^{\mathsf{v}})^2 - c_\times \alpha_t^{\mathsf{v}}\alpha_t^{\mathsf{h}} - c_M \alpha_t^{\mathsf{h}} = \frac{2\lambda C_{\mathsf{v}}}{(t+1)^\nu} - \frac{c_B C_{\mathsf{v}}^2}{(t+1)^{2\nu}} - \frac{c_\times C_{\mathsf{v}} C_{\mathsf{h}}}{(t+1)^{\nu+\sigma}} - \frac{c_M C_{\mathsf{h}}}{(t+1)^\sigma}.
$$

Hence, there exists $T_0 < \infty$ and $\tilde{c} \in (0, \lambda C_{\mathsf{v}}]$ such that for all $t \geq T_0$,

$$
\eta_t \geq \frac{\tilde{c}}{(t+1)^\nu}. \tag{167}
$$

Using $\log(1-x) \leq -x$ for $x \in (0,1)$ and (167),

$$
\prod_{i=\hat{t}}^{t-1}(1 - \eta_i) \leq \exp\left(-\sum_{i=\hat{t}}^{t-1}\eta_i\right) \leq \exp\left(-\tilde{c}\sum_{i=\hat{t}}^{t-1}\frac{1}{(i+1)^\nu}\right). \tag{168}
$$

If $0 < \nu < 1$, then $\sum_{i=\hat{t}}^{t-1}(i+1)^{-\nu} \geq \frac{(t+1)^{1-\nu}-(\hat{t}+1)^{1-\nu}}{1-\nu}$, so

$$
\prod_{i=\hat{t}}^{t-1}(1 - \eta_i) \leq \exp\left(-\frac{\tilde{c}}{1-\nu}\left[(t+1)^{1-\nu} - (\hat{t}+1)^{1-\nu}\right]\right). \tag{169}
$$

When $\nu = 1$, we obtain $\prod_{i=\hat{t}}^{t-1}(1 - \eta_i) \leq \left(\frac{\hat{t}+1}{t+1}\right)^{\tilde{c}}$. In both cases, the product decays at least polynomially.

Write $B_i$ in the separated form

$$B_i = \underbrace{M_k q_V^2 \varepsilon_{\text{PV}}^2 + \frac{C_b}{n_{\text{eff}}}}_{b_0} + \underbrace{\frac{C_B}{n_{\text{eff}}} M_V^2 \, \alpha_i^{\text{v}}}_{b_1} + \underbrace{\left( \frac{C_B}{n_{\text{eff}}} \frac{C_B}{n_{\text{eff}}} M_V^2 + \frac{C_b}{n_{\text{eff}}} + 1 \right) L_V G_h \, \alpha_i^{\text{v}} \alpha_i^{\text{h}}}_{b_2} + \underbrace{L_V G_h \, \alpha_i^{\text{h}}}_{b_3}$$

$$+ \underbrace{\left( \frac{C_B}{n_{\text{eff}}} M_V^2 + \frac{C_b}{n_{\text{eff}}} \right) (\alpha_i^{\text{v}})^2}_{b_4} + \underbrace{L_V^2 G_h^2 (\alpha_i^{\text{h}})^2}_{b_5}. \tag{170}$$

Define the weights $W_{i,t} := \prod_{j=i+1}^{t-1} (1 - \eta_j)$. Using the same argument as in (169),

$$W_{i,t} \le \exp\left( -\frac{\tilde{c}}{1-\nu} \left[ (t+1)^{1-\nu} - (i+1)^{1-\nu} \right] \right). \tag{171}$$

Next, we ignore constant factors and focus solely on the decay induced by the learning rate. For ease of exposition, we take the step size to follow the schedule $(i+1)^{-\rho}$.

We will repeatedly use the following calculus-type estimate (change variable $u = (i+1)^{1-\nu}$, compare sum with integral): for any $\rho > 0$ there exist finite constants $C_1(\rho), C_2(\rho)$ (independent of $t$) such that for $\rho > \nu$

$$\sum_{i=\hat{t}}^{t-1} W_{i,t} \frac{1}{(i+1)^{\rho}} \le C_1(\rho) \exp\left( -\frac{\tilde{c}}{2(1-\nu)} \left[ (t+1)^{1-\nu} - (\hat{t}+1)^{1-\nu} \right] \right) + \frac{C_2(\rho)}{(t+1)^{\rho-\nu}}), \tag{172}$$

and for $\rho = \nu$ the second term becomes $\frac{K_2(\nu) \log(t+1)}{(t+1)^{\epsilon}}$ for any fixed $\epsilon \in (0, \nu)$ (or equivalently a log-loss). Applying (172) to each summand in $B_i$ with

$$\alpha_i^{\text{v}} = \frac{C_{\text{v}}}{(i+1)^{\nu}}, \ \alpha_i^{\text{h}} = \frac{C_{\text{h}}}{(i+1)^{\sigma}}, \ \alpha_i^{\text{v}} \alpha_i^{\text{h}} = \frac{C_{\text{v}} C_{\text{h}}}{(i+1)^{\nu+\sigma}}, \ (\alpha_i^{\text{v}})^2 = \frac{C_{\text{v}}^2}{(i+1)^{2\nu}}, \ (\alpha_i^{\text{h}})^2 = \frac{C_{\text{h}}^2}{(i+1)^{2\sigma}},$$

and noting $0 < \nu < \sigma$, we obtain that there exists constants $\bar{C}_0, \ldots, \bar{C}_5 < \infty$ (depending only on $\tilde{c}, \nu, \sigma, C_{\text{h}}, C_{\text{v}}$ and $b_\ell$'s) such that

$$\sum_{i=\hat{t}}^{t-1} W_{i,t} B_i \le \bar{C}_0 \exp\left( -\frac{\tilde{c}}{2(1-\nu)} \left[ (t+1)^{1-\nu} - (\hat{t}+1)^{1-\nu} \right] \right)$$

$$+ \frac{\bar{C}_1}{(t+1)^{\nu}} + \frac{\bar{C}_2}{(t+1)^{\nu+\sigma-\nu}} + \frac{\bar{C}_3}{(t+1)^{\sigma-\nu}} + \frac{\bar{C}_4}{(t+1)^{2\nu-\nu}} + \frac{\bar{C}_5}{(t+1)^{2\sigma-\nu}} + b_0^{\star}$$

$$\le \bar{C}_0 \exp\left( -\frac{\tilde{c}}{2(1-\nu)} \left[ (t+1)^{1-\nu} - (\hat{t}+1)^{1-\nu} \right] \right) + \frac{C_6}{(t+1)^{\nu}} + b_0^{\star}, \tag{173}$$

where in the last inequality we use $\sigma > \nu$ so that the slowest-decaying polynomial term is $(t+1)^{-\nu}$, and we set $b_0^{\star} := M_k q_V^2 \varepsilon_{\text{PV}}^2 + \frac{C_b}{n_{\text{eff}}}$ (the non-vanishing bias floor).

Plugging (169) and (173) into (166) yields, for $t \ge T_0$,

$$\mathbb{E}\left[ \|e_t\|_{\mathcal{H}_\psi}^2 \right] \le \exp\left( -\frac{\tilde{c}}{1-\nu} \left[ (t+1)^{1-\nu} - (\hat{t}+1)^{1-\nu} \right] \right) \mathbb{E} \|e_{\hat{t}}\|_{\mathcal{H}_\psi}^2$$

$$+ \bar{C}_0 \exp\left( -\frac{\tilde{c}}{2(1-\nu)} \left[ (t+1)^{1-\nu} - (\hat{t}+1)^{1-\nu} \right] \right) + \frac{C_6}{(t+1)^{\nu}} + b_0^{\star}.$$

As a result, we conclude that there exist finite constants $\widetilde{C}$ and $\widetilde{C}_b$ such that

$$\mathbb{E}\left[ \|e_t\|_{\mathcal{H}_\psi}^2 \right] \le \frac{\widetilde{C}}{(t+1)^{\nu}} + M_k q_V^2 \varepsilon_{\text{PV}}^2 + \frac{\widetilde{C}_b}{n_{\text{eff}}}, \qquad t \ge T_0,$$

which is the decay bound under the two time-scale stepsizes with $\sigma > \nu$. The borderline case $\nu = 1$ follows similarly, replacing (169) and (172) by their logarithmic counterparts, incurring at most a polylogarithmic loss.

**Step 3: Recursively refining tracking error bound**  Starting from the one-step supermartingale recursion

$$\mathbb{E}\left[\|e_{t+1}\|^2_{\mathcal{H}_\psi} \mid \mathcal{F}_{t-\tau_t}\right] \le (1-\eta_t)\|e_t\|^2_{\mathcal{H}_\psi} + B_t + \underbrace{2\langle e_t, w^\star_{\mathsf{V},t} - w^\star_{\mathsf{V},t+1}\rangle_{\mathcal{H}_k}}_{\text{slow-drift term}}. \tag{174}$$

Ignoring the slow-drift term in (174) and unrolling via the discrete Grönwall inequality, we obtain

$$\mathbb{E}\left[\|e_t\|^2_{\mathcal{H}_\psi}\right] \le \exp\left(-\tfrac{\tilde{c}}{1-\nu}\left[(t+1)^{1-\nu} - (\hat{t}+1)^{1-\nu}\right]\right)\mathbb{E}\|e_{\hat{t}}\|^2_{\mathcal{H}_\psi} + \sum_{i=\hat{t}}^{t-1} W_{i,t}\mathbb{E}\left[B_i\right], \tag{175}$$

where

$$W_{i,t} = \prod_{j=i+1}^{t-1}(1-\eta_j) \le \exp\left(-\tfrac{\tilde{c}}{1-\nu}[(t+1)^{1-\nu} - (i+1)^{1-\nu}]\right).$$

$B_i$ is built from linear, quadratic, and cross terms of $\alpha^\mathsf{v}_i$ and $\alpha^\mathsf{h}_i$, plus a bias floor. We estimate the sums by comparing them with integrals via the substitution $u = (i+1)^{1-\nu}$. For large $t$, this gives finite constants $D_0, D_b$ that do not depend on $t$ as follows.

$$\mathbb{E}\left[\|e_t\|^2_{\mathcal{H}_\psi}\right] \le \frac{D_0}{(t+1)^\nu} + M_k q^2_\mathsf{V}\varepsilon^2_{\mathsf{PV}} + \frac{D_b}{n_{\text{eff}}}, \tag{176}$$

i.e., an initial polynomial decay at rate $(t+1)^{-\nu}$ up to the irreducible bias floor.

By Lipschitz continuity of the population fixed point in the Actor parameter on the slow time-scale, there exists $L_\mathsf{V} < \infty$ such that

$$\|w^\star_{\mathsf{V},t+1} - w^\star_{\mathsf{V},t}\|_{\mathcal{H}_\psi} \le L_\mathsf{V}\|h_{t+1} - h_t\| \le L_\mathsf{V}G_h\alpha^\mathsf{h}_t = \frac{L_\mathsf{V}G_hC_\alpha}{(t+1)^\sigma}. \tag{177}$$

Conditioning on $\mathcal{F}_{t-\tau_t}$ and using Cauchy-Schwarz,

$$\begin{aligned}\mathbb{E}\big[2\langle e_t, w^\star_{\mathsf{V},t} - w^\star_{\mathsf{V},t+1}\rangle_{\mathcal{H}_k} \mid \mathcal{F}_{t-\tau_t}\big] &\le 2\sqrt{\mathbb{E}\left[\|e_t\|^2_{\mathcal{H}_\psi}\right]} \cdot \frac{L_\mathsf{V}G_hC_\alpha}{(t+1)^\sigma} \\ &\le \frac{C_{\text{sd}}}{(t+1)^{\sigma+\nu/2}} + \frac{C_{\text{sd}}}{(t+1)^\sigma}\left(M^{1/2}_k q_\mathsf{V}\varepsilon_{\mathsf{PV}} + n^{-1/2}_{\text{eff}}\right),\end{aligned} \tag{178}$$

where (176) was used in the last inequality.

Returning to (174), taking total expectation and unrolling as before yields (175) with an augmented noise term

$$\tilde{B}_t := B_t + \frac{C_{\text{sd}}}{(t+1)^{\sigma+\nu/2}} + \frac{C_{\text{sd}}}{(t+1)^\sigma}\left(M^{1/2}_k q_\mathsf{V}\varepsilon_{\mathsf{PV}} + n^{-1/2}_{\text{eff}}\right).$$

Applying the same weighted-sum estimate gives

$$\mathbb{E}\left[\|e_t\|^2_{\mathcal{H}_\psi}\right] \le \frac{D_1}{(t+1)^\nu} + \frac{D_2}{(t+1)^{\sigma-\nu}} + \frac{D_3}{(t+1)^{\sigma+\nu/2-\nu}} + M_k q^2_\mathsf{V}\varepsilon^2_{\mathsf{PV}} + \frac{D'_b}{n_{\text{eff}}}. \tag{179}$$

- **Fast slow-time scale:** $\sigma > \frac{3}{2}\nu.$

$$\mathbb{E}\left[\|e_t\|^2_{\mathcal{H}_\psi}\right] \le \frac{C}{(t+1)^\nu} + M_k q^2_\mathsf{V}\varepsilon^2_{\mathsf{PV}} + \frac{C_b}{n_{\text{eff}}}.$$

- **Critical case:** $\sigma = \frac{3}{2}\nu.$

$$\mathbb{E}\left[\|e_t\|^2_{\mathcal{H}_\psi}\right] \le \frac{C_{\text{crit}}\log^2(t+1)}{(t+1)^\nu} + M_k q^2_\mathsf{V}\varepsilon^2_{\mathsf{PV}} + \frac{C_b}{n_{\text{eff}}}.$$

- **Slow slow-time scale:** $\nu < \sigma < \frac{3}{2}\nu.$

$$\mathbb{E}\left[\|e_t\|^2_{\mathcal{H}_\psi}\right] \le \frac{C}{(t+1)^{2(\sigma-\nu)}} + M_k q^2_\mathsf{V}\varepsilon^2_{\mathsf{PV}} + \frac{C_b}{n_{\text{eff}}}.$$

Combining the above cases, we obtain the unified bound for large $t$ as follows. Let $\nu \in (0, 1)$ and $\sigma > \nu$ be the Critic and Actor step-size exponents, respectively. The tracking error satisfies

$$\mathbb{E}\left[\left\|w_{\mathsf{V},t} - w_{\mathsf{V},t}^\star\right\|_{\mathcal{H}_\psi}^2\right] = \mathbb{E}\left[\|e_t\|_{\mathcal{H}_\psi}^2\right] \leq \begin{cases} \frac{C}{(t+1)^\nu} + M_k q_{\mathsf{V}}^2 \varepsilon_{\mathsf{PV}}^2 + \frac{C_b}{n_{\mathrm{eff}}}, & \sigma > \frac{3}{2}\nu, \\[2mm] \frac{C_{\mathrm{crit}} \log^2(t+1)}{(t+1)^\nu} + M_k q_{\mathsf{V}}^2 \varepsilon_{\mathsf{PV}}^2 + \frac{C_b}{n_{\mathrm{eff}}}, & \sigma = \frac{3}{2}\nu, \\[2mm] \frac{C}{(t+1)^{2(\sigma-\nu)}} + M_k q_{\mathsf{V}}^2 \varepsilon_{\mathsf{PV}}^2 + \frac{C_b}{n_{\mathrm{eff}}}, & \nu < \sigma < \frac{3}{2}\nu, \end{cases} \tag{180}$$

for some finite constants $C, C_{\mathrm{crit}}$, and $C_b$ independent of $t$.

