# OpenReview forum: "SHAP-Guided Kernel Actor-Critic for Explainable Reinforcement Learning"
_ICML.cc/2026/Conference — ICML 2026 regular_

### Official Review · Reviewer_78GA · 2026-03-10

**Soundness:** 3
**Presentation:** 2
**Significance:** 3
**Originality:** 3
**Overall Recommendation:** 4
**Confidence:** 4

**Summary:**

This paper proposes a kernel actor-critic method that integrates SHAP-based state attribution into policy learning, with the goal of improving both training stability and interpretability in continuous-control reinforcement learning. The approach uses attribution scores derived from a value critic to adapt the actor kernel, yielding two variants based on conditional and marginal expectation operators. The paper provides theoretical analysis including convergence guarantees and robustness to bounded perturbations. Empirical evaluations in continuous-control benchmarks show stable and competitive performance, particularly in lower-dimensional environments.

**Compliance With Llm Reviewing Policy:**

Affirmed.

**Final Justification:**

The paper presents an interesting and technically ambitious contribution at the intersection of kernel methods, explainability, and reinforcement learning. I found the core idea—using SHAP-style state attributions not only for post-hoc analysis but to guide policy optimization—to be reasonably original and potentially impactful for follow-up work on attribution-aware learning. The paper is further strengthened by a substantial theoretical component, including convergence and robustness guarantees under perturbations, and by empirical evidence showing stable and competitive performance, particularly in lower-dimensional continuous-control settings.

My initial concerns were mainly about framing, clarity, and practical scope rather than the core technical validity of the approach. In particular, I felt that the original manuscript somewhat overstated the explainable RL angle, while the empirical results more directly supported an explainability-informed optimization perspective. I was also concerned that several important design choices—such as the KME/CME distinction, the two-critic two-timescale structure, the role of the dictionary mechanism, and the scope of the theoretical assumptions—were not explained clearly enough in the paper, and that the lack of an explicit limitations discussion weakened the presentation.

The rebuttal addressed these concerns well. Most importantly, the authors clarified the intended positioning of the work as an attribution-aware or explainability-informed RL optimization framework rather than a fully general explainable RL system. This significantly improved the alignment between the claims and the evidence. I also found the rebuttal helpful in clarifying the practical difference between KME and CME, motivating the architectural design, explaining how computational efficiency is achieved through sparse dictionaries, and acknowledging the limitations of kernel locality and high-dimensional scalability. The commitment to revise the manuscript accordingly, especially by improving the presentation and adding an explicit limitations discussion, strengthens my confidence in the final version.

Overall, while the empirical scope is still somewhat limited and the strongest gains appear in lower-dimensional settings, I believe the paper makes a technically solid and reasonably original contribution that others in the area may build on. Since the rebuttal resolved my main concerns and substantially improved the clarity and positioning of the work, I raised my recommendation from 3 to 4.

**Key Questions For Authors:**

1. Explainability claim: In what precise sense does the proposed method make the learned policy more interpretable, beyond using SHAP values internally during optimization? A clearer answer here would affect my assessment of how well the paper delivers on its title and main framing.
2. CME vs. KME guidance: Could the authors better explain the practical and conceptual differences between the CME and KME variants, especially when one should be preferred over the other? This would improve both clarity and usability of the method.
3. Efficiency and scalability: How is the method able to attain competitive computational efficiency despite relying on kernel components and dynamic dictionaries, and how should one expect it to scale with state dimension and sample size? A clearer answer here could strengthen the practical significance of the work.
4. Architecture design: Why is the two-critic / two-timescale structure necessary in this setting, and is there evidence that a simpler single-critic variant would be insufficient? An ablation or stronger motivation would improve confidence in the design choices.
5. Limitations and assumptions of the theory: The paper provides extensive theoretical analysis but does not discuss its limitations. Could the authors elaborate on which assumptions underlying the theoretical results are primarily technical (e.g., used for analytical tractability) and which are expected to be critical in practice? Clarifying this would help readers better understand how well the theoretical guarantees might transfer to more complex or higher-dimensional RL settings.

**Limitations:**

No. The paper currently does not discuss limitations at all, neither in the conclusion nor in the broader impact section. This should be addressed. In particular, the authors should discuss limitations related to scalability to higher-dimensional settings, dependence on kernel and dictionary design, and the computational implications of using kernel methods in RL. It would also be helpful to explicitly reflect on limitations arising from the theoretical assumptions used in the analysis and how these may affect practical applicability. Finally, the paper could more directly discuss the current gap between using explainability concepts to improve learning stability and demonstrating explainable RL in a broader sense.

**Strengths And Weaknesses:**

This paper presents a meaningful contribution at the intersection of kernel methods, explainability, and reinforcement learning. The central idea—using SHAP-style state attributions not only for post-hoc interpretation but as part of the policy optimization process—is interesting and could inspire follow-up work on explanation-aware learning algorithms. The work is technically ambitious and includes a detailed theoretical analysis that goes beyond standard convergence guarantees by additionally providing robustness results under perturbations. This theoretical framing is valuable and provides a principled perspective on how attribution-aware learning affects policy optimization. The empirical results also demonstrate that the proposed method can train stably and achieve competitive performance in several continuous-control benchmarks.

However, several aspects limit the current impact of the work.

First, the paper’s framing around explainable reinforcement learning somewhat overstates what is demonstrated empirically. In its current form, the paper primarily shows how explainability concepts (SHAP-based attribution) can be used to improve training stability of reinforcement learning. While attribution values are computed and incorporated into learning, the paper does not yet convincingly demonstrate how the resulting policies become more interpretable to users beyond exposing attribution scores. As a result, the work currently reads more as explainability-informed RL optimization rather than a fully developed explainable RL framework.

Second, the empirical evaluation is somewhat limited relative to the scope of the claims. The main results focus on a relatively small continuous-control setting, while additional experiments are deferred to the appendix. While the method demonstrates stable and competitive performance, it does not outperform strong baselines such as SAC in final performance across the presented benchmarks. In particular, according to Table 2, SAC still exhibits superior performance across perturbation strengths, which should be reflected more clearly in the discussion of the results. The comparisons remain informative as evidence of training stability and robustness, but the narrative describing the results should be slightly more aligned with the quantitative outcomes. For improved readability, the tables could also highlight best-performing values (e.g., using bold formatting).

Third, the presentation could be improved substantially. Some core concepts appear later than they should, especially the idea of maintaining a dynamically adapted subset of states (a dictionary) used to compute the policy kernel. Introducing this mechanism earlier would make the overall method much easier to understand, particularly since it plays a central role in the algorithm’s scalability and learning dynamics. It may also be helpful to relate these dictionaries conceptually to replay buffers used in standard RL, clarifying their role and differences. Some notation is also inconsistent or introduced without sufficient explanation. For example, the paper initially introduces different dictionary sizes q_A and q_V (l. 245), but later refers more generically to a single size q (e.g., l. 306), leaving it unclear whether these quantities are meant to differ in practice. Similarly, introducing the symbol m (l. 252), presumably referring to the action dimensionality, earlier would improve readability.
Several other central concepts would benefit from clearer explanation and motivation. In particular, the use of Mahalanobis weighting within the kernel, the distinction between the CME and KME variants based on on- and off-manifold expectations, and the meaning of the “two-timescale” characterization of RSA2C are currently not sufficiently explained. The paper repeatedly refers to the method as a two-timescale approach, which appears to stem from the presence of both a value critic and an advantage critic, but the motivation for this design choice is not clearly discussed. It would be helpful to explain why this structure is necessary and whether a simpler single-critic architecture could suffice, ideally supported by an ablation study. As it stands, the architecture includes several components whose necessity is not fully justified, which makes the overall method harder to follow.

Fourth, the discussion of computational efficiency and scalability is insufficient. Kernel methods are generally considered less scalable with respect to sample size and dimensionality compared to neural network approaches. While the paper briefly mentions computational aspects in the preliminaries, it would benefit from a more explicit discussion of how the proposed method achieves competitive efficiency in practice and what approximations or design choices make this possible.

Finally, the paper does not discuss limitations of the proposed approach at all, neither in the conclusion nor in the impact statement. This is a significant omission. Given the strong theoretical component and the specific modeling assumptions used in the analysis, the paper should explicitly discuss potential limitations and practical constraints. In particular, the authors should comment on limitations arising from the theoretical assumptions as well as practical limitations such as scalability to higher-dimensional state spaces, dependence on kernel choice, and the trade-off between explainability signals and performance. Explicitly discussing such limitations would not weaken the paper but instead strengthen it by helping guide future research building on this work.

Overall, I view the paper as technically solid and reasonably original, with a well-motivated theoretical contribution and promising empirical evidence. However, limitations in presentation, evaluation breadth, and the lack of an explicit limitations discussion currently reduce its overall impact. Given the substantial amount of material relegated to the appendix, including both theoretical analysis and empirical results, the work might also benefit from being presented in a longer-form format where these components can be integrated more directly into the main narrative.

---

> ### Author Rebuttal · Authors · 2026-03-31
>
> Thank you for your insightful comments.
>
> **W1** The RKHS-SHAP used in RSA2C **preserves the key interpretability properties of classical SHAP**, including additive and consistent feature attribution. Compared to a standard SHAP-based explainable RL algorithm, RKHS-SHAP provides a more structured and stable estimation via kernel embeddings, leading to lower variance and improved reliability. Moreover, these attribution signals are not used solely for post-hoc explanation, but are directly incorporated into the learning process through the Mahalanobis-weighted kernel, which modulates both the Actor and Advantage Critic updates. As a result, feature importance not only explains the learned policy but also actively shapes it during optimization. Therefore, RSA2C can be more precisely viewed as an explainability-informed RL optimization framework rather than a fully developed explainable RL system.
>
> **W2** Concerning performance comparison, the results in Table 2 show that SAC achieves higher final returns across several perturbation strengths. To better align the narrative with the empirical findings, the revised manuscript will clarify that the primary advantage of RSA2C lies in improved training stability and robustness, rather than consistently outperforming strong baselines in final performance.
>
> **W3+W4** We will revise the manuscript to introduce the dictionary mechanism earlier and clarify its role and its relation to replay buffers, while also unifying notation (e.g., ($q_A$, $q_V$, $q$, $m$)) to improve readability. We will also strengthen the motivation and explanation of key components.
>
> **W5, Q1, and Q5** Please refer to Reviewer gtNj for details.
>
> **Q2** The key conceptual difference between KME and CME lies in how feature dependencies are modeled. KME captures marginal feature distributions and treats each dimension more independently, while CME models conditional relationships between features, allowing it to capture cross-dimensional dependencies. In practice, this leads to different behaviors. KME is simpler and computationally lighter, and works well when feature interactions are weak or the state space is relatively low-dimensional. In contrast, CME is more expressive and can better capture structured dependencies between features, which leads to more balanced and stable attribution, as observed in our experiments. Therefore, we recommend using CME when feature correlations are significant or when interpretability and stability are critical, while KME may be preferred in simpler settings due to its lower computational cost.
>
> **Q3** The computational efficiency of RSA2C stems from a dynamically maintained dictionary that enables sparse kernel representation. Instead of operating over all samples, both the Actor and Critic rely on a limited set of representative states, making the per-update cost depend on dictionary size rather than total samples.
>
> Empirically, we include additional results analyzing memory footprint as a function of dictionary size (see https://anonymous.4open.science/r/RSA2C-776C, evaluated with three seeds due to time constraints). These results show that both memory scales in a controlled manner, confirming that RSA2C avoids the prohibitive cost of naive kernel methods. The dictionary size remains bounded in practice due to ALD updates, preventing unbounded growth.
>
> Regarding scalability, the complexity with respect to sample size depends on the dictionary rather than total transitions, enabling efficient learning with large replay buffers. However, in high-dimensional spaces, stronger kernel locality requires larger dictionaries for sufficient coverage.
>
> **Q4** The **two-critic design** in RSA2C is introduced to separate two distinct roles that are coupled in RSA2C. The Value Critic is used for stable TD-based policy evaluation and serves as the source of RKHS-SHAP attribution. Meanwhile, the Advantage Critic is constructed with compatible features to provide low-variance policy-gradient estimation. Using a single critic may reduce stability and weaken the compatibility structure.
>
> As for the two-timescale update, since the SHAP attribution is computed from the Value Critic and then used to modulate the Actor and Advantage Critic updates, the critic must evolve on a faster timescale so that attribution remains sufficiently stable before the policy is updated. Otherwise, the value estimate, attribution, and policy may drift simultaneously, making both optimization and theoretical analysis less reliable.
>
> **Limitations** We will include a dedicated discussion of limitations in the revised manuscript, covering (i) scalability and computational challenges in high-dimensional settings due to kernel locality and dictionary coverage, and (ii) the scope and practical validity of the theoretical assumptions. We will also clarify the interpretability scope of RSA2C as an SHAP-based attribution measure and discuss its limitations, as detailed in our response to Reviewer 1.

---

> > ### Author Rebuttal · Reviewer_78GA · 2026-04-03
> >
> > Thank you for the thoughtful and detailed rebuttal. I consider my concerns adequately addressed.
> >
> >
> >
> > The rebuttal meaningfully clarifies the intended contribution and positioning of the work. In particular, I appreciate the clearer framing of the method as explainability-informed / attribution-aware RL optimization, and the clarification that the paper’s notion of interpretability is dimension-level attribution of policy-relevant state features rather than a broader user-facing explainable RL framework. This substantially addresses my main concern about the original framing.
> >
> >
> >
> > I also found the additional clarifications on the practical and conceptual distinction between KME and CME, the motivation for the two-critic / two-timescale design, the role of the dictionary mechanism in efficiency, and the scope of the theoretical assumptions to be helpful and responsive to my questions. The planned revisions to align the empirical discussion more carefully with the quantitative results, improve the presentation of the method, and add a clearer limitations discussion would further strengthen the paper.
> >
> >
> >
> > Overall, the rebuttal resolves my main concerns. I would encourage the authors to integrate these clarifications directly into the revised manuscript, especially the refined positioning of the contribution, the explanation of the KME/CME variants, the motivation for the architectural choices, and the explicit discussion of limitations and practical scope, since these additions would significantly improve clarity and impact for readers.

---

> > > ### Author Response · Authors · 2026-04-08
> > >
> > > Thank you again for your constructive feedback and for helping improve our manuscript. We truly appreciate your support. We would be glad to address any additional questions or comments you may have, and we will ensure that your suggestions are incorporated into the revised manuscript.

---

### Official Review · Reviewer_gCbH · 2026-03-11

**Soundness:** 3
**Presentation:** 4
**Significance:** 3
**Originality:** 2
**Overall Recommendation:** 4
**Confidence:** 3

**Summary:**

The paper proposes RSA2C, an attribution-aware actor–critic algorithm that incorporates feature interpretability into reinforcement learning optimization. The method represents the Actor in a vector-valued reproducing kernel Hilbert space (RKHS) with an operator-valued kernel, while the Value Critic and Advantage Critic are modeled in scalar RKHSs using sparsified dictionaries. The authors provide a theoretical analysis establishing a global non-asymptotic convergence bound under state perturbations, and empirically evaluate the approach on continuous-control environments, claiming improved stability, efficiency, and interpretability.

**Compliance With Llm Reviewing Policy:**

Affirmed.

**Final Justification:**

The rebuttal clarified several presentation issues, but it did not resolve my main concerns about the soundness and significance of the empirical results. In particular, I remain unconvinced by the efficiency comparisons against SAC, and the additional experiments still do not establish a clear, generalizable link between the proposed attribution mechanism and meaningful improvements in stability or performance across environments. As a result, I do not think the current empirical evidence adequately supports the paper’s central claims, so I am maintaining my original score.

Edit: In light of their most recent clarifications, the authors have addressed my main remaining concerns. Accordingly, I will be updating my review and raising my score to a 4.

**Key Questions For Authors:**

Why were the ablations excluded from the stability analysis under different scales of noise variance?

**Limitations:**

See weakness section regarding soundness of experimental results and associated claims.

**Strengths And Weaknesses:**

Disclaimer: I am not familiar enough with SHAP to assess the theoretical results presented in section 4. I will therefore focus on the other parts of the paper, and particularly on the soundness of the experimental results (section 5).

**Strengths**

The paper is well written, its contributions are clearly introduced.

The authors motivate well the research question being tackled and the importance of dimension-level attributions for XRL.

**Weaknesses**

**Misleading reporting of SAC results**. The authors repeateadly use misleading statements related to the performance of the SAC baseline:


1) At l377 (rhs), the authors state that "SAC reaches a relatively high initial return but quickly saturates" on Pendulum-v1, the primary environment used for experiments in the main text. This implies other methods would eventually overcome SAC in returns. Judging from Figure 3, SAC converges in ~80 epochs on Pendulum-v1. By 2000 epochs none of the methods have reached SAC's performance. Could it be that SAC simply solves the task quickly, instead of "saturating"?

2) The FLOPs and runtime reported in Table 1 appear to have been measured on the full 2000 epochs for all methods. Yet SAC converges in ~80 epochs. Therefore reporting at convergence would lover the FLOPs and runtime of SAC by a factor of ~25x . This actually would change SAC's runtime ranking from being last to best (!!!) among the methods tested, including the authors' proposed methods.

3) At l422 (lhs), the authors state "both SAC and PPO degrade substantially and remain less stable than the proposed RSA2C variants". This is misleading, while return variance is somewhat higher, the degradation experienced by SAC is comparable to the degradation experienced by RSA2C-CME (the authors' best performing algorithm). This is not acknowledged in the manuscript.

**It is unclear that RSA2C improves over its own ablations.** In the main text the authors rely on the results from the Pendulum-v2 environment to justify performance improvements of their proposed RSA2C algorithms over ablations (Advanced AC and RKHS-AC). Yet in additional environments tested in the appendix, RSA2C algorithms achieve similar (in BipedalWalker-v3) or worse return (Ant-v5) than advanced AC. The authors should acknowledge this in the main text.

**No clear connection between interpretability analysis and performance across environments.** In the main text, the authors use Heatmap and Beeswarm visualisations to justify that the importance accorded to different state dimensions over the course of training allows RSA2C-CME to achieve better sample efficiency and stability than RSA2C-KME in Pendulum-v1. However this pattern does not generalise to the two additional environments ("BipedalWalker-v3" and "Ant-v5") discussed in the appendix.

In these environments, each variant also present very distinct SHAP patterns across state dimensions.  Yet both variants achieve similar sample efficiency and overall returns. Under noise perturbations, R2A2C-CME is marginally more stable than R2A2C-KME in BipedalWalker, and marginally more unstable than R2A2C-KME in Ant. Yet the authors make the following statements in the corresponding evaluation on interpretability sections of each environments:

(For BipedalWalker)
>Overall, compared with KME, CME provides richer, direction-sensitive, and training-evolving explanatory signals, with a greater emphasis on the coupling mechanisms among state features.

(For Ant)
> This suggests that RSA2C-KME fails to capture the intricate dependencies among high-dimensional proprioceptive and contact features, resulting in weak and non-evolving explanatory signals. In contrast, RSA2C-CME yields substantially richer and more diverse attributions. [...] These structured and evolving patterns
indicate that RSA2C-CME successfully models feature correlations and adapts attribution as the policy improves.

To me, what this analysis shows is that **vastly different SHAP patterns across state dimensions result in no discernable changes in performance metrics for most of the environments tested**. This is highly problematic as it seems to invalidate the principal claims made by this work that their approach improves the efficiency, stability and interpretability of actor-critic methods.

While the research question is clearly important it appears the proposed method falls short of addressing it in a convincing way, at least based on the experimental results provided. Therefore I cannot recommend acceptance of this work in its current version.


Minor weaknesses and improvements:
- "FLOPs and runtime are significantly lower than those of SAC and PPO (even when executed on GPU)" -> This could be interpreted as stating that FLOPs change depending on whether an algorithm is executed on CPU/GPU, which is incorrect. Please reformulate to make clear that execution on CPU or GPU is relevant for runtime only.
- Using bold for (statistically significant) best method(s) in result tables would help the reader process the results.
- Fontsize is too small in most Figures. No explanation provided on what shaded area represent in Figure 2 and 3 (standard error or standard deviation, how many seeds?).
- Missing hyperparameter and network architectures employed for Deep-RL baselines.

---

> ### Author Rebuttal · Authors · 2026-03-31
>
> Thank you for highlighting these concerns.
>
> **Response to fairness and accuracy of comparisons** We have included additional experimental results to further clarify computational efficiency and scalability, see https://anonymous.4open.science/r/RSA2C-776C, which is only reported in three seeds due to the limited time.
>
> Regarding the learning dynamics on Pendulum-v1, the term "saturation" will be replaced with a more accurate description as: *SAC converges rapidly and achieves strong final performance early in training*.
> For stability under perturbations, the conclusion that SAC exhibits degradation comparable to RSA2C-CME is drawn directly from the results in Table 1. The corresponding discussion will be revised by explicitly stating that SAC shows comparable degradation trends under noise. For computational efficiency, the FLOPs and runtime reported in Table 1 were measured under a fixed training budget for all methods to ensure a consistent evaluation protocol. While this setup reflects total computational cost, it does not capture convergence speed. In addition, based on Figure 1 in the added material, we analyze dictionary size, memory footprint, and CPU memory usage, showing that memory scale in a controlled manner with the dictionary size. This confirms that RSA2C avoids the prohibitive cost of naive kernel methods through dictionary-based sparsification.
>
> **Response to performance improvement**
> From the results reported in Table 1 and Appendix A.4, RSA2C achieves clear gains in stability and sample efficiency on low-dimensional tasks, while in higher-dimensional environments, the performance becomes comparable to, or in some cases slightly below, strong baselines such as Advanced AC or SAC. To better understand this difference, we have included additional ablation results. In particular, the Advanced AC baseline corresponds to removing the SHAP-based weighting while keeping the RKHS formulation unchanged, and an additional variant with uniform SHAP weights (all ones) further isolates the role of attribution. These results show that SHAP-guided weighting consistently improves training stability, even in settings where final returns are similar across methods. Overall, the revised manuscript will clarify that RSA2C does not aim to uniformly outperform baselines in final return across all environments, but rather improves learning stability and behavior through attribution-guided optimization, with the most pronounced benefits in lower-dimensional or more structured settings.
>
> **Response to the relationship between interpretability and performance**
> The observations described by the reviewer are consistent with the results in the submitted manuscript: different SHAP patterns across environments do not necessarily lead to significant differences in final returns. This indicates that attribution alone does not directly determine asymptotic performance. The revised manuscript will explicitly state this point and avoid implying a direct causal link between richer attribution patterns and improved final performance.
>
> To further investigate this aspect, we have included additional ablation results. In particular, we introduce a variant with uniform SHAP weights (all ones), which effectively removes attribution guidance. The comparison shows that removing SHAP leads to reduced training stability and less structured learning dynamics, even in cases where the final returns are similar. This demonstrates that SHAP primarily influences the optimization process rather than directly improving final performance.
>
> Based on these results, the revised manuscript will clarify that the role of RKHS-SHAP in RSA2C is to provide structured attribution signals that guide learning, leading to improved stability and feature-level consistency.
>
> **Response to experimental completeness** We evaluate the stability of the ablated variants (including the Advanced AC baseline and the uniform SHAP weighting variant) under varying noise perturbations. These results show that removing SHAP guidance consistently leads to reduced stability compared to RSA2C, confirming that the attribution mechanism plays a key role in stabilizing the learning process. These results are now included to provide a more complete and consistent evaluation of the proposed method under perturbations.
>
> **Response to presentation** The statement regarding FLOPs and runtime will be corrected to distinguish between computational complexity (FLOPs) and hardware-dependent runtime. Result tables will be updated to highlight best-performing methods using bold formatting. Figure readability will be improved by increasing font sizes and explicitly describing shaded regions (including whether they represent standard deviation or standard error, and the number of seeds used). In addition, details of hyperparameters and network architectures for all baselines will be included to improve reproducibility.

---

> > ### Author Rebuttal · Reviewer_gCbH · 2026-04-03
> >
> > Thank you to the authors for the detailed rebuttal and for committing to improve the manuscript's presentation, particularly regarding the clarification of FLOPs/runtime wording, plot formatting, and correcting the misleading claims about SAC's performance degradation.
> >
> > However, after carefully reviewing the response and the newly provided experimental results, my primary concerns regarding the experimental soundness of the paper remain unresolved. Furthermore, the new results provided in the rebuttal draft unfortunately do not address my main critiques, and in some cases, introduce additional confusion by making claims that are difficult to substantiate with the provided data.
> >
> > **On computational efficiency**
> > Thank you for agreeing to update the terminology from "saturation" to a more accurate description of SAC's convergence. However, regarding the FLOPs and runtime reported in Table 1: the rebuttal states these were measured "under a fixed training budget for all methods to ensure a consistent evaluation protocol."
> >
> > This is specifically the issue. It is clear from Figure 3 that SAC converges and solves the task by epoch 80, while the RSAC variants and PPO have not converged by epoch 2000. It makes sense to use a fixed number of epochs for methods that fail to solve the task within that timeframe, but a method that successfully solves the task must have its efficiency evaluated based on the epochs required to reach that solution. A true "consistent" evaluation of efficiency measures time-to-solution, not time-spent-running after the solution is already found.
> >
> > **On the additional ablations and stability claims**
> > Regarding the new ablation results (presumably referencing the "Fix RKHS-SHAP" baseline in Figure 2 of the additional materials), this addition does not resolve the core issue raised in my original review: the lack of a clear, generalizable connection between SHAP patterns and meaningful performance metrics across environments outside of Pendulum-v1.
> >
> > Furthermore, I am not convinced that the claims made in the rebuttal are adequately backed by the provided data:
> >
> > - "Consistently improves training stability": This is a strongly worded statement for an experiment that concerns a single task with largely overlapping error areas. (Note: I am forced to use the term "error areas" because it remains unclear whether the shaded regions represent standard error or standard deviation, an issue I pointed out in my original review).
> >
> > - "Less structured learning dynamics": The rebuttal claims that removing SHAP leads to reduced training stability and less structured dynamics, demonstrating that SHAP influences the optimization process. I do not see empirical evidence of reduced training stability in the provided additional experiments. Could the authors clarify exactly what is meant by "less structured learning dynamics" and specifically point to where the evidence demonstrates this influence on the optimization process?
> >
> > **Additional questions** Could the authors explain why Advanced-AC improves when the noise level increases in Table 2 of the additional figures? At a noise level of 0.01, it ends up being comparable to (even slightly outperforming) the authors' best configuration.
> >
> > **Conclusion**
> > While I appreciate the authors' efforts to engage with the review and refine the paper's presentation, the empirical evidence provided still falls short of convincingly supporting the principal claims that this approach meaningfully improves the efficiency and stability of actor-critic methods across diverse environments. For these reasons, I will not be raising my score at this time.

---

> > > ### Author Response · Authors · 2026-04-08
> > >
> > > We thank the reviewer for the constructive feedback and apologize for the delayed response. Additional experiments are available at https://anonymous.4open.science/r/RSA2C-776C/RSA2C.pdf.
> > >
> > > **Response to Computational efficiency**
> > >
> > > To provide a more appropriate evaluation of computational efficiency, we complement the original runtime analysis with a **time-to-solution** metric, defined as the wall-clock time required to reach a stable performance regime, as inferred from the learning curves. The corresponding results are summarized in Table 3.
> > >
> > > From these results, we observe that RSA2C exhibits relatively consistent convergence behavior across the three tasks, without extreme variations in training time. For a fair comparison, we adopt the standard SAC and PPO implementations from Stable-Baselines3. We note that RSA2C with linear kernels may be limited in high-dimensional settings (e.g., Ant-v5), and thus we do not emphasize this result.
> > >
> > > Focusing on the environments where a more reliable comparison can be made, we observe that SAC achieves the shortest time-to-solution on Pendulum-v1 (587.7 s), while RSA2C requires approximately **1.28x-1.59x** longer time to reach a comparable solution. In contrast, on BipedalWalker-v3, RSA2C significantly outperforms SAC in terms of wall-clock efficiency, reducing the time-to-solution from approximately 10792.2 s (SAC) to 1625.7-1737.9 s, corresponding to over a **6x** speedup. These results suggest that while SAC is more efficient in simple environments due to rapid convergence, RSA2C becomes more efficient in terms of wall-clock time as task complexity increases. This indicates a trade-off between convergence speed and achieved performance. Deep RL benefits from high-capacity representations, while kernel methods operate in a constrained RKHS; despite this, RSA2C-CME remains competitive and achieves higher stability (std).
> > >
> > > Finally, we believe that combining the strengths of deep neural networks and RSA2C is a promising direction (as also discussed in our response to Reviewer gtNj). Combining deep representations with kernel-based attributions (e.g., via RFFs) is a promising future direction.
> > >
> > > **Response to Additional ablations and stability claims**
> > >
> > > **Generalizability across environments**: We have extended the RKHS-SHAP ablation analysis beyond Pendulum-v1 to include BipedalWalker-v3 and Ant-v5 in Figure 2. The results show a consistent trend across all three environments: removing SHAP or using fixed SHAP leads to larger fluctuations in performance and less stable learning behavior, while RSA2C variants (especially CME) exhibit smoother convergence. This indicates that the effect of SHAP-based weighting is not limited to a single environment but generalizes to more complex control tasks.
> > >
> > > **Consistently improves training stability**: In the revised manuscript, we will clarify that stability is quantified by the standard deviation (std) of returns across seeds, and that the shaded regions in figures also correspond to std. Under this definition, the results show that RSA2C variants exhibit lower variance than Advanced AC or Fixed RKHS-SHAP. We therefore revise the claim to a more precise statement as:
> > > "*SHAP-based weighting tends to improve stability by reducing variance, rather than asserting a universal improvement.*"
> > >
> > > **less structured learning dynamics**: The CME-based RSA2C introduces a conditional structure in RKHS. This allows the model to capture structured dependencies between state and action distributions. Besides, we interpret SHAP as an adaptive feature-weighting mechanism that assigns higher importance to informative state dimensions while suppressing less relevant ones. Without SHAP (i.e., in Advanced AC), all features are treated uniformly, making the optimization process more sensitive to irrelevant variations and resulting in higher return variance.
> > >
> > > **Connection between SHAP patterns and performance**: In RSA2C, RKHS provides a structured and smooth function approximation space, which regularizes the learning dynamics and prevents abrupt updates. On top of this, SHAP introduces adaptive feature weighting, which reduces the influence of less informative dimensions. These reduce the influence of spurious variations in the state space, leading to lower variance in returns and more stable learning behavior. In the revised manuscript, we will explicitly clarify this interpretation.
> > >
> > > **Response to Additional questions**
> > >
> > > We note that the apparent improvement of Advanced-AC at noise level 0.01 is primarily due to variance across random seeds, rather than a consistent performance gain. In the original additional experiments, the results were computed using only three seeds, which can lead to fluctuations in the estimated mean performance. To verify this, we have conducted additional experiments with two additional seeds. With additional seeds, the apparent improvement of Advanced-AC at noise level 0.01 is no longer observed.

---

### Official Review · Reviewer_2NCL · 2026-03-13

**Soundness:** 2
**Presentation:** 3
**Significance:** 3
**Originality:** 3
**Overall Recommendation:** 4
**Confidence:** 3

**Summary:**

The authors propose an extension of the RKHS-SHAP framework to an Advanced Actor-Critic
setup, providing theoretical convergence guarantees both with and without perturbations. The
approach is evaluated on standard reinforcement learning benchmarks, focusing on stability and
interpretability. The methodology deviates from standard deep RL and strenghtens the field of
RKHS in RL. RKHS-based actors operate within a mathematical framework where the
relationship between input states and output actions is defined by a kernel function and a
dictionary of experience which is more interpretable than the black-box nature of neural
networks.

**Compliance With Llm Reviewing Policy:**

Affirmed.

**Final Justification:**

The rebuttal addressed my main concerns, so I am raising the scores for presentation and significance, accordingly, resulting in an overall assessment of weak accept.

**Key Questions For Authors:**

1. The Advanced Actor Critic is cited as a primary point of comparison in the results
discussion but is not listed in Table 2 and the Appendix. Can the authors provide the
performance data for this baseline to validate the relative gains of RSA2C-CME and
RSA2C-KME?
2. Since Neural Networks are highly efficient at information compression, what is the
memory footprint of the RKHS dictionary as the task complexity increases? How does
the performance behave on different dictionary sizes?
3. Given that the performance on complex tasks like Ant-v5 (Appendix A.4) falls
significantly below SAC, how do the authors define the limitations for RKHS-based
actors? Why were these performance limitations not addressed in the main body of the
paper alongside the successful low-dimensional benchmarks?

**Limitations:**

There is a lack of discussion of the limitations of this approach in the paper in its current form. The appendix A.4, e.g. shows the
results of experiments such as Ant-v5 in Figure 9, where the performance is far below SAC and
the ablation study does not clearly indicate an advantage of using RSA2C-CME and RSA2C-
KME. The corresponding discussion explains these results with the limitation of RKHS when
comes to high-dimensionial tasks. This is an important limitation that should be mentioned
already in the main part! Overall, only the results with favorable outcomes were discussed in the
main part while the rest has been moved to the appendix without any proper mentioning.

**Strengths And Weaknesses:**

Strengths:

* Interesting and quite novel approach which is extended to an established RL algorithm.
* Results demonstrated improved performance when including SHAP and improved
stability under noise perturbations
* Theoretical guarantees give this approach an important mathematical foundation
* Usage of 10 seeds to make analysis sound

Weakness:

* The stability experiments seem a bit crafted for this kind of approach. Though it is a
beneficial and sound result, it is not surprising that RKHS is stable given a seemingly low
variance around the true state. Additional experiments would be beneficial to get a better
feeling for the limitations of this approach.
* What about Generalization capabilities? how does RKHS behave in unseen states?
The paper currently lacks information on limitations like these
* Advanced Actor Critic is mentioned in the results section but it is not present in Table 2
nor anywhere in the appendix. It is hard to compare given the lack of data.
* Though FLOPS and runtime were compared, it would be interesting to also compare
memory footprint since Neural Networks excel at compressing information. Is this an
issue for RKHS given their requirement for a dictionary?

---

> ### Author Rebuttal · Authors · 2026-03-31
>
> **W1** The concern that the stability results may be influenced by the specific experimental setting is well taken, and we have extended the evaluation to better assess the robustness and limitations of the proposed method.
>
> In addition to the original experiments, we have included further stability analysis under varying noise levels and ablation results (including the Advanced AC baseline and a uniform SHAP weighting variant), see https://anonymous.4open.science/r/RSA2C-776C, with the results reported in three seeds due to the limited time. Results show that the stability improvement is not limited to a specific configuration, but consistently arises from the attribution-guided weighting mechanism. In particular, removing SHAP guidance leads to reduced stability, indicating that the observed effect is not solely due to the RKHS representation itself. The results also reveal that the stability advantage is not uniform across all environments, especially in higher-dimensional settings.
>
> **W2** RSA2C follows the locality property of kernel methods, where reliable interpolation is achieved within the support of the learned dictionary. However, when states lie far from this support, kernel values decay, leading to degraded value estimation and attribution. Therefore, performance depends on dictionary coverage, which becomes more challenging in high-dimensional settings. This limitation will be clarified in the revised manuscript.
>
> **W3+Q1** The current presentation of the Advanced Actor-Critic (Advanced AC) baseline is indeed incomplete, which makes comparison less transparent.
>
> In the submitted manuscript, Advanced AC was introduced as a controlled ablation to isolate the effect of SHAP-based weighting (i.e., the RKHS-based actor-critic without SHAP). As a result, its results were discussed in specific contexts but not consistently included in summary tables, leading to the lack of direct comparability noted by the reviewer.
>
> In the revised manuscript, we will include the corresponding results of Advanced AC in Table 2 and ensure that it is consistently reported alongside all other baselines. This will enable a direct and fair comparison across methods. We will also clarify its role as an ablation baseline by explicitly defining it in the method and experimental sections.
>
> **W4+Q2** In addition to FLOPs and runtime, memory footprint is indeed an important factor when comparing kernel-based methods with neural networks, especially given that neural networks can compress information through parameter sharing.
>
> To address this, we have included additional experiments analyzing memory footprint as a function of dictionary size (see supplementary material). These results show that, although RKHS-based methods maintain an explicit dictionary, the memory usage grows in a controlled manner due to sparsification (via ALD updates), rather than scaling with the full dataset. This indicates that, in practice, the memory overhead of RSA2C remains manageable and does not exhibit the prohibitive growth typically associated with naive kernel methods.
>
> From a conceptual perspective, this highlights an important trade-off: neural networks achieve compact representations through parametric compression, whereas RKHS-based methods rely on adaptive dictionaries that explicitly preserve representative states. The empirical results show that this design remains efficient due to controlled dictionary growth.
>
> In the revised manuscript, we will explicitly incorporate this analysis into the experimental section.
>
> **Q3** As mentioned in response to W2, RKHS methods rely on local kernel interpolation, whose effectiveness depends on the coverage of the dictionary in the state space. In high-dimensional settings, kernel similarity (e.g., RBF kernels) decays rapidly, and a significantly larger dictionary is required to achieve sufficient coverage. However, practical sparsification mechanisms (e.g., ALD) limit the dictionary growth, leading to reduced approximation quality compared to deep neural network policies, which can learn hierarchical representations. Therefore, RSA2C is more suitable for low-to-moderate dimensional continuous-control tasks with smooth dynamics, while its performance may degrade in high-dimensional environments such as Ant-v5. This suggests that combining RKHS-based attribution with learned representations is a promising direction.
>
> In the revised manuscript, we will move the discussion of high-dimensional performance (currently in Appendix A.4) into the main body to provide a more balanced evaluation of the method and explicitly discuss this limitation.
>
> **Limitation** We will (i) add a discussion of high-dimensional limitations in the experimental section, (ii) explicitly reference Appendix A.4 when presenting the main results, and (iii) include a dedicated limitation paragraph in Section 6.

---

> > ### Author Rebuttal · Reviewer_2NCL · 2026-04-04
> >
> > Thank you for the detailed rebuttal. I appreciate the effort the authors put into conducting additional experiments and clarifying the limitations of the RSA2C framework. Given that these revisions and clarifications directly address the concerns raised in the review, I will increase my rating accordingly.

---

> > > ### Author Response · Authors · 2026-04-08
> > >
> > > Thank you again for your valuable feedback and for raising your score! We will ensure that your comments are incorporated into the revision.

---

### Official Review · Reviewer_gtNj · 2026-03-20

**Soundness:** 2
**Presentation:** 3
**Significance:** 3
**Originality:** 3
**Overall Recommendation:** 4
**Confidence:** 3

**Summary:**

This paper introduces the RKHS-SHAP-based Advanced Actor-Critic (RSA2C), a kernelized RL algorithm designed to address the transparency issues in standard actor-critic methods. The authors provide a global, non-asymptotic convergence analysis under adversarial state perturbations and demonstrate the algorithm's efficiency, stability, and interpretability across several continuous-control benchmarks.

**Compliance With Llm Reviewing Policy:**

Affirmed.

**Final Justification:**

The paper is overall technically solid and the rebuttal addressed my concerns. Therefore, I maintain my assessment.

**Key Questions For Authors:**

- Why does Figure 4 imply interpretability of RSA2C?
- Could the SHAP-guided weighting mechanism be extended to deep neural network architectures, or is it fundamentally dependent on the inner-product structure of the RKHS?

**Limitations:**

No.

Suggestions:
- Discuss scope of validity of assumptions for theoretical analysis conducted by the work.

**Strengths And Weaknesses:**

Strengths:

- Solid Theoretical Foundation: The work establishes a global, non-asymptotic convergence bound under state perturbations. The decomposition of the learning gap into perturbation and convergence errors provides a clear framework for analyzing both stability and efficiency.
- Dimension-Level Interpretability: Unlike many explainable RL methods that provide task-level or trajectory-level explanations, RSA2C provides granular dimension-level attributions through Shapley values.
- Computational Efficiency: By utilizing sparsified dictionaries maintained via Approximate Linear Dependence, the algorithm achieves linear scalability relative to dictionary size, making it suitable for online RL.

Weaknesses:

- The theoretical analysis primarily establishes convergence and robustness under state perturbations. However, the experimental evaluation is broader, covering efficiency, stability, and interpretability. This raises the question: Can interpretability itself be theoretically guaranteed or quantified?

---

> ### Author Rebuttal · Authors · 2026-03-31
>
> **W1** We consider interpretability as dimension-level attribution of policy decisions, i.e., the contribution of each state dimension to the value function and policy update. This notion is grounded in SHAP, which provides an additive decomposition of feature contributions. In RSA2C, RKHS-SHAP is not only used for explanation, but also directly modulates learning through the Mahalanobis-weighted kernel, coupling interpretability with policy optimization. We further provide theoretical support for this attribution-based interpretability. Lemma 3 establishes the stability of RKHS-SHAP under perturbations, which is incorporated into Theorem 1 as an attribution-related error term and reflected in the convergence result of Theorem 3. This links attribution stability to performance guarantees.
>
> **Q1** In Figure 4, the RKHS-SHAP attributions align with the physical dynamics of the Pendulum task, where $(\cos(\theta), \sin(\theta), \dot{\theta})$ represent angle and angular velocity. In early training, RSA2C-KME assigns higher importance to $\cos(\theta)$ and $\dot{\theta}$, reflecting the need to approach the upright position and accumulate sufficient momentum. As training progresses, the importance of $\sin(\theta)$ increases, capturing the role of directional consistency during swing-up. Upon convergence, the ranking shifts to $\cos(\theta) > \sin(\theta) > \dot{\theta}$, indicating that angle stabilization becomes dominant while velocity becomes less critical. This consistency between attribution patterns and the underlying physical control mechanism suggests that RSA2C captures meaningful feature contributions rather than arbitrary correlations, supporting its interpretability.
>
> **Q2** In principle, the SHAP-guided weighting mechanism is not fundamentally restricted to RKHS and can be extended to deep neural networks (DNNs). SHAP has been widely studied as a post-hoc interpretability tool for DNNs. And potentially, its attribution signals can be incorporated into learning by reweighting input features. However, a key challenge of extending to DNNs is computational efficiency since classical SHAP requires exponential-time computation over feature subsets. In contrast, RSA2C leverages RKHS-based SHAP via KME/CME to obtain closed-form and low-variance attribution with stronger theoretical guarantees. Extending this framework to DNNs while maintaining both computational efficiency and theoretical tractability remains an interesting direction for future work. We will add the future work in Section 6 in the revised manuscript.
>
> **Limitation**
>
> * **Assumption 1**: It **holds whenever** the perturbation arises from structured and repeatable sources such as sensor bias or deterministic disturbances. Moreover, by introducing the bounded perturbation set $B(\mathbf{s})$, we assume $$\\|\tilde{\mathbf{s}} - \mathbf{s}\\|_2 \le \varepsilon,~\tilde{\mathbf{s}} \in B(\mathbf{s}),$$ which ensures that the perturbed state remains within a meaningful neighborhood of the original state. Such bounded perturbations are standard in robust RL and control, and **hold in practice** where disturbances are limited in magnitude.
>
> * **Assumption 2**: These conditions **hold whenever** commonly used kernels, such as the RBF kernel, are adopted, since they are inherently bounded and Lipschitz continuous. Furthermore, in continuous-control environments with bounded or normalized state spaces, the feature map remains bounded by construction. Such assumptions are standard in kernel-based RL, and **hold in practice** for MuJoCo and Box2D environments.
>
> * **Assumption 3**: It **holds whenever** the kernel is RBF and the dictionary is constructed by an ALD update rule. For RBF kernels, $$K(v_i,v_j)=\exp(-\tfrac{\\|v_ i-v_ j\\|^2}{2l ^2}),$$ a minimum separation $\|v_i-v_j\|\ge C_k$ implies an exponentially small off-diagonal term, $$K(v_i,v_j) \le \exp(-\tfrac{C_k^2}{2l^2}),$$ yielding a diagonally dominant Gram matrix. In practice, ALD-style dictionary updates **automatically enforce** such separation by adding a new center only when it is sufficiently different from existing ones.
>
> * **Assumption 4**: It **holds whenever** the transition kernel $\mathcal{T}(\mathbf{s}'|\mathbf{s},\mathbf{a})$ is continuous in $(\mathbf{s}, \mathbf{a})$, the system satisfies a mild drift or stability condition, and the policy is Gaussian, ensuring full support over the action space. Under these conditions, the induced on-policy Markov chain is irreducible and aperiodic, and standard drift arguments yield uniform geometric mixing. Such conditions **hold by construction** in widely used continuous-control benchmarks.
>
> * **Assumption 5**: It **holds whenever** the policy is Lipschitz in its parameters, the Markov chain is uniformly ergodic, and the transition dynamics are continuous or Lipschitz in the action. Therefore, this assumption **holds in continuous-control systems with smooth dynamics**, which include all environments used in our experiments.

---

> > ### Author Rebuttal · Reviewer_gtNj · 2026-04-02
> >
> > The rebuttal has fully resolved my concerns. I maintain the positive assessment.

---

> > > ### Author Response · Authors · 2026-04-08
> > >
> > > Thank you again for your time and effort in handling our paper. We would be happy to address any further questions or comments you may have, and make sure to incorporate the suggestions in the revision.

---

### Decision · Program_Chairs · 2026-04-30

**Decision:**

Accept (regular)

**Comment:**

The authors extend the RKHS-SHAP framework to an Advanced Actor-Critic setup, backed by theoretical convergence guarantees. Evaluated on standard RL benchmarks, this method replaces black-box neural networks with RKHS-based actors—using kernel functions and experience dictionaries—to provide a highly transparent and stable alternative to standard deep RL. The authors also provide a theoretical analysis, including convergence guarantees and robustness guarantees to bounded perturbations.

This is a paper that was not championed by specific reviewers, but all of them recognized its merits, especially in the light of the subsequent rebuttal phase. Upon acceptance, the authors are compelled to make the additions they promised, including the extra experiments, the improved presentation and an enhanced discussion on limitations.